# Sequence action representations contextualize during early skill learning

**Debadatta Dash[1], Fumiaki Iwane[1], William Hayward[1], Roberto F Salamanca-Giron[1], Marlene Bönstrup[2], Ethan R Buch[1]\*, Leonardo G Cohen[1]\***

[1]Human Cortical Physiology and Neurorehabilitation Section, NINDS, NIH, Bethesda, United States; [2]Department of Neurology, University of Leipzig Medical Center, Leipzig, Germany

## eLife Assessment

This **valuable** study asks how the neural representation of individual finger movements changes during the early periods of sequence learning. By combining a new method for extracting features from human magnetoencephalography data and decoding analyses, the authors provide **solid** evidence of an early, swift change in the brain regions correlated with sequence learning, including a set of previously unreported frontal cortical regions. The authors also show that offline contextualization during short rest periods is the basis for improved performance. Further confirmation of these results on multiple movement sequences would further strengthen the key claims.

**\*For correspondence:**
ethan.buch@nih.gov (ERB);
cohenl@ninds.nih.gov (LGC)

**Competing interest:** The authors declare that no competing interests exist.

**Abstract** Activities of daily living rely on our ability to acquire new motor skills composed of precise action sequences. Here, we asked in humans if the millisecond-level neural representation of an action performed at different contextual sequence locations within a skill differentiates or remains stable during early motor learning. We first optimized machine learning decoders predictive of sequence-embedded finger movements from magnetoencephalographic (MEG) activity. Using this approach, we found that the neural representation of the same action performed in different contextual sequence locations progressively differentiated—primarily during rest intervals of early learning (offline)—correlating with skill gains. In contrast, representational differentiation during practice (online) did not reflect learning. The regions contributing to this representational differentiation evolved with learning, shifting from the contralateral pre- and post-central cortex during early learning (trials 1–11) to increased involvement of the superior and middle frontal cortex once skill performance plateaued (trials 12–36). Thus, the neural substrates supporting finger movements and their representational differentiation during early skill learning differ from those supporting stable performance during the subsequent skill plateau period. Representational contextualization extended to Day 2, exhibiting specificity for the practiced skill sequence. Altogether, our findings indicate that sequence action representations in the human brain contextually differentiate during early skill learning, an issue relevant to brain-computer interface applications in neurorehabilitation.

## Introduction

Motor learning is required to perform a wide array of activities of daily living, intricate athletic endeavors, and professional skills. Whether it's learning to type more quickly on a keyboard (**Bönstrup et al., 2019a**), improve one's tennis game (**Schmidt, 2018**), or play a piece of music on the piano (**Doyon and Benali, 2005**) – all these skills require the ability to execute sequences of actions with precise temporal coordination. Action sequences thus form the building blocks of fine motor skills (**Dehaene et al., 2015**). Practicing a new motor skill elicits rapid performance improvements (early

learning; *Bönstrup et al., 2019a*) that precede skill performance plateaus (*Walker and Stickgold, 2004*). Skill gains during early learning accumulate over rest periods (micro-offline) interspersed with practice (*Bönstrup et al., 2019a*; *Buch et al., 2021*; *Jacobacci et al., 2020*; *Mylonas et al., 2024*; *Hayward et al., 2024*; *Brooks et al., 2024*), and are up to four times larger than offline performance improvements reported following overnight sleep (*Bönstrup et al., 2019a*). During this initial interval of prominent learning, retroactive interference immediately following each practice interval reduces learning rates relative to interference after passage of time, consistent with stabilization of the motor memory (*Bönstrup et al., 2020*). Micro-offline gains observed during early learning are reproducible (*Jacobacci et al., 2020*; *Brooks et al., 2024*; *Bönstrup et al., 2020*; *Chen et al., 2024*; *Sjøgård, 2024*) and are similar in magnitude even when practice periods are reduced by half to 5 seconds in length, thereby confirming that they are not merely a result of recovery from performance fatigue (*Bönstrup et al., 2020*). Additionally, they are unaffected by the random termination of practice periods, which eliminates the possibility of predictive motor slowing as a contributing factor (*Bönstrup et al., 2020*). Collectively, these behavioral findings point towards the interpretation that micro-offline gains during early learning represent a form of memory consolidation (*Bönstrup et al., 2019a*).

This interpretation has been further supported by brain imaging and electrophysiological studies linking known memory-related networks and consolidation mechanisms to rapid offline performance improvements. In humans, the rate of hippocampo-neocortical neural replay predicts micro-offline gains (*Buch et al., 2021*). Consistent with these findings, *Chen et al., 2024* and *Sjøgård, 2024* furnished direct evidence from intracranial human EEG studies, demonstrating a connection between the density of hippocampal sharp-wave ripples (80–120 Hz)—recognized markers of neural replay— and micro-offline gains during early learning. Further, Griffin et al. reported that neural replay of task-related ensembles in the motor cortex of macaques during brief rest periods—akin to those observed in humans (*Bönstrup et al., 2019a*; *Buch et al., 2021*; *Jacobacci et al., 2020*; *Mylonas et al., 2024*; *Wamsley et al., 2023*)—is not merely correlated with, but are causal drivers of micro-offline learning (*Griffin et al., 2025*). Specifically, the same reach directions that were replayed the most during rest breaks showed the greatest reduction in path length (i.e. more efficient movement path between two locations in the reach sequence) during subsequent trials, while stimulation applied during rest intervals preceding performance plateau reduced reactivation rates and virtually abolished micro-offline gains (*Griffin et al., 2025*). Thus, converging evidence in humans and non-human primates across indirect non-invasive and direct invasive recording techniques links hippocampal activity, neural replay dynamics, and offline skill gains in early motor learning that precede performance plateau.

During skill learning, the neural representation of a sequential skill binds discrete individual actions (e.g. single piano keypress) into complex, temporally and spatially precise sequence representations (e.g. a refrain from a piece of music; *Karni et al., 1995*; *Song and Cohen, 2014*; *Natraj et al., 2022*; *Ghilardi et al., 2009*; *Yokoi and Diedrichsen, 2019*). After a skill is learned over extended periods (i.e. weeks), the neural representation of the sequence changes significantly (*Yokoi and Diedrichsen, 2019*), while the representation of its individual action components (e.g. finger movements) does not (*Beukema et al., 2019*). On the other hand, it is not known whether individual sequence action representations differentiate or remain stable during the early stages of skill learning, when the memory is still not fully formed (*Bönstrup et al., 2019a*). Furthermore, it is unknown whether the neural representations of identical movements, performed at different positions within a skill sequence (i.e. the skill *context*), differentiate with learning—an important consideration for advancing robust brain-computer interface (BCI) applications (*Merino et al., 2023*; *Liu et al., 2023*; *Lee et al., 2022*; *Zhao et al., 2022*; *Yao et al., 2022*).

Examining the millisecond-level differentiation of discrete action representations during learning is challenging, as evolving neural dynamics concurrently encode skill sequences and their individual action components (*Yokoi and Diedrichsen, 2019*; *Hikosaka et al., 1999*) across multiple spatial scales (*Munn et al., 2024*). To address this problem, we first optimized a multi-scale decoder aimed at predicting keypress actions from magnetoencephalographic (MEG) neural activity. Using this optimized approach, we report that an individual sequence action representation differentiates depending on the sequence context and correlates with early skill learning. This representational contextualization developed predominantly over rest rather than during practice intervals—in parallel with rapid consolidation of skill.

# Results

Participants engaged in a well-characterized sequential skill learning task (***Bönstrup et al., 2019a***; ***Buch et al., 2021***; ***Bönstrup et al., 2020***) that involved repetitive typing of a sequence (4-1-3-2-4) performed with their (non-dominant) left hand over 36 trials with alternating periods of 10 s *practice* and 10 s *rest* (*inter-practice rest; Day 1 Training*; ***Figure 1A***), a practice schedule that minimizes reactive inhibition effects (***Bönstrup et al., 2020***; ***Pan and Rickard, 2015***; see Materials and methods). Individual keypress times and finger keypress identities were recorded and used to quantify skill as the correct sequence speed (keypresses/s; ***Bönstrup et al., 2019a***).

Participants reached 95% of maximal skill (i.e., - Early Learning) within the initial 11 practice trials (***Figure 1B***), with improvements developing over inter-practice rest periods (micro-offline gains) accounting for almost all total learning across participants (***Figure 1B, inset***; ***Bönstrup et al., 2019a***). In addition to the reduction in sequence duration during early learning, individual keypress transition times became more consistent across repeated sequence iterations (***Figure 1C***). On average across subjects, 2.32% ± 1.48% (mean ± SD) of all keypresses performed were errors, which were evenly distributed across the four possible keypress responses. While errors increased progressively over practice trials, they did so in proportion to the increase in correct keypresses, so that the overall ratio of correct-to-incorrect keypresses remained stable over the training session.

On the following day, participants were retested on performance of the same sequence (4-1-3-2-4) over 9 trials (*Day 2 Retest*), as well as on the single-trial performance of 9 different untrained control sequences (*Day 2 Controls*: 2-1-3-4-2, 4-2-4-3-1, 3-4-2-3-1, 1-4-3-4-2, 3-2-4-3-1, 1-4-2-3-1, 3-2-4-2-1, 3-2-1-4-2, and 4-2-3-1-4). As expected, an upward shift in performance of the trained sequence (0.68 ± SD 0.56 keypresses/s; *t*=7.21, p<0.001) was observed during *Day 2 Retest*, indicative of an overnight skill consolidation effect (***Figure 1—figure supplement 1A***).

## Keypress actions are represented in multi-scale hybrid-space manifolds

We investigated the differentiation of neural representations of the same index finger keypress performed at different positions of the skill sequence. A set of decoders was constructed to predict keypress actions from MEG activity as a function of both the learning state and the ordinal position of the keypress within the sequence. We first characterized the spectral and spatial features of keypress state representations by comparing performance of decoders constructed around broadband (1–100 Hz) or narrowband [delta- (1–3 Hz), theta- (4–7 Hz), alpha- (8–14 Hz), beta- (15–24 Hz), gamma- (25–50 Hz), and high gamma-band (51–100 Hz)] MEG oscillatory activity. We found that decoders trained on broadband activity consistently outperformed those trained on narrowband activity. Whole-brain parcel-space (70.11% ± SD 7.11% accuracy; n=148 brain regions; *t*=1.89, p=0.035, df = 25, Cohen's *d*=0.17, ***Figure 2A***; also see ***Figure 2B*** for topographic map of feature importance scores) and voxel-space (74.51% ± SD 7.34% accuracy; n=15684; *t*=7.18, p<0.001, df = 25, Cohen's *d*=0.76, ***Figure 2A***; also see ***Figure 2C*** for topographic map of feature importance scores; ***Destrieux et al., 2010***) decoders exhibited greater accuracy than all regional voxel-space decoders constructed from individual brain areas (***Figure 2D***; maximum accuracy of 68.77% ± SD 7.6%; see also ***Figure 2—figure supplements 1 and 2***). Thus, optimal decoding required information from multiple brain regions, predominantly contralateral to the hand engaged in the skill task (***Figure 2B and C***).

Next, given that the brain simultaneously processes information more efficiently across multiple spatial and temporal scales (***Munn et al., 2024***; ***Buch et al., 2017***; ***Lisman and Buzsáki, 2008***), we asked if the combination of lower resolution whole-brain and higher resolution regional brain activity patterns further improve keypress prediction accuracy. We constructed hybrid-space decoders (N=1295 ± 20 features; ***Figure 3A***) combining whole-brain parcel-space activity (n=148 features; ***Figure 2B***) with regional voxel-space activity from a data-driven subset of brain areas (n=1147 ± 20 features; ***Figure 2D***). This subset covers brain regions showing the highest regional voxel-space decoding performances (top regions across all subjects shown in ***Figure 2D***; Materials and methods – *Hybrid Spatial Approach*). Accuracy was higher for hybrid- (78.15% ± SD 7.03%; weighted mean F1 score of 0.78 ± SD 0.07) than for voxel- (74.51% ± SD 7.34%; paired *t*-test: *t*=6.30, p<0.001, df = 25, Cohen's *d*=0.39) and parcel-space decoders (70.11% ± SD 7.48%; paired *t*-test: *t*=12.08, p<0.001, df = 25, Cohen's *d*=0.98, ***Figure 3B***, ***Figure 3—figure supplements 1 and 6***). Note that while features from contralateral brain regions were more important for whole-brain decoding (in both parcel- and voxel-spaces), regional voxel-space decoders performed best for bilateral sensorimotor

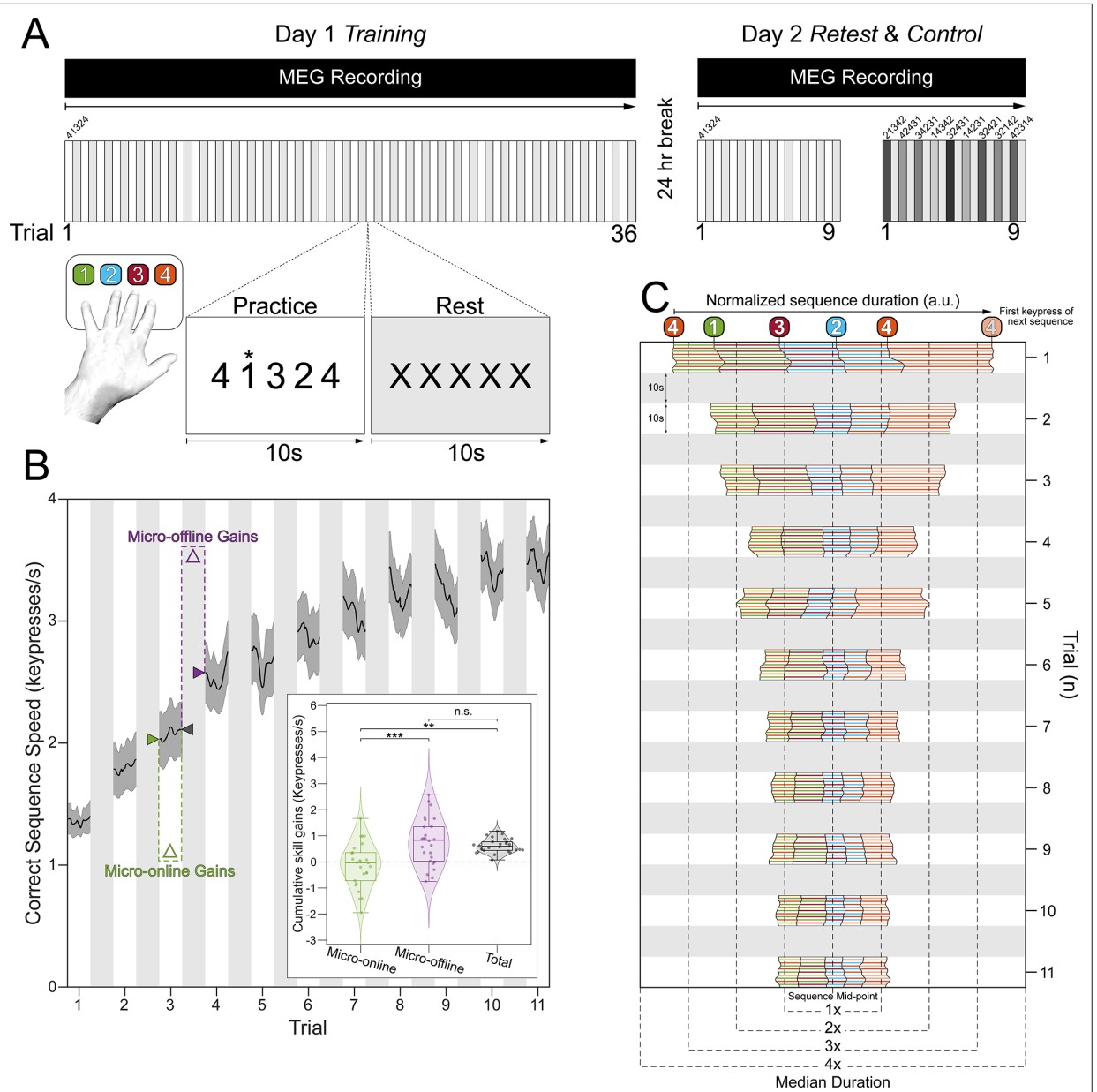

**Figure 1.** Experimental design and behavioral performance. (**A**) Skill learning task. Participants engaged in a procedural motor skill learning task, which required them to repeatedly type a keypress sequence, "4-1-3-2-4" (1=little finger, 2=ring finger, 3=middle finger, and 4=index finger) with their non-dominant, left hand. The *Day 1 Training* session included 36 trials, with each trial consisting of alternating 10 s practice and rest intervals. The rationale for this task design was to minimize reactive inhibition effects during the period of steep performance improvements (early learning; *Bönstrup et al., 2020*; *Pan and Rickard, 2015*; see Materials and methods). After a 24-hr break, participants were retested on performance of the same sequence (4-1-3-2-4) for nine trials (*Day 2 Retest*) to inform on the generalizability of the findings over time and MEG recording sessions, as well as single-trial performance on nine different control sequences (*Day 2 Control*; 2-1-3-4-2, 4-2-4-3-1, 3-4-2-3-1, 1-4-3-4-2, 3-2-4-3-1, 1-4-2-3-1, 3-2-4-2-1, 3-2-1-4-2, and 4-2-3-1-4) to inform on specificity of the findings to the learned skill. MEG was recorded during both Day 1 and Day 2 sessions with a 275-channel CTF magnetoencephalography (MEG) system (CTF Systems, Inc, Canada). (**B**) *Skill Learning*. As reported previously[1], participants on average reached 95% of peak performance by trial 11 of the *Day 1 Training* session (see *Figure 1—figure supplement 1A* for results over all *Day 1 Training* and *Day 2 Retest* trials). Shaded regions in main plot indicate the 95% confidence interval of the group mean. At the group level, total early learning was exclusively accounted for by micro-offline gains during inter-practice rest intervals (**B, inset**; $F_{[2,75]}=14.79$, $p=3.86 \times 10^{-6}$; micro-online vs. micro-offline: $p=7.98 \times 10^{-6}$; micro-online vs. total: $p=0.0002$; micro-offline vs. total: $p=0.669$). These results were not impacted by potential preplanning effects on initial skill performance (*Ariani and Diedrichsen, 2019*) since alternative measurements of cumulative micro-online and -offline gains remain unchanged after omission of the first 3 keypresses in each trial from the correct sequence speed computation (paired t-tests; micro-online: $t_{25}=-0.0223$, $p=0.982$; micro-offline: $t_{25}=-0.879$, $p=0.388$). Center line of box plots shown in inset indicate the group median, while box limits indicate the 1st and 3rd quartiles.

*Figure 1 continued on next page*

*Figure 1 continued*

Whisker lengths are set at the extreme value ≤1.5×IQR. (**C**) Keypress transition time (KTT) variability. Distribution of KTTs normalized to the median correct sequence time for each participant and centered on the mid-point for each full sequence iteration during early learning (see *Figure 1—figure supplement 1B* for results over all *Day 1 Training* and *Day 2 Retest* trials). Note the initial variability of the relative KTT composition of the sequence (i.e., – 4–1, 1–3, 3–2, 2–4, 4–4), before it stabilizes in the early learning period.

The online version of this article includes the following figure supplement(s) for figure 1:

**Figure supplement 1.** Behavioral performance during skill learning.

areas on average across the group. Thus, a multi-scale hybrid-space representation best characterizes the keypress action manifolds.

We implemented different dimensionality reduction or manifold extraction strategies including principal component analysis (PCA), multi-dimensional scaling (MDS), minimum redundant maximum relevance (MRMR), and linear discriminant analysis (LDA; *Maaten and Postma, 2009*) to map the input feature (parcel, voxel, or hybrid) space to a low-dimensional latent space (*Natraj et al., 2022*). LDA-based manifold extraction led to the greatest classifier performance gains, improving keypress decoding accuracy to 90.47% ± SD 3.44% (*Figure 3B*; weighted mean F1 score = 0.91 ± SD 0.05). In comparison to the hybrid-space decoder, whole-brain parcel-space decoder performance also improved following LDA-based dimensionality reduction (82.95% ± SD 5.48%), while whole-brain voxel-space decoder accuracy dropped substantially (40.38% ± SD 6.78%; also see *Figure 3—figure supplement 2*).

Notably, decoding of index finger keypresses (executed at two different ordinal positions in the sequence) exhibited the highest false negative (0.115 per keypress) and false positive (0.067 per prediction) misclassification rates compared with all other digits (false negative rate range = [0.067 0.114]; false positive rate range = [0.085 0.131]; *Figure 3C*), raising the hypothesis that the same action could be differentially represented when executed within different contexts (i.e. at different locations within the skill sequence). Testing the keypress state (4-class) hybrid decoder performance on Day 1 after randomly shuffling keypress labels for held-out test data resulted in a performance drop approaching expected chance levels (22.12% ± SD 9.1%; *Figure 3—figure supplement 3C*). An alternate decoder trained on ICA components labeled as movement or physiological artifacts (e.g. head movement, ECG, eye movements, and blinks; *Figure 3—figure supplement 3A, D*) and removed from the original input feature set during the pre-processing stage approached chance-level performance (*Figure 4—figure supplement 3*), indicating that the 4-class hybrid decoder results were not driven by task-related artifacts.

Utilizing the highest performing decoders that included LDA-based manifold extraction, we assessed the robustness of hybrid-space decoding over multiple sessions by applying it to data collected on the following day during the *Day 2 Retest* (9-trial retest of the trained sequence) and *Day 2 Control* (single-trial performance of 9 different untrained sequences) blocks. The decoding accuracy for *Day 2* MEG data remained high (87.11% ± SD 8.54% for the trained sequence during *Retest*, and 79.44% ± SD 5.54% for the untrained *Control* sequences; *Figure 3—figure supplement 4*). Thus, index finger classifiers constructed using the hybrid decoding approach robustly generalized from Day 1 to Day 2 across trained and untrained keypress sequences.

## Inclusion of keypress sequence context location optimized decoding performance

Next, we tracked the trial-by-trial evolution of keypress action manifolds as training progressed. Within-subject keypress neural representations progressively differentiated during early learning. A representative example in *Figure 4A* (top row) depicts increased four-digit representation clustering across trials 1, 11, and 36. The cortical representation of these clusters changed over the course of training, beginning with predominant involvement of contralateral pre-central areas in trial 1 before transitioning to greater contralateral post-central, superior frontal, and middle frontal cortex contributions in trials 11 and 36 (*Figure 4A*, bottom row), paralleling improvements in decoding performance (see *Figure 4—figure supplement 1* for trial-by-trial quantitative feature importance score changes during skill learning).

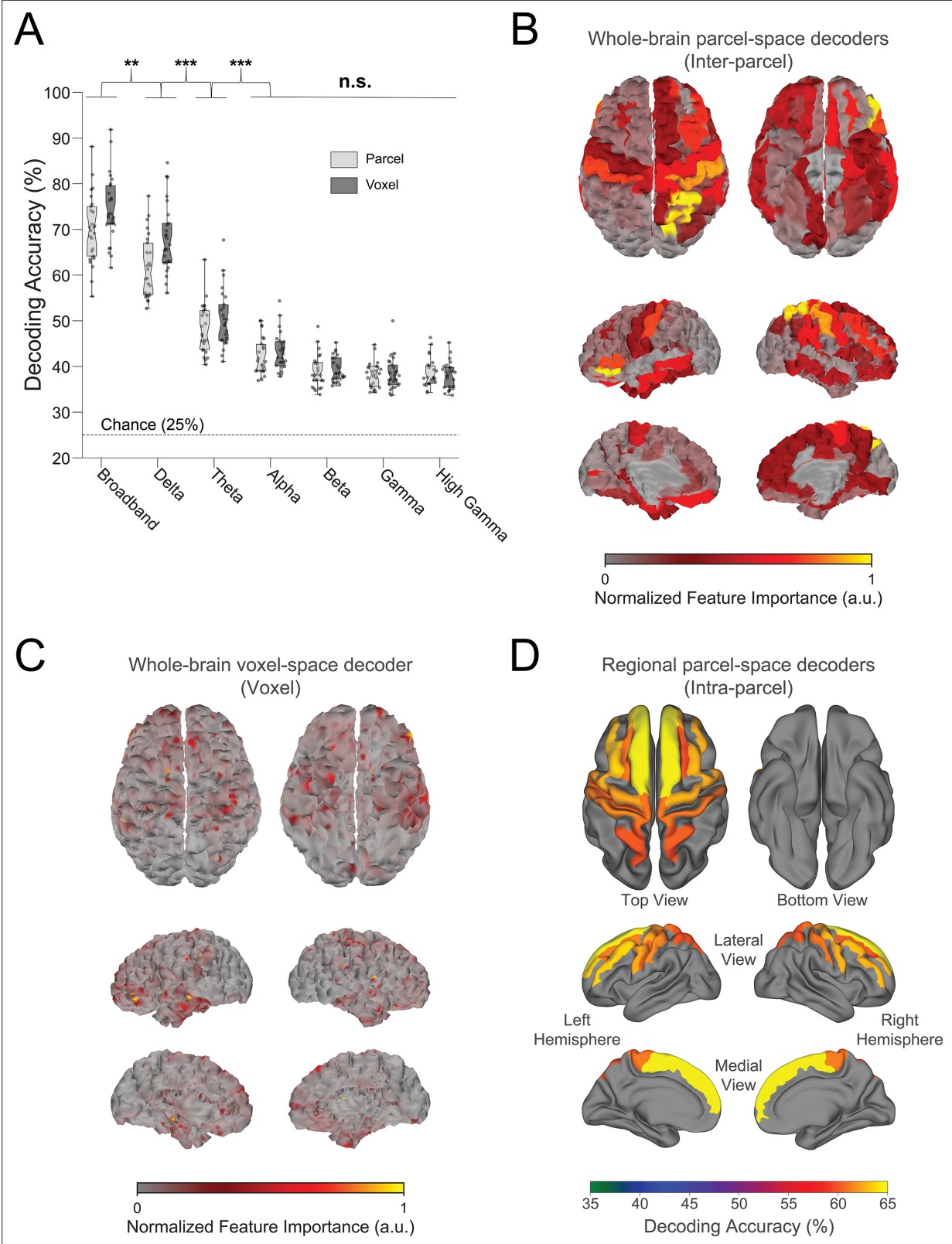

**Figure 2.** Spatial and oscillatory contributions to neural decoding of finger identities. (**A**) Contribution of whole-brain oscillatory frequencies to decoding. When trained on broadband activity relative to narrow frequency band features, decoding accuracy (i.e. test sample performance) was highest for whole-brain voxel-space (74.51% ± SD 7.34%, $t$=8.08, p<0.001) and parcel-space (70.11% ± SD 7.11%, $t$=13.22, p<0.001) MEG activity. Thus, decoders trained on whole-brain broadband data consistently outperformed those trained on narrowband activity. Dots depict decoding accuracy

*Figure 2 continued on next page*

*Figure 2 continued*

for each participant. Center line of box plots indicate the group median, while notches represent the 95% confidence interval of the group median. Box limits indicate the 1st and 3rd quartiles while whisker lengths are set at the extreme value ≤1.5×IQR. Outlier values located outside of the whisker range are marked with "+" symbols. *p<0.05, **p<0.01, ***p<0.001, n.s. - no statistical significance (p>0.05). (**B**) Whole-brain parcel-space decoding. Color-coded brain surface plot displaying the relative importance of individual brain regions (parcels) to broadband whole-brain parcel-space decoding performance (far-left light gray box plot in **A**). (**C**) Whole-brain voxel space decoding. Color-coded brain surface plot displaying the relative importance of individual voxels to broadband whole-brain voxel-space decoding performance (far-left dark gray box plot in **A**). (**D**) Regional voxel-space decoding. Broadband voxel-space decoding performance for top-ranked brain regions across the group is displayed on a standard (FreeSurfer fsaverage) brain surface and color-coded by accuracy. Note that while whole-brain parcel- and voxel-space decoders relied more on information from brain regions contralateral to the engaged hand, regional voxel-space decoders performed similarly for bilateral sensorimotor regions.

The online version of this article includes the following figure supplement(s) for figure 2:

**Figure supplement 1.** Oscillatory contributions at individual brain regions.

**Figure supplement 2.** Distribution of correlation coefficients between parcel-space time-series and their constituent voxels.

The trained skill sequence required pressing the index finger twice (**4**-1-3-2-**4**) at two contextually different ordinal positions (sequence positions 1 and 5). Inclusion of sequence location information (i.e. sequence context) for each keypress action (five sequence elements with the one keypress represented twice at two different locations) improved decoding accuracy ($t$=7.09, p<0.001, df = 25, Cohen's $d$=0.86, *Figure 4B*) from 90.47% (± SD 3.44%) to 94.15% (± SD 4.84%; weighted mean F1 score: 0.94), and reduced overall misclassifications by 54.3% (from 219 to 119; *Figures 3C and 4B*). The improved decoding accuracy is supported by greater differentiation in neural representations of the index finger keypresses performed at positions 1 and 5 of the sequence (*Figure 4A*), and by the trial-by-trial increase in 2-class decoding accuracy over early learning (*Figure 4C*) across different decoder window durations (*Figure 4—figure supplement 2*). As expected, the 5-class hybrid-space decoder performance approached chance levels when tested with randomly shuffled keypress labels (18.41% ± SD 7.4% for Day 1 data; *Figure 4—figure supplement 3C*). Task-related eye movements did not explain these results since an alternate 5-class decoder constructed from three eye movement features (gaze position at the KeyDown event, gaze position 200ms later, and peak eye movement velocity within this window; *Figure 4—figure supplement 3A*) performed at chance levels (cross-validated test accuracy = 0.2181; *Figure 4—figure supplement 3B, C*).

On Day 2, incorporating contextual information into the hybrid-space decoder enhanced classification accuracy for the trained sequence only (improving from 87.11% for 4-class to 90.22% for 5-class), while performing at or below-chance levels for the control sequences (≤30.22% ± SD 0.44%). Thus, the accuracy improvements resulting from inclusion of contextual information in the decoding framework were specific to the trained skill sequence.

## Neural representation of keypress sequence location diverged during early skill learning

We used a Euclidean distance measure to evaluate the differentiation of the neural representation manifold of the same action (i.e. an index-finger keypress) executed within different local sequence contexts (i.e. ordinal position 1 vs. ordinal position 5; *Figure 5*). To make these distance measures comparable across participants, a new set of classifiers was then trained with group-optimal parameters (i.e. broadband hybrid-space MEG data with subsequent manifold extraction *Figure 3—figure supplement 2*) and LDA classifiers (*Figure 3—figure supplement 7*) trained on 200ms duration windows aligned to the KeyDown event (see Materials and methods, *Figure 3—figure supplement 5*).

The Euclidean distance between neural representations of Index$_{OP1}$ (i.e. index finger keypress at ordinal position 1 of the sequence) and Index$_{OP5}$ (i.e. index finger keypress at ordinal position 5 of the sequence) increased progressively during early learning (*Figure 5A*)—predominantly during rest intervals (*offline contextualization*) rather than during practice (*online*) ($t$=4.84, p<0.001, df = 25, Cohen's $d$=1.2; *Figure 5B*; *Figure 5—figure supplement 1A*). An alternative online contextualization determination equaling the time interval between online and offline comparisons (*Trial-based;* 10 s between Index$_{OP1}$ and Index$_{OP5}$ observations in both cases) rendered a similar result (*Figure 5—figure supplement 2B*).

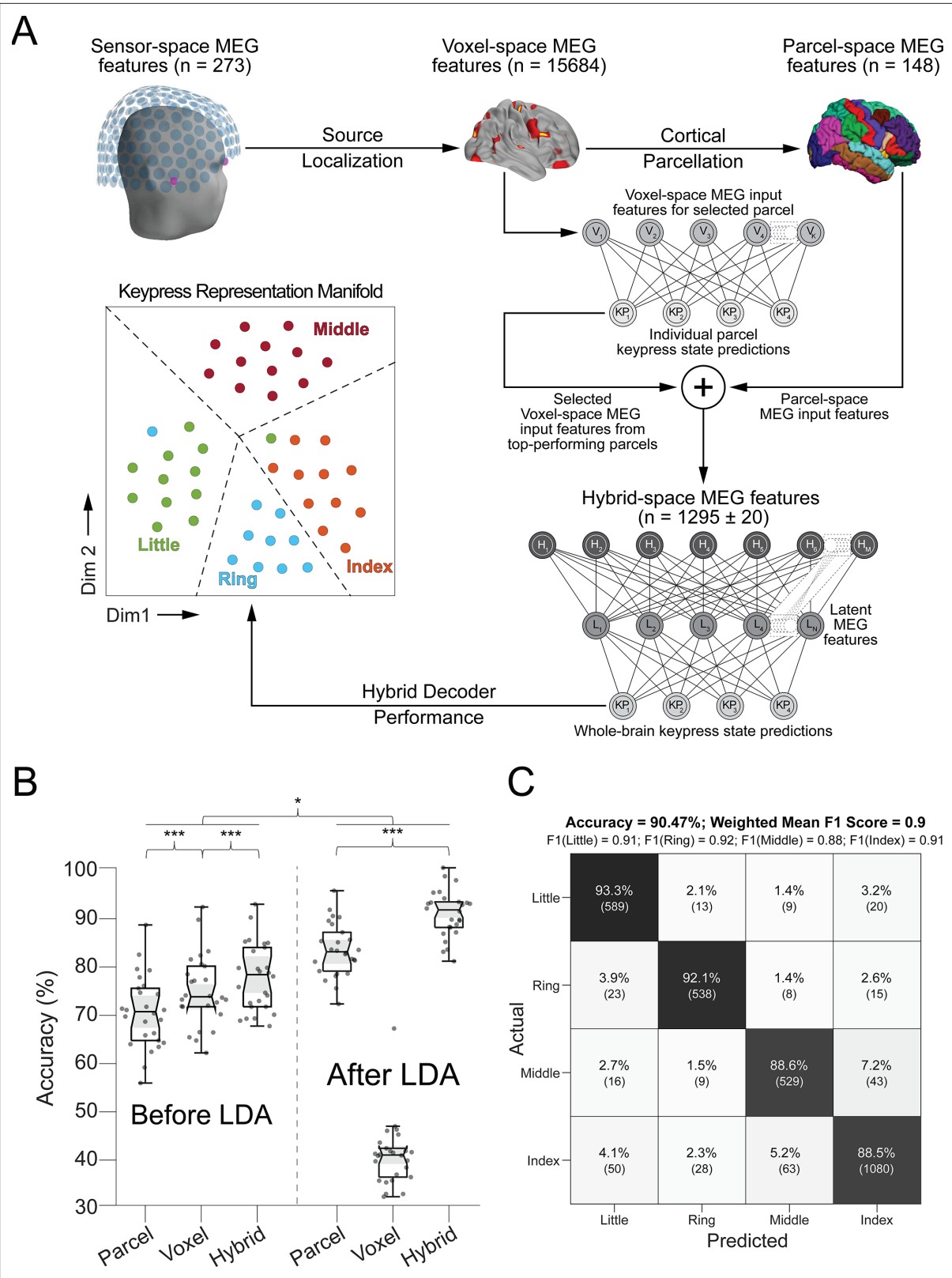

**Figure 3.** Hybrid spatial approach for neural decoding during skill learning. (**A**) Pipeline. Sensor-space MEG data (*N*=272 channels) were source-localized (voxel-space features; *N*=15,684 voxels), and then parcellated (parcel-space features; *N*=148) by averaging the activity of all voxels located within an individual region defined in a standard template space (Desikan-Killiany Atlas). Individual regional voxel-space decoders were then constructed and ranked. The final hybrid-space keypress state (i.e. 4-class) decoder was constructed using all whole-brain parcel-space features and

*Figure 3 continued*

top-ranked regional voxel-space features (see Materials and methods). (**B**) Decoding performance across parcel, voxel, and hybrid spaces. Note that decoding performance was highest for the hybrid space approach compared to performance obtained for whole-brain voxel- and parcel spaces. Addition of linear discriminant analysis (LDA)-based dimensionality reduction further improved decoding performance for both parcel- and hybrid-space approaches. Each dot represents accuracy for a single participant and method. Center line of box plots indicates the group median, while notches (and shaded areas) represent the 95% confidence interval of the group median. Box limits indicate the 1st and 3rd quartiles while whisker lengths are set at the extreme value ≤1.5×IQR. Outlier values located outside of the whisker range are marked with "+" symbols. ***p<0.001 and *p<0.05. (**C**) Confusion matrix of individual finger identity decoding for hybrid-space manifold features. True predictions are located on the main diagonal. Off-diagonal elements in each row depict false-negative predictions for each finger, while off-diagonal elements in each column indicate false-positive predictions. Please note that the index finger keypress had the highest false-negative misclassification rate (11.55%).

The online version of this article includes the following figure supplement(s) for figure 3:

**Figure supplement 1.** Contribution of whole-brain oscillatory frequencies to decoding.

**Figure supplement 2.** Comparison of different dimensionality reduction techniques.

**Figure supplement 3.** ICA artefacts do not contribute to decoding.

**Figure supplement 4.** Confusion matrices for decoding performance on *Day 2 Retest* (**A**) and *Day 2 Control* (**B**) data.

**Figure supplement 5.** Decoding performance across temporal scales.

**Figure supplement 6.** Comparison of decoding performances with two different hybrid approaches.

**Figure supplement 7.** Comparison of different decoder methods.

Offline contextualization strongly correlated with cumulative micro-offline gains ($r$=0.903, $R^2$=0.816, p<0.001; *Figure 5—figure supplement 1A*, inset) across decoder window durations ranging from 50 to 250 ms (*Figure 5—figure supplement 1B, C*). The offline contextualization between the final sequence of each trial and the second sequence of the subsequent trial (excluding the first sequence) yielded comparable results. This indicates that pre-planning at the start of each practice trial did not directly influence the offline contextualization measure (*Ariani and Diedrichsen, 2019*; *Figure 5—figure supplements 2A 1st vs. 2nd Sequence approaches*). Conversely, online contextualization (using either measurement approach) did not explain early online learning gains (i.e. *Figure 5—figure supplement 3*). Within-subject correlations were consistent with these group-level findings. The average correlation between offline contextualization and micro-offline gains within individuals was significantly greater than zero (*Figure 5—figure supplement 4*, left; $t$=3.87, p=0.00035, df = 25, Cohen's $d$=0.76) and stronger than correlations between online contextualization and either micro-online (*Figure 5—figure supplement 4*, middle; $t$=3.28, p=0.0015, df = 25, Cohen's $d$=1.2) or micro-offline gains (*Figure 5—figure supplement 4*, right; $t$=3.7021, p=5.3013e-04, df = 25, Cohen's $d$=0.69). These findings were not explained by behavioral changes of typing rhythm ($t$=–0.03, p=0.976; *Figure 5—figure supplement 5*), adjacent keypress transition times ($R^2$=0.00507, F [1,3202]=16.3; *Figure 5—figure supplement 6*), or overall typing speed (between-subject; $R^2$=0.028, p=0.41; *Figure 5—figure supplement 7*).

Finally, contextualization of Index$_{OP1}$ vs. Index$_{OP5}$ representations observed on *Day 1* generalized to *Day 2 Retest* of the trained skill sequence. Distances between representations for the same keypress performed twice within untrained sequences were lower in magnitude (*Day 2 Control*)—pointing to specificity of the contextualization effect (*Figure 5C*).

## Discussion

The main findings of this study during which subjects engaged in a naturalistic, self-paced task were that individual sequence action representations differentiate during early skill learning in a manner reflecting the local sequence context in which they were performed, and that the degree of representational differentiation—particularly prominent over rest intervals—correlated with skill gains.

### Optimizing decoding of sequential finger movements from MEG activity

The initial phase of the study focused on optimizing the accuracy of decoding individual finger keypresses from MEG brain activity. Recent work showed that the brain simultaneously processes information more efficiently across multiple—rather than a single—spatial scale(s) (*Munn et al.,*

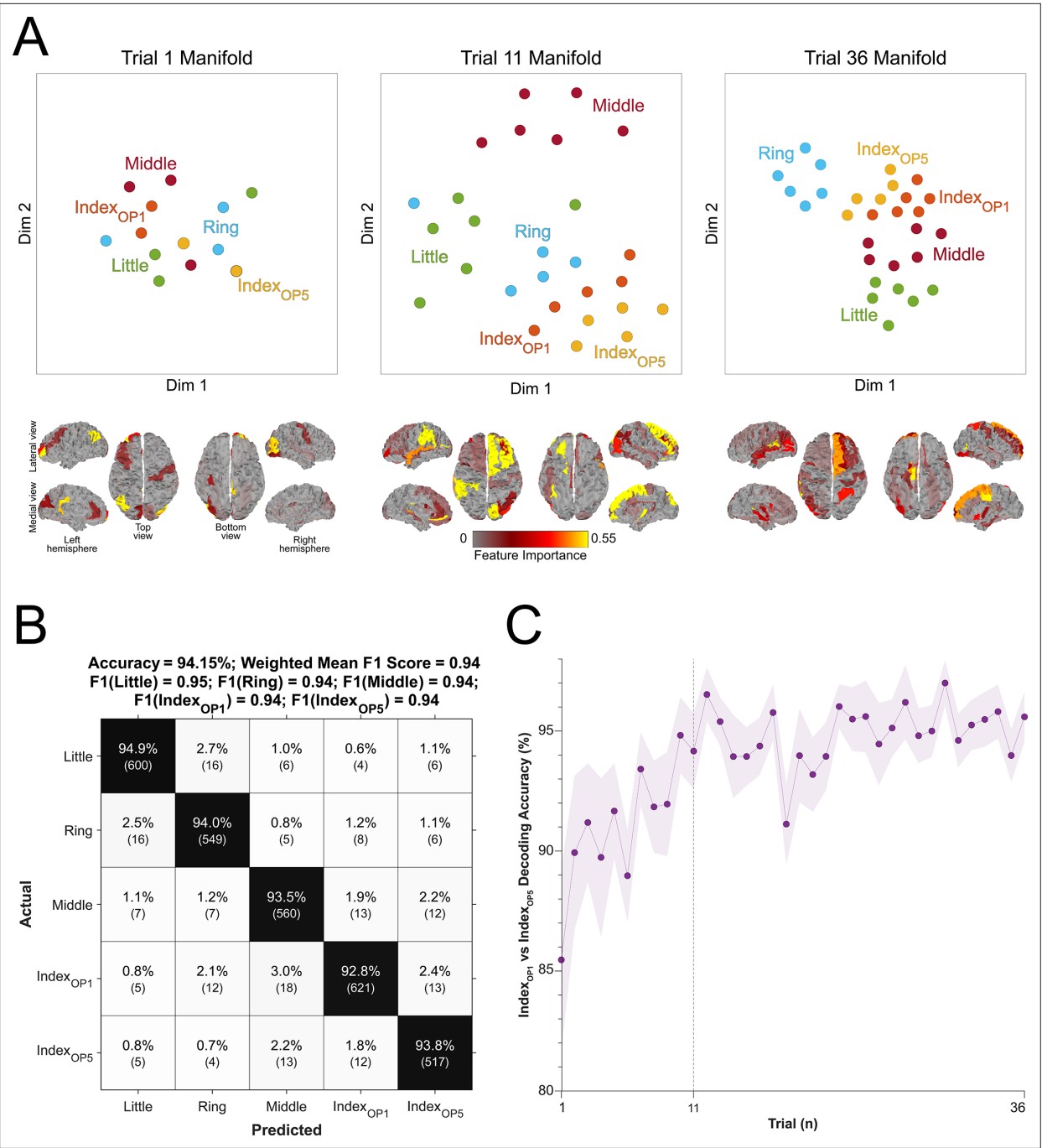

**Figure 4.** Evolution of keypress neural representations with skill learning. (**A**) Keypress neural representations differentiate during early learning. t-SNE distribution of neural representation of each keypress (top scatter plots) is shown for trial 1 (start of training; top-left), 11 (end of early learning; top-center), and 36 (end of training; top-right) for a single representative participant. Individual keypress manifold representation clustering in trial 11 (top-center; end of early learning) depicts sub-clustering for the index finger keypress performed at the two different ordinal positions in the sequence (Index_OP1 and Index_OP5), which remains present by trial 36 (top-right). Spatial distribution of regional contributions to decoding (bottom brain surface maps). The surface color heatmap indicates feature importance scores across the brain. Note that decoding contributions shifted from contralateral right pre-central cortex at trial 1 (bottom-left) to contralateral superior and middle frontal cortex at trials 11 (bottom-center) and 36 (bottom-right). (**B**) Confusion matrix for 5-class decoding of individual sequence items. Decoders were trained to classify contextual representations of the keypresses (i.e. 5-class classification of the sequence elements 4-1-2-3-4). Note that the decoding accuracy increased to 94.15% ± SD 4.84% and the misclassification of keypress 4 was significantly reduced (from 141 to 82). (**C**) Trial-by-trial classification accuracy for 2-class decoder (Index_OP1 vs. Index_OP5). A decoder (200ms window duration aligned to the KeyDown event) was trained to differentiate between the two index finger keypresses embedded at different positions within the practiced skill sequence (Index_OP1=index finger keypress at ordinal position 1 of the sequence; Index_OP5=index finger

*Figure 4 continued on next page*

*Figure 4 continued*

keypress at ordinal position 5 of the sequence). Decoder accuracy progressively improved over early learning, stabilizing around 96% by trial 11 (end of early learning). Similar results were observed for other decoding window sizes (50, 100, 150, 250, and 300ms; see *Figure 4—figure supplement 2*). Taken together, these findings indicate that the neural feature space evolves over early learning to incorporate sequence location information. Shaded region indicates the 95% confidence interval of the group mean.

The online version of this article includes the following figure supplement(s) for figure 4:

**Figure supplement 1.** Quantification of trial-by-trial parcel-space feature importance scores during skill learning.

**Figure supplement 2.** Trial-by-trial classification accuracy for 2-class decoder (Index$_{OP1}$ vs. Index$_{OP5}$).

**Figure supplement 3.** Eye movement features do not contribute to decoding.

*2024*; *Buch et al., 2017*). To this effect, we developed a novel hybrid-space approach designed to integrate neural representation dynamics over two different spatial scales: (1) *whole-brain parcel-space* (i.e. spatial activity patterns across all cortical brain regions) and (2) *regional voxel-space* (i.e. spatial activity patterns within select brain regions) activity. We found consistent spatial differences between whole-brain parcel-space feature importance (predominantly contralateral frontoparietal, *Figure 2B*) and regional voxel-space decoder accuracy (bilateral sensorimotor regions, *Figure 2D*). The *whole-brain parcel-space* decoder likely emphasized more stable activity patterns in contralateral frontoparietal regions that differed between individual finger movements (*Beukema et al., 2019*; *Lemon, 2008*), while the *regional voxel-space* decoder likely incorporated information related to adaptive interhemispheric interactions operating during motor sequence learning (*Buch et al., 2017*; *Zimerman et al., 2014*; *Waters et al., 2017*), particularly pertinent when the skill is performed with the non-dominant hand (*Sawamura et al., 2019*; *Lee et al., 2019*; *Grafton et al., 2002*). The observation of increased cross-validated test accuracy (as shown in *Figure 3—figure supplement 6*) indicates that the spatially overlapping information in parcel- and voxel-space time-series in the hybrid decoder was complementary, rather than redundant (*Yu and Liu, 2004*). The hybrid-space decoder, which achieved an accuracy exceeding 90%—and robustly generalized to Day 2 across trained and untrained sequences—surpassed the performance of both parcel-space and voxel-space decoders and compared favorably to other neuroimaging-based finger movement decoding strategies (*Buch et al., 2021*; *Lee et al., 2022*; *Liao et al., 2014*; *Quandt et al., 2012*; *Kornysheva et al., 2019*).

Evaluation of individual brain oscillatory activity revealed that low-frequency oscillations (LFOs) result in higher decoding accuracy compared to other narrow-band activity (*Natraj et al., 2022*; *Reddy et al., 2021*). Task-related movements—which also express in lower frequency ranges—did not explain these results given the near chance-level performance of alternative decoders trained on (a) artifact-related ICA components removed during MEG pre-processing (*Figure 3—figure supplement 3A–C*) and on (b) task-related eye movement features (*Figure 4—figure supplement 3B, C*). This explanation is also inconsistent with the minimal average head motion of 1.159 mm (±1.077 SD) across the MEG recording (*Figure 3—figure supplement 3D*). How could LFOs contribute to keypress decoding accuracy? LFOs, observed during movement onset in the cerebral cortex of animals (*Bansal et al., 2011*; *Mollazadeh et al., 2011*) and humans (*Bönstrup et al., 2019b*; *Cruikshank et al., 2012*; *Tomassini et al., 2017*), encode information about movement trajectories and velocity (*Bansal et al., 2011*; *Mollazadeh et al., 2011*). They also contain information related to movement timing (*Ramanathan et al., 2018*; *Hall et al., 2014*; *Stefanics et al., 2010*), preparation (*Flint et al., 2012*; *Krasoulis et al., 2014*), sensorimotor integration (*Cruikshank et al., 2012*), kinematics (*Flint et al., 2012*; *Krasoulis et al., 2014*) and may contribute to the precise temporal coordination of movements required for sequencing (*Churchland et al., 2012*). Within clinical contexts, LFOs in the frontoparietal regions, resulting in high decoding accuracy in the present study, have been linked to recovery of motor function after brain lesions like stroke (*Bönstrup et al., 2019b*; *Ramanathan et al., 2018*; *Frohlich et al., 2021*).

## Neural representations of individual sequence actions differentiate during early skill learning

Next, we exploited the hybrid decoding approach to investigate if individual sequence action representations differentiate or remain stable during early skill learning, when the memory is not yet fully formed (*Bönstrup et al., 2019a*). The first hint of representational differentiation was the highest

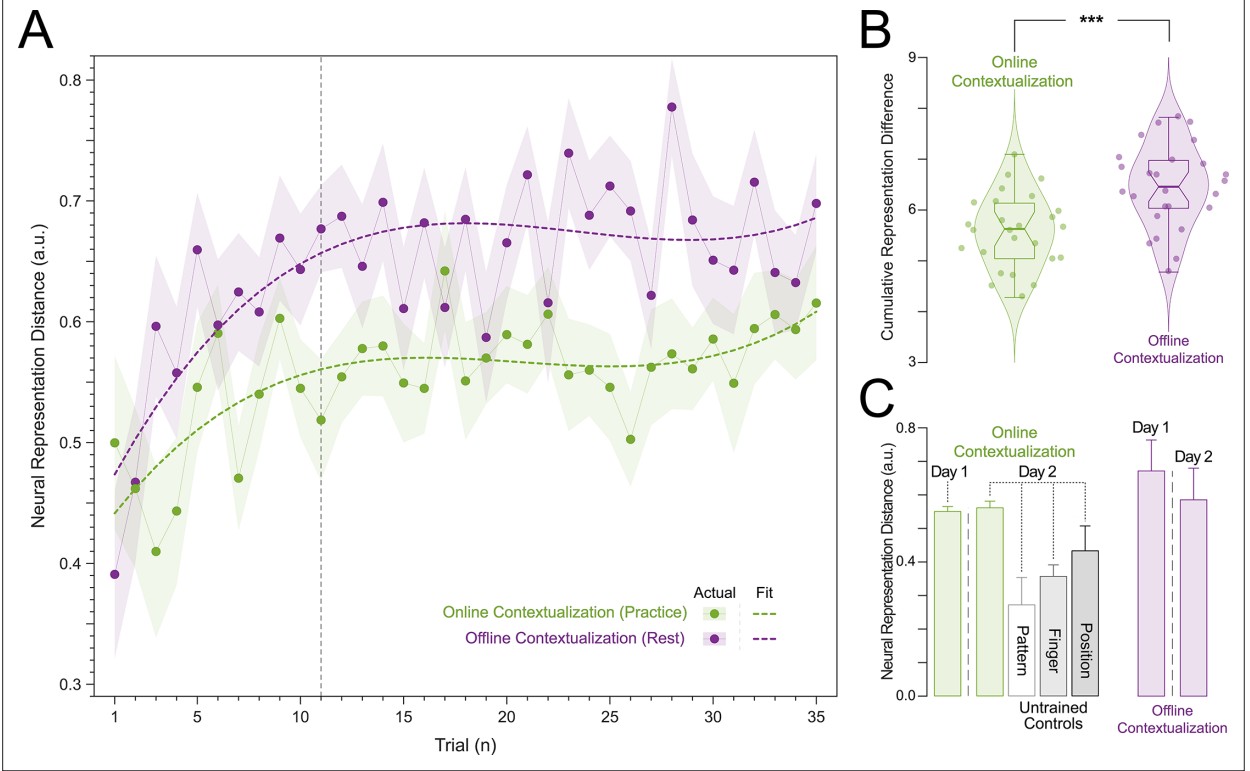

**Figure 5.** Neural representation distance between index finger keypresses performed at two different ordinal positions within a sequence. (**A**) Contextualization increases over Early Learning during Day 1 Training. Online (green) and offline (purple) neural representation distances (contextualization) between two index finger key presses performed at ordinal positions 1 and 5 of the trained sequence (4-1-3-2-4) are shown for each trial during Day 1 Training. Both online and offline contextualization between the two index finger representations increases sharply over Early Learning before stabilizing across later Day 1 Training trials. Shaded regions indicate the 95% confidence interval of the group mean. (**B**) Contextualization develops predominantly during rest periods (offline) on Day 1. The cumulative neural representation differences during early learning were significantly greater over rest (Offline contextualization; right) than during practice (Online contextualization; left) periods (t=4.84, p<0.001, df = 25, Cohen's d=1.2). Center line of box plot indicates the group median, while notches represent the 95% confidence interval of the group median. Box limits indicate the 1st and 3rd quartiles while whisker lengths are set at the extreme value ≤1.5×IQR. (**C**) Contextualization acquired on Day 1 was retained on Day 2 specifically for the trained sequence. The neural representation differences assessed across both rest and practice for the trained sequence (4-1-3-2-4) were retained at Day 2 Retest. This is in stark contrast with the reduction in contextualization for several untrained sequences controlling for: (1) index finger keypresses located at the same ordinal positions 1 and 5 but within a different intervening sequence pattern (Pattern Specificity Control: 4-2-3-1-4, 51.05% lower contextualization); (2) use of a finger different than the index (little or ring finger) in both ordinal positions 1 and 5 (Finger Specificity Control: 2-1-3-4-2, 1-4-2-3-1 and 2-3-1-4-2; 35.80% lower contextualization); and (3) multiple index finger keypresses occurring at ordinal positions other than 1 and 5 (Position Specificity Control: 4-2-4-3-1 and 1-4-3-4-2; 22.06% lower contextualization). Note that offline contextualization cannot be measured for the Day 2 Control sequences as each sequence was only performed over a single trial. Error bars indicate S.E.M.

The online version of this article includes the following figure supplement(s) for figure 5:

**Figure supplement 1.** Relationship between offline neural representational changes and micro-offline learning.

**Figure supplement 2.** Trial-by-trial trends for different measurement approaches of offline and online contextualization changes.

**Figure supplement 3.** Online contextualization versus micro-online learning.

**Figure supplement 4.** Within-subject correlations between online and offline contextualization changes versus learning.

**Figure supplement 5.** Online versus offline changes in keypress transition patterns.

**Figure supplement 6.** The relationship between adjacent index finger transitions and online contextualization.

**Figure supplement 7.** Between-subject differences in typing speed versus online contextualization.

false-negative and lowest false-positive misclassification rates for index finger keypresses performed at different locations in the sequence compared with all other digits (*Figure 3C*). This was further supported by the progressive differentiation of neural representations of the index finger keypress (*Figure 4A*) and by the robust trial-by-trial increase in 2-class (Index$_{OP1}$ vs Index$_{OP5}$) decoding accuracy across time windows ranging between 50 and 250ms (*Figure 4C*; *Figure 4—figure supplement 2*).

Further, the 5-class classifier—which directly incorporated information about the sequence location context of each keypress into the decoding pipeline—improved decoding accuracy relative to the 4-class classifier (*Figure 4C*). Importantly, testing on Day 2 revealed specificity of this representational differentiation for the trained skill but not for the same keypresses performed during various unpracticed control sequences (*Figure 5C*).

The main region contributing information to representational differentiation during early practice (trials 1–10) was the primary motor cortex, followed by the somatosensory cortex (trial 11), both of which are known to be actively engaged in skill acquisition (*Buch et al., 2021*; *Karni et al., 1995*; *Classen et al., 1998*; *Kleim et al., 1998*; *Kumar et al., 2019*; *Pavlides et al., 1993*). Concurrently, information from the superior frontal and middle frontal cortex—which encodes hierarchical structures of skill sequences (*Yokoi and Diedrichsen, 2019*)—steadily increased in importance and emerged as the two most crucial decoding contributors once skill performance plateau had been reached (trials 15–36; *Figure 4—figure supplement 1*; *Hikosaka et al., 1999*; *Dayan and Cohen, 2011*). Thus, the neural substrates supporting finger movements and their representational differentiation during early skill learning (the time period during which 95% skill gains in the training session occur *Bönstrup et al., 2019a*; *Pan and Rickard, 2015*, trials 1–11 in this study) differed from those supporting stable performance during the subsequent skill plateau period (*Karni et al., 1995*; *Robertson and Cohen, 2006*; trials 12–36 in this study).

## Differentiation of neural representations developed predominantly during rest periods interspersed with practice

We then focused on the timeline of differentiation of index finger keypress neural representations—which we refer to as contextualization—over early learning. We found that contextualization increased progressively during early learning—predominantly during short rest breaks (offline) rather than during practice (online; *Figure 5*, *Figure 5—figure supplement 2B*). Offline contextualization consistently correlated with early learning gains across a range of decoding windows (50–250ms; *Figure 5—figure supplement 1*). This result remained unchanged when measuring offline contextualization between the last and second sequence of consecutive trials, inconsistent with a possible confounding effect of pre-planning (*Ariani and Diedrichsen, 2019*; *Figure 5—figure supplement 2A*). On the other hand, online contextualization did not predict learning (*Figure 5—figure supplement 3*). Consistent with these results, the average within-subject correlation between offline contextualization and micro-offline gains was significantly stronger than within-subject correlations between online contextualization and either micro-online or micro-offline gains (*Figure 5—figure supplement 4*).

Offline contextualization was not driven by trial-by-trial behavioral differences, including typing rhythm (*Figure 5—figure supplement 5*) and adjacent keypress transition times (*Figure 5—figure supplement 6*) nor by between-subject differences in overall typing speed (*Figure 5—figure supplement 7*)—ruling out a reliance on differences in the temporal overlap of keypresses. Importantly, offline contextualization documented on Day 1 stabilized once a performance plateau was reached (trials 11–36) and was retained on Day 2, documenting overnight consolidation of the differentiated neural representations. A possible neural mechanism supporting contextualization could be the emergence and stabilization of conjunctive 'what–where' representations of procedural memories (*Komorowski et al., 2009*) with the corresponding modulation of neuronal population dynamics (*Georgopoulos, 1994*; *Georgopoulos et al., 1982*) during early learning. Exploring the link between contextualization and neural replay could provide additional insights into this issue (*Buch et al., 2021*; *Chen et al., 2024*; *Sjøgård, 2024*; *Griffin et al., 2025*).

In this study, classifiers were trained on MEG activity recorded during or immediately after each keypress, emphasizing neural representations related to action execution, memory consolidation, and recall over those related to planning. An important direction for future research is determining whether separate decoders can be developed to distinguish the representations or networks separately supporting these processes. Ongoing work in our lab is addressing this question. The present accuracy results across varied decoding window durations and alignment with each keypress action support the feasibility of this approach (*Figure 3—figure supplement 5*).

## Limitations

One limitation of this study is that contextualization was investigated for only one finger movement (index finger or digit 4) embedded within a relatively short 5-item skill sequence. Determining if representational contextualization is exhibited across multiple finger movements embedded within, for example, longer sequences (e.g. two index finger and two little finger keypresses performed within a short piece of piano music) will be an important extension to the present results. While a supervised manifold learning approach (LDA) was used here because it optimized hybrid-space decoder performance, unsupervised strategies (e.g. PCA and MDS, which also substantially improved decoding accuracy in the present study; *Figure 3—figure supplement 2*), are likely more suitable for real-time BCI applications. Finally, caution should be exercised when extrapolating findings during early skill learning, a period of steep performance improvements, to findings reported after insufficient practice (*Das et al., 2024*), post-plateau performance periods (*Gupta and Rickard, 2022*), or non-learning situations (e.g. performance of non-repeating keypress sequences in *Das et al., 2024*) when reactive inhibition or contextual interference effects are prominent. Ultimately, it will be important to develop new paradigms allowing one to independently estimate the different coincident or antagonistic features (e.g. memory consolidation, planning, working memory, and reactive inhibition) contributing to micro-online and micro-offline gains during and after early skill learning within a unifying framework.

## Summary

In summary, individual sequence action representations contextualize during early learning of a new skill, and the degree of differentiation parallels skill gains. Differentiation of the neural representations developed during rest intervals of early learning to a larger extent than during practice in parallel with rapid consolidation of skill. It is possible that the systematic inclusion of contextualized information into sequence skill practice environments could improve learning in areas as diverse as music education, sports training, and rehabilitation of motor skills after brain lesions.

# Materials and methods

## Study participants

The study was approved by the Combined Neuroscience Institutional Review Board of the National Institutes of Health (NIH). A total of thirty-three young and healthy adults (16 females) with a mean age of 26.6 years (±0.87 SEM) participated in the study after providing written informed consent and undergoing a standard neurological examination. No participants were actively engaged in playing musical instruments in their daily lives, as per guidelines outlined in prior research (*Ruiz et al., 2009*; *Maidhof et al., 2009*). All study scientific data were de-identified and permanently unlinked from all personal identifiable information (PII) before the analysis. These data are publicly available (https://doi.org/10.18112/openneuro.ds006502.v1.0.0; Accession Number: ds006502). Two participants were excluded from the analysis due to MEG system malfunction during data acquisition. An additional 5 subjects were excluded because they failed to generate any correct sequences in two or more consecutive trials. The study was powered to determine the minimum sample size needed to detect a significant change in skill performance following training using a one-sample t-test (two-sided; alpha = 0.05; 95% statistical power; Cohen's *d* effect size = 0.8115 calculated from previously acquired data in our lab *Censor et al., 2014*). The calculated minimum sample size was 22. The included study sample size (n=26) exceeded this minimum (*Bönstrup et al., 2019a*).

## Experimental setup

Participants practiced a procedural motor skill learning task that involved repetitively typing a 5-item numerical sequence (4-1-3-2-4) displayed on a computer screen. They were instructed to perform the task as quickly and accurately as possible on a response pad (Cedrus LS-LINE, Cedrus Corp) using their non-dominant, left hand. Each numbered sequence item corresponded to a specific finger keypress: 1 for the little finger, 2 for the ring finger, 3 for the middle finger, and 4 for the index finger. Individual keypress times and identities were recorded and used to assess skill learning and performance. The response pad was positioned in a manner that minimized wrist, arm, or more proximal body movements during the task. The head was restrained with an inflatable air bladder, and head

position was assessed at the beginning and at the end of each recording. The mean measured head movement across the study group was 1.159 mm (±1.077 SD; *Figure 3—figure supplement 3*).

Participants practiced the skill for 36 trials. Each trial spanned a total of 20 s and included a 10 s practice round followed by a 10 s rest break. The study design followed specific recommendations by *Pan and Rickard, 2015*: (1) utilizing 10 s practice trials and (2) constraining analysis of micro-offline gains to early learning trials (where performance monotonically increases and 95% of overall performance gains occur) that precede the emergence of 'scalloped' performance dynamics strongly linked to reactive inhibition effects (*Pan and Rickard, 2015*; *Brawn et al., 2010*). This is precisely the portion of the learning curve Pan and Rickard referred to when they stated "…*rapid learning during that period masks any reactive inhibition effect*" (*Pan and Rickard, 2015*).

The five-item sequence was displayed on the computer screen for the duration of each practice round, and participants were directed to fix their gaze on the sequence. Small asterisks were displayed above a sequence item after each successive keypress, signaling the participants' present position within the sequence. Inclusion of this feedback minimizes working memory loads during task performance (*Walker et al., 2002*). Following the completion of a full sequence iteration, the asterisk returned to the first sequence item. The asterisk did not provide error feedback as it appeared for both correct and incorrect keypresses. At the end of each practice round, the displayed number sequence was replaced by a string of five 'X' symbols displayed on the computer screen, which remained for the duration of the rest break. Participants were instructed to focus their gaze on the screen during this time. The behavior in this explicit, motor learning task consists of generative action sequences rather than sequences of stimulus-induced responses as in the serial reaction time task (SRTT). A similar real-world example would be manually inputting a long password into a secure online application in which one intrinsically generates the sequence from memory and receives similar feedback about the password sequence position (also provided as asterisks), which is typically ignored by the user.

On the next day, participants were tested (*Day 2 Retest*) with the same trained sequence (4-1-3-2-4) for nine trials as well as for nine different unpracticed control sequences (*Day 2 Control*; 2-1-3-4-2; 4-2-4-3-1; 3-4-2-3-1; 1-4-3-4-2; 3-2-4-3-1; 1-4-2-3-1; 3-2-4-2-1; 2-3-1-4-2; 4-2-3-1-4) each for one trial. The practice schedule structure for Day 2 was the same as Day 1, with 10 s practice trials interleaved with 10 s rest breaks.

## Behavioral data analysis

### Skill

Skill, in the context of the present task, is quantified as the *correct sequence typing speed*, (i.e. the number of correctly typed sequence keypresses per second; kp/s). That is, improvements in the speed/accuracy trade-off equate to greater skill. Keypress transition times (KTT) were calculated as the difference in time between the *keyDown* events recorded for consecutive keypresses. Since the sequence was repeatedly typed within a single trial, individual keypresses were marked as correct if they were members of a five consecutive keypress set that matched any possible circular shift of the displayed five-item sequence. The instantaneous correct sequence speed was calculated as the inverse of the average KTT across a single correct sequence iteration and was updated for each correct keypress. Trial-by-trial skill changes were assessed by computing the median correct sequence typing speed for each trial.

### Early learning

The *early learning* period was defined as the trial range (1 T trials) over which 95% of the total skill performance was first attained at the group level. We quantified this by fitting the group average trial-by-trial correct sequence speed data with an exponential model of the form:

$$L\left(t\right) = C1 + C2\left(1 - e^{-kt}\right)$$

Here, the trial number is denoted by *t*, and *L(t)* signifies the group-averaged performance at trial *t*. Parameters *C1* and *C2* correspond to the pre-training performance baseline and asymptote, respectively, while *k* denotes the learning rate. The values for *C1*, *C2*, and *k* were computed using a constrained nonlinear least-squares method (MATLAB's lsqcurvefit function, trust-region-reflective algorithm) and were determined to be 0.5, 0.15, and 0.2, respectively. The early learning trial cut-off,

denoted as $T$, was identified as the first trial where 95% of the learning had been achieved. In this study, $T$ was determined to be trial 11.

## Micro-offline and -online gains

Performance improvements over each 10 s rest break (*micro-offline gains)* were calculated as the net performance change (instantaneous correct sequence typing speed) from the end of one practice period to the onset of the next, while *micro-online gains* were computed as the net performance change over a single practice trial. Total early learning was derived as the sum of all *micro-online* and *micro-offline* gains over trials 1–11. Cumulative micro-offline gains, micro-online gains, and total early learning were statistically compared using one-way ANOVAs and post-hoc Tukey tests. Possible pre-planning effects on initial skill performance (*Ariani and Diedrichsen, 2019*) were assessed by using paired t-tests to statistically compare cumulative micro-offline and -online computed for all keypresses with their measurement counterparts calculated after omitting the first 3 keypresses in each trial from the correct sequence speed computation.

## MRI acquisition

We acquired T1-weighted high-resolution anatomical MRI volumes images (1 mm$^3$ isotropic MPRAGE sequence) for each participant on a 3T MRI scanner (GE Excite HDxt or Siemens Skyra) equipped with a 32-channel head coil. These data allowed for spatial co-registration of an individual participant's brain with the MEG sensors, and individual head models required for surface-based cortical dipole estimation from MEG signals (i.e. MEG source-space modeling).

## MEG acquisition

We recorded continuous magnetoencephalography (MEG) at a sampling frequency of 600 Hz using a CTF 275 MEG system (CTF Systems, Inc, Canada) while participants were seated in an upright position. The MEG system comprises a whole-head array featuring 275 radial 1$^{st}$-order gradiometer/SQUID channels housed in a magnetically shielded room (Vacuumschmelze, Germany). Three of the gradiometers (two non-functional and one with high channel noise after visual inspection) were excluded from the analysis resulting in a total of 272 useable MEG sensor channels. Synthetic third-order gradient balancing was applied to eliminate background noise in real-time data collection. Temporal alignment of behavioral and MEG data was achieved using a TTL trigger. Head position in the scanner coordinate space was assessed at the beginning and end of each recording using head localization coils at the nasion, left, and right pre-auricular locations. These fiducial positions were co-registered in the participants' T1-MRI coordinate space using a stereotactic neuronavigation system (BrainSight, Rogue Research Inc). MEG data was acquired starting 6 min before the task (resting-state baseline) and continued through the end of the 12 min training session.

## MEG data analysis

### Preprocessing

MEG data were preprocessed using the FieldTrip (*Oostenveld et al., 2011*) and EEGLAB (*Delorme and Makeig, 2004*) toolboxes on MATLAB 2022a. Continuous raw MEG data were band-pass filtered between 1–100 Hz with a fourth-order noncausal Butterworth filter. 60 Hz line noise was removed with a narrow-band discrete Fourier transform (DFT) notch filter. Independent component analysis (ICA) was used to remove typical MEG signal artifacts associated with eye blinks or movement, muscle contraction or cardiac pulsation. All recordings were visually inspected and marked to denoise segments containing other large amplitude artifacts due to movements. Eye movements were simultaneously recorded with MEG (EyeLink 1000 Plus).

### Source reconstruction and parcellation

For each participant, individual volume conduction models were computed to estimate the propagation of brain-generated currents through tissue resulting in externally measurable magnetic fields. This was accomplished through a single-shell head corrected-sphere approach based on the brain volume segmentation of the participant's high-resolution T1 MRI. Source models and surface labels from the Desikan-Killiany Atlas (*Destrieux et al., 2010*) were created for each participant using inner-skull and pial layer surfaces obtained through FreeSurfer segmentation (*Dale et al., 1999*; *Marcus et al., 2011*)

and Connectome Workbench resampling (*Dale et al., 1999*; *Marcus et al., 2011*). Aligning sensor positions in the MEG helmet to individual head space involved rigid-body registration of the mean MEG head coil position to the same fiducial locations marked in the MRI and applying the same affine transformation to all MEG sensors.

The individual source, volume conduction model, and sensor positions were then utilized to generate the forward solution at each source dipole location, describing the propagation of source activity from each cortical location on the grid to each MEG sensor. The Linearly Constrained Minimum-Variance (LCMV) beamformer was employed for computing the inverse solution. Each trial of MEG activity contributed to calculating the inverse solution data covariance matrix. The individual sample noise covariance matrix was derived from 6 min of pre-training rest MEG data recorded in the same subject during the same session. A total of 15,684 surface-based cortical dipoles (i.e. source-space voxels) were estimated.

Source-space parcellation was carried out by averaging all voxel time-series located within distinct anatomical regions defined in the Desikan-Killiany Atlas (*Destrieux et al., 2010*). Since source time-series estimated with beamforming approaches are inherently sign-ambiguous, a custom Matlab-based implementation of the *mne.extract_label_time_course* with '*mean_flip*' sign-flipping procedure in MNE-Python (*Gramfort et al., 2013*) was applied prior to averaging to prevent within-parcel signal cancellation. All voxel time-series within each parcel were extracted, and the time series sign was flipped at locations where the orientation difference was greater than 90° from the parcel mode. A mean time series was then computed across all voxels within the parcel after sign-flipping.

## Feature selection for decoding

Several MEG activity features were extracted over different spatial, spectral, and temporal scales.

### Oscillatory analysis

MEG signals were constrained to broadband (1–100 Hz) or standard neural oscillatory frequency bands defined as delta (1–3 Hz), theta (4–7 Hz), alpha (8–14 Hz), beta (15–24 Hz), gamma (25–50 Hz), and high-gamma (51–100 Hz) using a fourth-order non-causal Butterworth filter. Subsequent decoding analyses were independently conducted for each band of MEG activity.

### Spatial analysis

Decoding was performed in both sensor and source spaces. The sensor-space decoding feature dimension was 272 (corresponding to the 272 usable MEG channels), while source-space decoding was carried out at both the higher feature dimension voxel (i.e. higher spatially sampled; N=15,684) and lower feature dimension parcel space (i.e. lower spatially sampled; N=148) across all oscillatory frequency bands (i.e. broadband, delta, theta, alpha, beta, gamma, and high-gamma) for comprehensive comparison.

### Temporal analysis

MEG activity time series corresponding to each keypress was defined using the time window, [t + △t], where t ∈ [0 : 10ms: 100ms] and △t ∈ [25ms: 25ms: 350ms]. In other words, a sliding window of variable width (from 25 ms to 350 ms with 25ms increments), and with onsets ranging from the *keyDown* event (i.e. t=0ms) to +100ms after the *keyDown* event (with increments of 10ms) was used. This approach generated a set of 140 different time windows associated with each keypress for each participant. MEG activity was averaged over time within each of these windows and independently analyzed for decoding. The optimal time window was selected for each subject that resulted in the maximum cross-validation performance (*Figure 3—figure supplement 5*). This window optimization analysis was performed for each frequency band and spatial scale.

### Hybrid spatial approach

First, we evaluated the decoding performance of each individual brain region in accurately decoding finger keypresses from regional voxel-space (i.e. all voxels within a brain region as defined by the Desikan-Killiany Atlas) activity. Brain regions were then ranked from 1 to 148 based on their decoding accuracy at the group level. In a stepwise manner, we then constructed a 'hybrid-space' decoder

by incrementally concatenating regional voxel-space activity of brain regions—starting with the top-ranked region—with whole-brain parcel-level features and assessed decoding accuracy. Subsequently, we added the regional voxel-space features of the second-ranked brain region and continued this process until decoding accuracy reached saturation. The optimal 'hybrid-space' input feature set over the group included the 148 parcel-space features and regional voxel-space features from a total of 8 brain regions (bilateral superior frontal, middle frontal, pre-central, and post-central; N=1295 ± 20 features).

## Dimension reduction

We independently applied several supervised and unsupervised dimension reduction techniques as an additional feature extraction step for each broadband MEG activity space (i.e. sensor, parcel, voxel, and hybrid), including: linear discriminant analysis (LDA), minimum redundant maximum relevance (MRMR), principal component analysis (PCA), Autoencoder, Diffusion maps, factor analysis, large margin nearest neighbor (LMNN), multi-dimensional scaling (MDS), neighbor component analysis (NCA), spatial predictor envelope (SPE; *Maaten and Postma, 2009*). Among these techniques, PCA, MDS, MRMR, and LDA emerged as particularly effective in significantly improving decoding performance across all broadband MEG activity spaces.

PCA, a method for unsupervised dimensionality reduction, transforms the high-dimensional dataset into a new coordinate system of orthogonal principal components. These components, capturing the maximum variance in the data, were iteratively added to reconstruct the feature space and execution of decoding. MDS finds a configuration of points in a lower-dimensional space such that the distances between these points reflect the dissimilarities or similarities between the corresponding objects in the original high-dimensional space. MRMR, an approach combining relevance and redundancy metrics, ranks features based on their significance to the target variable and their non-redundancy with other features. The decoding process started with the highest-ranked feature and iteratively incorporated subsequent features until decoding accuracy reached saturation. LDA finds the linear combinations of features (dimensions) that best separate different classes in a dataset. It projects the original features onto a lower-dimensional space (number of classes −1) while preserving the class-discriminatory information. This transformation maximizes the ratio of the between-class variance to the within-class variance. In our study, LDA transformed the features to a 3/4-dimensional hyperdimensional space that was used for decoding. Dimension reduction was first applied to training data, and then with the tuned parameters of the dimension reduction model, an independent test data subset was transformed for decoder metrics evaluation. Decoding accuracies were systematically compared between the original and reduced dimension feature spaces, providing insight into the effectiveness of each dimension reduction technique. By rigorously assessing the impact of dimension reduction on decoding accuracy, the study aimed to identify techniques that not only reduced the computational burden associated with high-dimensional data but also enhanced the discriminative power of the selected features. This comprehensive approach aimed at optimizing the neural representation of each finger keypress for decoding performance across various spatial contexts.

## Decoding analysis

Decoding analysis was conducted for each participant individually, employing a randomized split of the data into independent training (90%) and test (10%) samples over eight iterations. For each iteration, an eightfold cross-validation was applied to the training samples to optimize decoder configuration, allowing for the fine-tuning of hyperparameters and selection of the most effective model. On average, the total number of individual keypress samples for the entire duration of training was 219 ± SD: 66 (keypress 1: little), 205 ± SD: 66 (keypress 2: ring), 209±66 (keypress 3: middle), and 426 ± SD: 131 (keypress 4: index) across participants. Only keypresses belonging to correctly typed sequence iterations (94.64% ± 4.04% of all keypresses) were considered. The total number of index finger keypresses (i.e. keypress 4) was approximately twice that of the others, as it was the only action that occurred more than once in the trained sequence (4-1-3-2-4), albeit at two different ordinal positions. Considering the higher (2 x) number of samples for one-class, we independently oversampled the keypresses 1, 2, and 3 to avoid overfitting to the over-represented class. Importantly, oversampling was applied independently for each keypress class, ensuring that validation folds were never oversampled, and training folds did not share common oversampled patterns. The decoder configuration

demonstrating the best validation performance was selected for each iteration, and subsequently, test accuracy was evaluated on the independent/unseen test samples. This process was repeated for the eight different iterations of train-test splitting, and the average test accuracy over all iterations was reported. This rigorous methodology aimed at generalizing decoding performance to ensure robust and reliable results across participants. Finally, decoding evaluation was also performed on the Day 2 data, for both the trained (*Day 2 Retest*; 9 trials) and untrained sequences (*Day 2 Control*; 9 different single-trial tests).

## Machine learning classifiers

We employed a diverse set of machine learning-based decoders—including Naïve Bayes (NB), decision trees (DT), ensembles (EN), k-nearest neighbor (KNN), linear discriminant analysis (LDA), support vector machines (SVM), and artificial neural network (ANN)—to train features generated over all possible combinations of spatial and temporal scales and oscillation frequency-bands in order to carry out a comprehensive comparative analysis. The hyperparameters of these decoders underwent fine-tuning using Bayesian optimization search.

All NB classifiers were configured with a normal distribution predictor and Gaussian Kernel, while KNN classifiers had a K value of 4 (for keypress decoding) and utilized the Euclidean distance metric. For DT classifiers, the maximum number of splits was set to 4 (for keypress decoding), with leaves being merged based on the sum of risk values greater than or equal to the risk associated with the parent node. The optimal sequence of pruned trees was estimated, and the predictor selection method was 'Standard CART', selecting the split predictor that maximizes the split-criterion gain over all possible splits of all predictors. The split criterion used was 'gdi' (Gini's diversity index). EN classifiers employed the bagging method with random predictor selections at each split, forming a random forest. The maximum number of learning cycles was set to 100 with a weak learner based on discriminant analysis. For SVM, the RBF kernel was selected through cross-validation (CV), and the 'C' parameter and kernel scale were optimized using Bayesian optimization search. In the case of LDA, the linear coefficient threshold and the amount of regularization were computed based on Bayesian optimization search. Finally, all ANN decoders consisted of one hidden layer with 128 nodes, followed by a sigmoid and a softmax layer, each with four nodes (for keypress decoding). Training utilized a scaled conjugate gradient optimizer with backpropagation, employing a learning rate of 0.01 (coarse to fine tuning) for a maximum of 100 epochs, with early stopping validation patience set to 6 epochs.

## Decoding performance metric

Decoding performance was assessed using several metrics, including accuracy (%), which indicates the proportion of correct classifications among all test samples. Confusion matrices provide a detailed comparison of the number of correctly predicted samples for each class against the ground truth. The F1 score—defined as the harmonic mean of the precision (percentage of true predictions that are actually true positive) and recall (percentage of true positives that were correctly predicted as true) scores—was used as a comprehensive metric for each one-versus-all keypress state decoder to assess class-wise performance that accounts for both false-positive and false-negative prediction tendencies (*Rijsbergen, 1979*; *Schütze et al., 2008*). A weighted mean F1 score was then computed across all classes to assess the overall prediction performance of the multi-class model. Test accuracies based on the best decoder performance (LDA in our case) were reported and used for statistical comparisons.

## Decoding during skill learning progression

We systematically assessed decoding performance of a 2-class decoder (Index$_{OP1}$ vs Index$_{OP5}$; i.e. decoding of two index finger keypresses occurring at different locations within the training sequence) at each trial during the skill learning process to capture the evolving relationship between differentiated index finger decoding proficiency and the acquired skill. Our approach involved evaluating decoder performance individually for each *Day 1 Training* trial. We ensured an equal number of samples (first *k* keypresses) in each trial were used to mitigate the influence of increasing samples available in later trials.

We used t-distributed stochastic neighborhood estimation (t-SNE) to visualize the evolution of neural representations corresponding to each keypress at each trial of the learning period. Within

t-SNE distributions, index finger keypresses were separately labeled based upon their sequence location (i.e. $Index_{OP1}$ and $Index_{OP5}$ for ordinal positions 1 and 5, respectively).

### Decoding sequence elements

We performed 5-class decoding of each action based on its location within the sequence (i.e. $Index_{OP1}$, Little, Middle, Ring, and $Index_{OP5}$). The same decoding strategy was utilized as for the above 4-class keypress-based decoding (i.e. 90%–10% split for train and test, 8-fold cross-validation of training samples to select best decoder configuration, hybrid spatial features, and LDA-based dimension reduction). Note, oversampling was not needed after sub-grouping the index finger keypresses into two separate classes based on their sequence context. 5-class sequence-based decoding was evaluated for both *Day 1 Training*, *Day 2 Retest,* and *Day 2 Control* data.

### Feature importance scores

The relative contribution of source-space voxels and parcels to decoding performance (i.e. feature importance score) was calculated using minimum redundant maximum relevance (MRMR; *Ding and Peng, 2005*) and highlighted in topography plots. MRMR, an approach that combines both relevance and redundancy metrics, ranked individual features based upon their significance to the target variable (i.e. keypress state identity) prediction accuracy and their non-redundancy with other features.

## Neural representation analysis

We evaluated the *online* (within-trial) and *offline* (between-trial) changes in the neural representation of the contextual actions ($Index_{OP1}$ and $Index_{OP5}$) for each trial during training. For offline differentiation, we evaluated the Euclidean distance between the hybrid spatial features of the last index finger keypress of a trial ($Index_{OP5}$) to the first index finger keypress ($Index_{OP1}$) of the subsequent trial, mirroring the approach used to calculate micro-offline gains in skill. This offline distance provided insight into the net change in contextual representation of the index finger keypress over each interleaved rest break. For online differentiation, we calculated either the mean Euclidean distance between $Index_{OP1}$ and $Index_{OP5}$ of all the correctly typed sequences (*sequence-based*) or the distance between the first $Index_{OP1}$ and last $Index_{OP5}$ (*trial-based*) within the same practice trial. Online differentiation informed on the net change in the contextual representation of the index finger keypress occurring within each practice trial. Cumulative offline and online representation distances across participants were statistically compared using paired *t*-tests. As a control analysis, we computed the difference in neural representation between $Index_{OP1}$ and $Index_{OP5}$ on *Day 2 Retest* data for the same sequence (**4**-1-3-2-**4**) as well as for different *Day 2 Control* untrained sequences where the same action was performed at ordinal positions 1 and 5 (**2**-1-3-4-**2; 1**-4-2-3-**1; 2**-3-1-4-**2; 4**-2-3-1-**4**). We also assessed for specificity of contextualization to the trained sequence, by evaluating differentiation between index finger keypress representations performed at two different positions within untrained sequences (**4**-2-**4**-3-1 and 1-**4**-3-**4**-2). The cumulative differences were compared across participants with paired *t*-tests.

Finally, we computed trial-by-trial differences in offline and online representations during early learning, exploring their temporal relationships with cumulative micro-offline and -online gains in skill, respectively, through regression analysis and Pearson correlation analysis. Linear regression models were trained utilizing the *fitlm* function in MATLAB. The model employed M-estimation, formulating estimating equations and solving them through the Iteratively Reweighted Least Squares (IRLS) method (*Holland and Welsch, 1977*). Key metrics such as the square root of the mean squared error (RMSE), which estimates the standard deviation of the prediction error distribution, the coefficient of explained variance ($R^2$), the F-statistic as a test statistic for the F-test on the regression model, examining whether the model significantly outperforms a degenerate model consisting only of a constant term, and the p-value for the F-test on the model were computed and compared across different models. This multifaceted approach aimed to uncover the nuanced dynamics of neural representation changes in response to skill acquisition.

## Acknowledgements

We thank Ms. Tasneem Malik, Ms. Michele Richman, NIMH MEG Core Facility staff, and NIH NMRF and FMRIF Core Facility staff for their support. This work utilized the computational resources of the

NIH HPC Biowulf cluster (http://hpc.nih.gov). This research was supported by the Intramural Research Program of the National Institutes of Health (NIH). The contributions of the NIH author(s) were made as part of their official duties as NIH federal employees, are in compliance with agency policy requirements, and are considered works of the United States Government. However, the findings and conclusions presented in this paper are those of the author(s) and do not necessarily reflect the views of the NIH or the U.S. Department of Health and Human Services.

## Additional information

### Funding

| Funder | Grant reference number | Author |
| --- | --- | --- |
| National Institute of Neurological Disorders and Stroke | NINDS Intramural Research Program | Leonardo G Cohen |

The funders had no role in study design, data collection and interpretation, or the decision to submit the work for publication.

### Author contributions

Debadatta Dash, Conceptualization, Software, Formal analysis, Validation, Investigation, Visualization, Methodology, Writing – original draft, Writing – review and editing; Fumiaki Iwane, William Hayward, Roberto F Salamanca-Giron, Writing – review and editing; Marlene Bönstrup, Data curation, Writing – review and editing; Ethan R Buch, Conceptualization, Resources, Data curation, Software, Formal analysis, Supervision, Validation, Investigation, Visualization, Methodology, Writing – original draft, Project administration, Writing – review and editing; Leonardo G Cohen, Conceptualization, Resources, Data curation, Software, Formal analysis, Supervision, Funding acquisition, Validation, Investigation, Visualization, Methodology, Writing – original draft, Project administration, Writing – review and editing

### Author ORCIDs

Debadatta Dash ⓘ https://orcid.org/0000-0002-0543-0304
Roberto F Salamanca-Giron ⓘ https://orcid.org/0009-0008-5743-6805
Ethan R Buch ⓘ https://orcid.org/0000-0002-5443-8222
Leonardo G Cohen ⓘ https://orcid.org/0000-0002-1705-8773

### Ethics

Human subjects: The study was approved by the Combined Neuroscience Institutional Review Board of the National Institutes of Health (NIH). All participants provided written informed consent for the study.

Reviewer #1 (Public review): https://doi.org/10.7554/eLife.102475.4.sa1
Reviewer #2 (Public review): https://doi.org/10.7554/eLife.102475.4.sa2
Reviewer #3 (Public review): https://doi.org/10.7554/eLife.102475.4.sa3
Author response https://doi.org/10.7554/eLife.102475.4.sa4

## Additional files

### Supplementary files
MDAR checklist

### Data availability

All de-identified and permanently unlinked from all personal identifiable information (PII) data are publicly available on the OpenNeuro platform. All custom analysis code is available in a publicly accessible repository hosted on GitHub (copy archived at *Dash and hcps-ninds, 2025*).

The following dataset was generated:

| Author(s) | Year | Dataset title | Dataset URL | Database and Identifier |
|---|---|---|---|---|
| Bönstrup M, Buch ER, Cohen LG | 2025 | Skill learning and consolidation in healthy humans | https://doi.org/10.18112/openneuro.ds006502.v1.0.0 | OpenNeuro, 10.18112/openneuro.ds006502.v1.0.0 |

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
