## [Editor Report · eLife Assessment]

This **valuable** study asks how the neural representation of individual finger movements changes during the early periods of sequence learning. By combining a new method for extracting features from human magnetoencephalography data and decoding analyses, the authors provide **solid** evidence of an early, swift change in the brain regions correlated with sequence learning, including a set of previously unreported frontal cortical regions. The authors also show that offline contextualization during short rest periods is the basis for improved performance. Further confirmation of these results on multiple movement sequences would further strengthen the key claims.

---

## [Referee Report · Reviewer #1 (Public review)]

Summary:

This study addresses the issue of rapid skill learning and whether individual sequence elements (here: finger presses) are differentially represented in human MEG data. The authors use a decoding approach to classify individual finger elements, and accomplish an accuracy of around 94%. A relevant finding is that the neural representations of individual finger elements dynamically change over the course of learning. This would be highly relevant for any attempts to develop better brain machine interfaces - one now can decode individual elements within a sequence with high precision, but these representations are not static but develop over the course of learning.

Strengths:

The work follows from a large body of work from the same group on the behavioural and neural foundations of sequence learning. The behavioural task is well established a neatly designed to allow for tracking learning and how individual sequence elements contribute. The inclusion of short offline rest periods between learning epochs has been influential because it has revealed that a lot, if not most of the gains in behaviour (ie speed of finger movements) occur in these so-called micro-offline rest periods.

The authors use a range of new decoding techniques, and exhaustively interrogate their data in different ways, using different decoding approaches. Regardless of the approach, impressively high decoding accuracies are observed, but when using a hybrid approach that combines the MEG data in different ways, the authors observe decoding accuracies of individual sequence elements from the MEG data of up to 94%.

---

## [Referee Report · Reviewer #2 (Public review)]

Summary:

The current paper consists of two parts. The first part is the rigorous feature optimization of the MEG signal to decode individual finger identity performed in a sequence (4-1-3-2-4; 1~4 corresponds to little~index fingers of the left hand). By optimizing various parameters for the MEG signal, in terms of (i) reconstructed source activity in voxel- and parcel-level resolution and their combination, (ii) frequency bands, and (iii) time window relative to press onset for each finger movement, as well as the choice of decoders, the resultant "hybrid decoder" achieved extremely high decoding accuracy (~95%).

In the second part of the paper, armed with the successful 'hybrid decoder,' the authors asked how neural representation of individual finger movement that is embedded in a sequence, changes during a very early period of skill learning and whether and how such representational change can predict skill learning. They assessed the difference in MEG feature patterns between the first and the last press 4 in sequence 41324 at each training trial and found that the pattern differentiation progressively increased over the course of early learning trials. Additionally, they found that this pattern differentiation specifically occurred during the rest period rather than during the practice trial. With a significant correlation between the trial-by-trial profile of this pattern differentiation and that for accumulation of offline learning, the authors argue that such "contextualization" of finger movement in a sequence (e.g., what-where association) underlies the early improvement of sequential skill. This is an important and timely topic for the field of motor learning and beyond.

Strengths:

The use of temporally rich neural information (MEG signal) has a significant advantage over previous studies testing sequential representations using fMRI. This allowed the authors to examine the earliest period (= the first few minutes of training) of skill learning with finer temporal resolution. Through the optimization of MEG feature extraction, the current study achieved extremely high decoding accuracy (approx. 94%) compared to previous works. The finding of the early "contextualization" of the finger movement in a sequence and its correlation to early (offline) skill improvement is interesting and important. The comparison between "online" and "offline" pattern distance is a neat idea.

Weaknesses:

One potential weakness, in terms of the generality, is that the study assessed the single sequence, the "41324" across all participants. Future confirmation test of using different sequences would be important.

---

## [Referee Report · Reviewer #3 (Public review)]

Summary:

One goal of this paper is to introduce a new approach for highly accurate decoding of finger movements from human magnetoencephalography data via dimension reduction of a "multi-scale, hybrid" feature space. Following this decoding approach, the authors aim to show that early skill learning involves "contextualization" of the neural coding of individual movements, relative to their position in a sequence of consecutive movements. Furthermore, they aim to show that this "contextualization" develops primarily during short rest periods interspersed with skill training, and correlates with a performance metric which the authors interpret as an indicator of offline learning.

Strengths:

A strength of the paper is the innovative decoding approach, which achieves impressive decoding accuracies via dimension reduction of a "multi-scale, hybrid space". This hybrid-space approach follows the neurobiologically plausible idea of concurrent distribution of neural coding across local circuits as well as large-scale networks.

Weaknesses:

A clear weakness of the paper lies in the authors' conclusions regarding "contextualization". Several potential confounds, which partly arise from the experimental design, and which are described below, question the neurobiological implications proposed by the authors, and offer a simpler explanation of the results. Furthermore, the paper follows the assumption that short breaks result in offline skill learning, while recent evidence casts doubt on this assumption.

Specifically:

The authors interpret the ordinal position information captured by their decoding approach as a reflection of neural coding dedicated to the local context of a movement (Figure 4). One way to dissociate ordinal position information from information about the moving effectors is to train a classifier on one sequence, and test the classifier on other sequences that require the same movements, but in different positions (Kornysheva et al., Neuron 2019). In the present study, however, participants trained to repeat a single sequence (4-1-3-2-4). As a result, ordinal position information is potentially confounded by the fixed finger transitions around each of the two critical positions (first and fifth press). Across consecutive correct sequences, the first keypress in a given sequence was always preceded by a movement of the index finger (=last movement of the preceding sequence), and followed by a little finger movement. The last keypress, on the other hand, was always preceded by a ring finger movement, and followed by an index finger movement (=first movement of the next sequence). Figure 3 - supplement 5 shows that finger identity can be decoded with high accuracy (>70%) across a large time window around the time of the keypress, up to at least {plus minus}100 ms (and likely beyond, given that decoding accuracy is still high at the boundaries of the window depicted in that figure). This time window approaches the keypress transition times in this study. Given that distinct finger transitions characterized the first and fifth keypress, the classifier could thus rely on persistent (or "lingering") information from the preceding finger movement, and/or "preparatory" information about the subsequent finger movement, in order to dissociate the first and fifth keypress. Currently, the manuscript provides little evidence that the context information captured by the decoding approach is more than a by-product of temporally extended, and therefore overlapping, but independent neural representations of consecutive keypresses that are executed in close temporal proximity - rather than a neural representation dedicated to context.

During the review process, the authors pointed out that a "mixing" of temporally overlapping information from consecutive keypresses, as described above, should result in systematic misclassifications and therefore be detectable in the confusion matrices in Figures 3C and 4B, which indeed do not provide any evidence that consecutive keypresses are systematically confused. However, such absence of evidence (of systematic misclassification) should be interpreted with caution. The authors also reported that there was only a weak relation between inter-press intervals and "online contextualization" (Figure 5 - figure supplement 6), however, their analysis suprisingly includes a keypress transition that is shared between OP1 and OP5 ("4-4"), rather than focusing solely on the two distinctive transitions ("2-4" and "4-1").

Such temporal overlap of consecutive, independent finger representations may also account for the dynamics of "ordinal coding"/"contextualization", i.e., the increase in 2-class decoding accuracy, across Day 1 (Figure 4C). As learning progresses, both tapping speed and the consistency of keypress transition times increase (Figure 1), i.e., consecutive keypresses are closer in time, and more consistently so. As a result, information related to a given keypress is increasingly overlapping in time with information related to the preceding and subsequent keypresses. Furthermore, learning should increase the number of (consecutively) correct sequences, and, thus, the consistency of finger transitions. Therefore, the increase in 2-class decoding accuracy may simply reflect an increasing overlap in time of increasingly consistent information from consecutive keypresses, which allows the classifier to dissociate the first and fifth keypress more reliably as learning progresses, simply based on the characteristic finger transitions associated with each. In other words, given that the physical context of a given keypress changes as learning progresses - keypresses move closer together in time, and are more consistently correct - it seems problematic to conclude that the mental representation of that context changes. During the review process, authors pointed at absence of evidence of a relation between tapping speed and "ordinal coding" (Figure 5 - figure supplement 7). However, a rigorous test of the idea that the mental representation of context changes would require a task design in which the physical context remains constant.

A similar difference in physical context may explain why neural representation distances ("differentiation") differ between rest and practice (Figure 5). The authors define "offline differentiation" by comparing the hybrid space features of the last index finger movement of a trial (ordinal position 5) and the first index finger movement of the next trial (ordinal position 1). However, the latter is not only the first movement in the sequence, but also the very first movement in that trial (at least in trials that started with a correct sequence), i.e., not preceded by any recent movement. In contrast, the last index finger of the last correct sequence in the preceding trial includes the characteristic finger transition from the fourth to the fifth movement. Thus, there is more overlapping information arising from the consistent, neighbouring keypresses for the last index finger movement, compared to the first index finger movement of the next trial. A strong difference (larger neural representation distance) between these two movements is, therefore, not surprising, given the task design, and this difference is also expected to increase with learning, given the increase in tapping speed, and the consequent stronger overlap in representations for consecutive keypresses.

A further complication in interpreting the results stems from the visual feedback that participants received during the task. Each keypress generated an asterisk shown above the string on the screen. It is not clear why the authors introduced this complicating visual feedback in their task, besides consistency with their previous studies. The resulting systematic link between the pattern of visual stimulation (the number of asterisks on the screen) and the ordinal position of a keypress makes the interpretation of "contextual information" that differentiates between ordinal positions difficult. While the authors report the surprising finding that their eye-tracking data could not predict asterisk position on the task display above chance level, the mean gaze position seemed to vary systematically as a function of ordinal position of a movement - see Figure 4 - figure supplement 3.

The authors report a significant correlation between "offline differentiation" and cumulative micro-offline gains. However, to reach the conclusion that "the degree of representational differentiation -particularly prominent over rest intervals - correlated with skill gains.", the critical question is rather whether "offline differentiation" correlates with micro-offline gains (not with cumulative micro-offline gains). That is, does the degree to which representations differentiate "during" a given rest period correlate with the degree to which performance improves from before to after the same rest period (not: does "offline differentiation" in a given rest period correlate with the degree to which performance has improved "during" all rest periods up to the current rest period - but this is what Figure 5 - figure supplements 1 and 4 show).

The authors follow the assumption that micro-offline gains reflect offline learning. However, there is no compelling evidence in the literature, and no evidence in the present manuscript, that micro-offline gains (during any training phase) reflect offline learning. Instead, emerging evidence in the literature indicates that they do not (Das et al., bioRxiv 2024), and instead reflect transient performance benefits when participants train with breaks, compared to participants who train without breaks, however, these benefits vanish within seconds after training if both groups of participants perform under comparable conditions (Das et al., bioRxiv 2024). During the review process, the authors argued that differences in the design between Das et al. (2024) on the one hand (Experiments 1 and 2), and the study by Bönstrup et al. (2019) on the other hand, may have prevented Das et al. (2024) from finding the assumed (lasting) learning benefit by micro-offline consolidation. However, the Supplementary Material of Das et al. (2024) includes an experiment (Experiment S1) whose design closely follows the early learning phase of Bönstrup et al. (2019), and which, nevertheless, demonstrates that there is no lasting benefit of taking breaks for the acquired skill level, despite the presence of micro-offline gains.

Along these lines, the authors argue that their practice schedule "minimizes reactive inhibition effects", in particular their short practice periods of 10 seconds each. However, 10 seconds are sufficient to result in motor slowing, as report in Bächinger et al., elife 2019, or Rodrigues et al., Exp Brain Res 2009.

An important conceptual problem with the current study is that the authors conclude that performance improves, and representation manifolds differentiate, "during" rest periods. However, micro-offline gains (as well as offline contextualization) are computed from data obtained during practice, not rest, and may, thus, just as well reflect a change that occurs "online", e.g., at the very onset of practice (like pre-planning) or throughout practice (like fatigue, or reactive inhibition).

The authors' conclusion that "low-frequency oscillations (LFOs) result in higher decoding accuracy compared to other narrow-band activity" should be taken with caution, given that the critical decoding analysis for this conclusion was based on data averaged across a time window of 200 ms (Figure 2), essentially smoothing out higher frequency components.

---

## [Author Response]

The following is the authors’ response to the previous reviews

Overview of reviewer's concerns after peer review:As for the initial submission, the reviewers' unanimous opinion is that the authors should perform additional controls to show that their key findings may not be affected by experimental or analysis artefacts, and clarify key aspects of their core methods, chiefly:(1) The fact that their extremely high decoding accuracy is driven by frequency bands that would reflect the key press movements and that these are located bilaterally in frontal brain regions (with the task being unilateral) are seen as key concerns,

The above statement that decoding was driven by bilateral frontal brain regions is not entirely consistent with our results. The confusion was likely caused by the way we originally presented our data in Figure 2. We have revised that figure to make it more clear that decoding performance at both the parcel- (Figure 2B) and voxel-space (Figure 2C) level is predominantly driven by contralateral (as opposed to ipsilateral) sensorimotor regions. Figure 2D, which highlights bilateral sensorimotor and premotor regions, displays accuracy of individual regional voxel-space decoders assessed independently. This was the criteria used to determine which regional voxel-spaces were included in the hybridspace decoder. This result is not surprising given that motor and premotor regions are known to display adaptive interhemispheric interactions during motor sequence learning [1, 2], and particularly so when the skill is performed with the non-dominant hand [3-5]. We now discuss this important detail in the revised manuscript:

Discussion (lines 348-353)

“The whole-brain parcel-space decoder likely emphasized more stable activity patterns in contralateral frontoparietal regions that differed between individual finger movements [21,35], while the regional voxel-space decoder likely incorporated information related to adaptive interhemispheric interactions operating during motor sequence learning [32,36,37], particularly pertinent when the skill is performed with the non-dominant hand [38-40].”

We now also include new control analyses that directly address the potential contribution of movement-related artefact to the results. These changes are reported in the revised manuscript as follows:

Results (lines 207-211):

“An alternate decoder trained on ICA components labeled as movement or physiological artefacts (e.g. – head movement, ECG, eye movements and blinks; Figure 3 – figure supplement 3A, D) and removed from the original input feature set during the pre-processing stage approached chance-level performance (Figure 4 – figure supplement 3), indicating that the 4-class hybrid decoder results were not driven by task-related artefacts.”

Results (lines 261-268):

“As expected, the 5-class hybrid-space decoder performance approached chance levels when tested with randomly shuffled keypress labels (18.41%± SD 7.4% for Day 1 data; Figure 4 – figure supplement 3C). Task-related eye movements did not explain these results since an alternate 5-class hybrid decoder constructed from three eye movement features (gaze position at the KeyDown event, gaze position 200ms later, and peak eye movement velocity within this window; Figure 4 – figure supplement 3A) performed at chance levels (cross-validated test accuracy = 0.2181; Figure 4 – figure supplement 3B, C). “

Discussion (Lines 362-368):

“Task-related movements—which also express in lower frequency ranges—did not explain these results given the near chance-level performance of alternative decoders trained on (a) artefact-related ICA components removed during MEG preprocessing (Figure 3 – figure supplement 3A-C) and on (b) task-related eye movement features (Figure 4 – figure supplement 3B, C). This explanation is also inconsistent with the minimal average head motion of 1.159 mm (± 1.077 SD) across the MEG recording (Figure 3 – figure supplement 3D).“

(2) Relatedly, the use of a wide time window (~200 ms) for a 250-330 ms typing speed makes it hard to pinpoint the changes underpinning learning,

The revised manuscript now includes analyses carried out with decoding time windows ranging from 50 to 250ms in duration. These additional results are now reported in:

Results (lines 258-261):

“The improved decoding accuracy is supported by greater differentiation in neural representations of the index finger keypresses performed at positions 1 and 5 of the sequence (Figure 4A), and by the trial-by-trial increase in 2-class decoding accuracy over early learning (Figure 4C) across different decoder window durations (Figure 4 – figure supplement 2).”

Results (lines 310-312):

“Offline contextualization strongly correlated with cumulative micro-offline gains (r = 0.903, R^2^ = 0.816, p < 0.001; Figure 5 – figure supplement 1A, inset) across decoder window durations ranging from 50 to 250ms (Figure 5 – figure supplement 1B, C).“

Discussion (lines 382-385):

“This was further supported by the progressive differentiation of neural representations of the index finger keypress (Figure 4A) and by the robust trial-bytrial increase in 2-class decoding accuracy across time windows ranging between 50 and 250ms (Figure 4C; Figure 4 – figure supplement 2).”

Discussion (lines 408-9):

“Offline contextualization consistently correlated with early learning gains across a range of decoding windows (50–250ms; Figure 5 – figure supplement 1).”

(3) These concerns make it hard to conclude from their data that learning is mediated by "contextualisation" ---a key claim in the manuscript;

We believe the revised manuscript now addresses all concerns raised in Editor points 1 and 2.

(4) The hybrid voxel + parcel space decoder ---a key contribution of the paper--- is not clearly explained;

We now provide additional details regarding the hybrid-space decoder approach in the following sections of the revised manuscript:

Results (lines 158-172):

“Next, given that the brain simultaneously processes information more efficiently across multiple spatial and temporal scales [28, 32, 33], we asked if the combination of lower resolution whole-brain and higher resolution regional brain activity patterns further improve keypress prediction accuracy. We constructed hybrid-space decoders (N = 1295 ± 20 features; Figure 3A) combining whole-brain parcel-space activity (n = 148 features; Figure 2B) with regional voxel-space activity from a datadriven subset of brain areas (n = 1147 ± 20 features; Figure 2D). This subset covers brain regions showing the highest regional voxel-space decoding performances (top regions across all subjects shown in Figure 2D; Methods – Hybrid Spatial Approach).

[…]

Note that while features from contralateral brain regions were more important for whole-brain decoding (in both parcel- and voxel-spaces), regional voxel-space decoders performed best for bilateral sensorimotor areas on average across the group. Thus, a multi-scale hybrid-space representation best characterizes the keypress action manifolds.”

Results (lines 275-282):

“We used a Euclidian distance measure to evaluate the differentiation of the neural representation manifold of the same action (i.e. - an index-finger keypress) executed within different local sequence contexts (i.e. - ordinal position 1 vs. ordinal position 5; Figure 5). To make these distance measures comparable across participants, a new set of classifiers was then trained with group-optimal parameters (i.e. – broadband hybrid-space MEG data with subsequent manifold extraction Figure 3 – figure supplements 2) and LDA classifiers (Figure 3 – figure supplements 7) trained on 200ms duration windows aligned to the KeyDown event (see Methods, Figure 3 – figure supplements 5). “

Discussion (lines 341-360):

“The initial phase of the study focused on optimizing the accuracy of decoding individual finger keypresses from MEG brain activity. Recent work showed that the brain simultaneously processes information more efficiently across multiple—rather than a single—spatial scale(s) [28, 32]. To this effect, we developed a novel hybridspace approach designed to integrate neural representation dynamics over two different spatial scales: (1) whole-brain parcel-space (i.e. – spatial activity patterns across all cortical brain regions) and (2) regional voxel-space (i.e. – spatial activity patterns within select brain regions) activity. We found consistent spatial differences between whole-brain parcel-space feature importance (predominantly contralateral frontoparietal, Figure 2B) and regional voxel-space decoder accuracy (bilateral sensorimotor regions, Figure 2D). The *whole-brain parcel-space* decoder likely emphasized more stable activity patterns in contralateral frontoparietal regions that differed between individual finger movements [21, 35], while the *regional voxelspace* decoder likely incorporated information related to adaptive interhemispheric interactions operating during motor sequence learning [32, 36, 37], particularly pertinent when the skill is performed with the non-dominant hand [38-40]. The observation of increased cross-validated test accuracy (as shown in Figure 3 – Figure Supplement 6) indicates that the spatially overlapping information in parcel- and voxel-space time-series in the hybrid decoder was complementary, rather than redundant [41]. The hybrid-space decoder which achieved an accuracy exceeding 90%—and robustly generalized to Day 2 across trained and untrained sequences— surpassed the performance of both parcel-space and voxel-space decoders and compared favorably to other neuroimaging-based finger movement decoding strategies [6, 24, 42-44].”

Methods (lines 636-647):

“Hybrid Spatial Approach. First, we evaluated the decoding performance of each individual brain region in accurately labeling finger keypresses from regional voxelspace (i.e. - all voxels within a brain region as defined by the Desikan-Killiany Atlas) activity. Brain regions were then ranked from 1 to 148 based on their decoding accuracy at the group level. In a stepwise manner, we then constructed a “hybridspace” decoder by incrementally concatenating regional voxel-space activity of brain regions—starting with the top-ranked region—with whole-brain parcel-level features and assessed decoding accuracy. Subsequently, we added the regional voxel-space features of the second-ranked brain region and continued this process until decoding accuracy reached saturation. The optimal “hybrid-space” input feature set over the group included the 148 parcel-space features and regional voxelspace features from a total of 8 brain regions (bilateral superior frontal, middle frontal, pre-central and post-central; N = 1295 ± 20 features).”

(5) More controls are needed to show that their decoder approach is capturing a neural representation dedicated to context rather than independent representations of consecutive keypresses;

These controls have been implemented and are now reported in the manuscript:

Results (lines 318-328):

“Within-subject correlations were consistent with these group-level findings. The average correlation between offline contextualization and micro-offline gains within individuals was significantly greater than zero (Figure 5 – figure supplement 4, left; t = 3.87, p = 0.00035, df = 25, Cohen's d = 0.76) and stronger than correlations between online contextualization and either micro-online (Figure 5 – figure supplement 4, middle; t = 3.28, p = 0.0015, df = 25, Cohen's d = 1.2) or micro-offline gains (Figure 5 – figure supplement 4, right; t = 3.7021, p = 5.3013e-04, df = 25, Cohen's d = 0.69). These findings were not explained by behavioral changes of typing rhythm (t = -0.03, *p* = 0.976; Figure 5 – figure supplement 5), adjacent keypress transition times (R2 = 0.00507, F[1,3202] = 16.3; Figure 5 – figure supplement 6), or overall typing speed (between-subject; R2 = 0.028, *p* = 0.41; Figure 5 – figure supplement 7).”

Results (lines 385-390):

“Further, the 5-class classifier—which directly incorporated information about the sequence location context of each keypress into the decoding pipeline—improved decoding accuracy relative to the 4-class classifier (Figure 4C). Importantly, testing on Day 2 revealed specificity of this representational differentiation for the trained skill but not for the same keypresses performed during various unpracticed control sequences (Figure 5C).”

Discussion (lines 408-423):

“Offline contextualization consistently correlated with early learning gains across a range of decoding windows (50–250ms; Figure 5 – figure supplement 1). This result remained unchanged when measuring offline contextualization between the last and second sequence of consecutive trials, inconsistent with a possible confounding effect of pre-planning [30] (Figure 5 – figure supplement 2A). On the other hand, online contextualization did not predict learning (Figure 5 – figure supplement 3). Consistent with these results the average within-subject correlation between offline contextualization and micro-offline gains was significantly stronger than withinsubject correlations between online contextualization and either micro-online or micro-offline gains (Figure 5 – figure supplement 4).

Offline contextualization was not driven by trial-by-trial behavioral differences, including typing rhythm (Figure 5 – figure supplement 5) and adjacent keypress transition times (Figure 5 – figure supplement 6) nor by between-subject differences in overall typing speed (Figure 5 – figure supplement 7)—ruling out a reliance on differences in the temporal overlap of keypresses. Importantly, offline contextualization documented on Day 1 stabilized once a performance plateau was reached (trials 11-36), and was retained on Day 2, documenting overnight consolidation of the differentiated neural representations.”

(6) The need to show more convincingly that their data is not affected by head movements, e.g., by regressing out signal components that are correlated with the fiducial signal;

We now include data in Figure 3 – figure supplement 3D showing that head movement was minimal in all participants (mean of 1.159 mm ± 1.077 SD). Further, the requested additional control analyses have been carried out and are reported in the revised manuscript:

Results (lines 204-211):

“Testing the keypress state (4-class) hybrid decoder performance on Day 1 after randomly shupling keypress labels for held-out test data resulted in a performance drop approaching expected chance levels (22.12%± SD 9.1%; Figure 3 – figure supplement 3C). An alternate decoder trained on ICA components labeled as movement or physiological artefacts (e.g. – head movement, ECG, eye movements and blinks; Figure 3 – figure supplement 3A, D) and removed from the original input feature set during the pre-processing stage approached chance-level performance (Figure 4 – figure supplement 3), indicating that the 4-class hybrid decoder results were not driven by task-related artefacts.” Results (lines 261-268):

“As expected, the 5-class hybrid-space decoder performance approached chance levels when tested with randomly shuffled keypress labels (18.41%± SD 7.4% for Day 1 data; Figure 4 – figure supplement 3C). Task-related eye movements did not explain these results since an alternate 5-class hybrid decoder constructed from three eye movement features (gaze position at the KeyDown event, gaze position 200ms later, and peak eye movement velocity within this window; Figure 4 – figure supplement 3A) performed at chance levels (cross-validated test accuracy = 0.2181; Figure 4 – figure supplement 3B, C). “

Discussion (Lines 362-368):

“Task-related movements—which also express in lower frequency ranges—did not explain these results given the near chance-level performance of alternative decoders trained on (a) artefact-related ICA components removed during MEG preprocessing (Figure 3 – figure supplement 3A-C) and on (b) task-related eye movement features (Figure 4 – figure supplement 3B, C). This explanation is also inconsistent with the minimal average head motion of 1.159 mm (± 1.077 SD) across the MEG recording (Figure 3 – figure supplement 3D). “

(7) The offline neural representation analysis as executed is a bit odd, since it seems to be based on comparing the last key press to the first key press of the next sequence, rather than focus on the inter-sequence interval

While we previously evaluated replay of skill sequences during rest intervals, identification of how offline reactivation patterns of a single keypress state representation evolve with learning presents non-trivial challenges. First, replay events tend to occur in clusters with irregular temporal spacing as previously shown by our group and others. Second, replay of experienced sequences is intermixed with replay of sequences that have never been experienced but are possible. Finally, and perhaps the most significant issue, replay is temporally compressed up to 20x with respect to the behavior [6]. That means our decoders would need to accurately evaluate spatial pattern changes related to individual keypresses over much smaller time windows (i.e. - less than 10 ms) than evaluated here. This future work, which is undoubtably of great interest to our research group, will require more substantial tool development before we can apply them to this question. We now articulate this future direction in the Discussion:

Discussion (lines 423-427):

“A possible neural mechanism supporting contextualization could be the emergence and stabilization of conjunctive “what–where” representations of procedural memories [64] with the corresponding modulation of neuronal population dynamics [65, 66] during early learning. Exploring the link between contextualization and neural replay could provide additional insights into this issue [6, 12, 13, 15].”

(8) And this analysis could be confounded by the fact that they are comparing the last element in a sequence vs the first movement in a new one.

We have now addressed this control analysis in the revised manuscript:

Results (Lines 310-316)

“Offline contextualization strongly correlated with cumulative micro-offline gains (r = 0.903, R^2^ = 0.816, p < 0.001; Figure 5 – figure supplement 1A, inset) across decoder window durations ranging from 50 to 250ms (Figure 5 – figure supplement 1B, C). The offline contextualization between the final sequence of each trial and the second sequence of the subsequent trial (excluding the first sequence) yielded comparable results. This indicates that pre-planning at the start of each practice trial did not directly influence the offline contextualization measure [30] (Figure 5 – figure supplement 2A, *1st vs. 2nd Sequence approaches*).”

Discussion (lines 408-416):

“Offline contextualization consistently correlated with early learning gains across a range of decoding windows (50–250ms; Figure 5 – figure supplement 1). This result remained unchanged when measuring offline contextualization between the last and second sequence of consecutive trials, inconsistent with a possible confounding effect of pre-planning [30] (Figure 5 – figure supplement 2A). On the other hand, online contextualization did not predict learning (Figure 5 – figure supplement 3). Consistent with these results the average within-subject correlation between offline contextualization and micro-offline gains was significantly stronger than within-subject correlations between online contextualization and either micro-online or micro-offline gains (Figure 5 – figure supplement 4).”

It also seems to be the case that many analyses suggested by the reviewers in the first round of revisions that could have helped strengthen the manuscript have not been included (they are only in the rebuttal). Moreover, some of the control analyses mentioned in the rebuttal seem not to be described anywhere, neither in the manuscript, nor in the rebuttal itself; please double check that.

All suggested analyses carried out and mentioned are now in the revised manuscript.

**eLife Assessment**
This valuable study investigates how the neural representation of individual finger movements changes during the early period of sequence learning. By combining a new method for extracting features from human magnetoencephalography data and decoding analyses, the authors provide incomplete evidence of an early, swift change in the brain regions correlated with sequence learning…

We have now included all the requested control analyses supporting “an early, swift change in the brain regions correlated with sequence learning”:

The addition of more control analyses to rule out that head movement artefacts influence the findings,

We now include data in Figure 3 – figure supplement 3D showing that head movement was minimal in all participants (mean of 1.159 mm ± 1.077 SD). Further, we have implemented the requested additional control analyses addressing this issue:

Results (lines 207-211):

“An alternate decoder trained on ICA components labeled as movement or physiological artefacts (e.g. – head movement, ECG, eye movements and blinks; Figure 3 – figure supplement 3A, D) and removed from the original input feature set during the pre-processing stage approached chance-level performance (Figure 4 – figure supplement 3), indicating that the 4-class hybrid decoder results were not driven by task-related artefacts.”

Results (lines 261-268):

“As expected, the 5-class hybrid-space decoder performance approached chance levels when tested with randomly shuffled keypress labels (18.41%± SD 7.4% for Day 1 data; Figure 4 – figure supplement 3C). Task-related eye movements did not explain these results since an alternate 5-class hybrid decoder constructed from three eye movement features (gaze position at the KeyDown event, gaze position 200ms later, and peak eye movement velocity within this window; Figure 4 – figure supplement 3A) performed at chance levels (cross-validated test accuracy = 0.2181; Figure 4 – figure supplement 3B, C). “

Discussion (Lines 362-368):

“Task-related movements—which also express in lower frequency ranges—did not explain these results given the near chance-level performance of alternative decoders trained on (a) artefact-related ICA components removed during MEG preprocessing (Figure 3 – figure supplement 3A-C) and on (b) task-related eye movement features (Figure 4 – figure supplement 3B, C). This explanation is also inconsistent with the minimal average head motion of 1.159 mm (± 1.077 SD) across the MEG recording (Figure 3 – figure supplement 3D).“

and to further explain the proposal of offline contextualization during short rest periods as the basis for improvement performance would strengthen the manuscript.

We have edited the manuscript to clarify that the degree of representational differentiation (contextualization) parallels skill learning. We have no evidence at this point to indicate that “offline contextualization during short rest periods is the basis for improvement in performance”. The following areas of the revised manuscript now clarify this point:

Summary (Lines 455-458):

“In summary, individual sequence action representations contextualize during early learning of a new skill and the degree of differentiation parallels skill gains. Differentiation of the neural representations developed during rest intervals of early learning to a larger extent than during practice in parallel with rapid consolidation of skill.”

Additional control analyses are also provided supporting a link between offline contextualization and early learning:

Results (lines 302-318):

“The Euclidian distance between neural representations of Index_OP1_ (i.e. - index finger keypress at ordinal position 1 of the sequence) and Index_OP5_ (i.e. - index finger keypress at ordinal position 5 of the sequence) increased progressively during early learning (Figure 5A)—predominantly during rest intervals (offline contextualization) rather than during practice (online) (t = 4.84, p < 0.001, df = 25, Cohen's d = 1.2; Figure 5B; Figure 5 – figure supplement 1A). An alternative online contextualization determination equaling the time interval between online and offline comparisons (*Trial-based;* 10 seconds between Index_OP1_ and Index_OP5_ observations in both cases) rendered a similar result (Figure 5 – figure supplement 2B).

Offline contextualization strongly correlated with cumulative micro-offline gains (r = 0.903, R^2^ = 0.816, p < 0.001; Figure 5 – figure supplement 1A, inset) across decoder window durations ranging from 50 to 250ms (Figure 5 – figure supplement 1B, C). The offline contextualization between the final sequence of each trial and the second sequence of the subsequent trial (excluding the first sequence) yielded comparable results. This indicates that pre-planning at the start of each practice trial did not directly influence the offline contextualization measure [30] (Figure 5 – figure supplement 2A, 1st vs. 2nd Sequence approaches). Conversely, online contextualization (using either measurement approach) did not explain early online learning gains (i.e. – Figure 5 – figure supplement 3).”

**Public Reviews:**

**Reviewer #1 (Public review):**
Summary:This study addresses the issue of rapid skill learning and whether individual sequence elements (here: finger presses) are differentially represented in human MEG data. The authors use a decoding approach to classify individual finger elements and accomplish an accuracy of around 94%. A relevant finding is that the neural representations of individual finger elements dynamically change over the course of learning. This would be highly relevant for any attempts to develop better brain machine interfaces - one now can decode individual elements within a sequence with high precision, but these representations are not static but develop over the course of learning.Strengths:The work follows a large body of work from the same group on the behavioural and neural foundations of sequence learning. The behavioural task is well established a neatly designed to allow for tracking learning and how individual sequence elements contribute. The inclusion of short offline rest periods between learning epochs has been influential because it has revealed that a lot, if not most of the gains in behaviour (ie speed of finger movements) occur in these so-called micro-offline rest periods.The authors use a range of new decoding techniques, and exhaustively interrogate their data in different ways, using different decoding approaches. Regardless of the approach, impressively high decoding accuracies are observed, but when using a hybrid approach that combines the MEG data in different ways, the authors observe decoding accuracies of individual sequence elements from the MEG data of up to 94%.Weaknesses:A formal analysis and quantification of how head movement may have contributed to the results should be included in the paper or supplemental material. The type of correlated head movements coming from vigorous key presses aren't necessarily visible to the naked eye, and even if arms etc are restricted, this will not preclude shoulder, neck or head movement necessarily; if ICA was conducted, for example, the authors are in the position to show the components that relate to such movement; but eye-balling the data would not seem sufficient. The related issue of eye movements is addressed via classifier analysis. A formal analysis which directly accounts for finger/eye movements in the same analysis as the main result (ie any variance related to these factors) should be presented.

We now present additional data related to head (Figure 3 – figure supplement 3; note that average measured head movement across participants was 1.159 mm ± 1.077 SD) and eye movements (Figure 4 – figure supplement 3) and have implemented the requested control analyses addressing this issue. They are reported in the revised manuscript in the following locations: Results (lines 207-211), Results (lines 261-268), Discussion (Lines 362-368).

This reviewer recommends inclusion of a formal analysis that the intra-vs inter parcels are indeed completely independent. For example, the authors state that the inter-parcel features reflect "lower spatially resolved whole-brain activity patterns or global brain dynamics". A formal quantitative demonstration that the signals indeed show "complete independence" (as claimed by the authors) and are orthogonal would be helpful.

Please note that we never claim in the manuscript that the parcel-space and regional voxelspace features show “complete independence”. More importantly, input feature orthogonality is not a requirement for the machine learning-based decoding methods utilized in the present study while non-redundancy is [7] (a requirement satisfied by our data, see below). Finally, our results show that the hybrid space decoder out-performed all other methods even after input features were fully orthogonalized with LDA (the procedure used in all contextualization analyses) or PCA dimensionality reduction procedures prior to the classification step (Figure 3 – figure supplement 2).

Relevant to this issue, please note that if spatially overlapping parcel- and voxel-space timeseries only provided redundant information, inclusion of both as input features should increase model over-fitting to the training dataset and decrease overall cross-validated test accuracy [8]. In the present study however, we see the opposite effect on decoder performance. First, Figure 3 – figure supplement 1 & 2 clearly show that decoders constructed from hybrid-space features outperform the other input feature (sensor-, wholebrain parcel- and whole-brain voxel-) spaces in every case (e.g. – wideband, all narrowband frequency ranges, and even after the input space is fully orthogonalized through dimensionality reduction procedures prior to the decoding step). Furthermore, Figure 3 – figure supplement 6 shows that hybrid-space decoder performance supers when parceltime series that spatially overlap with the included regional voxel-spaces are removed from the input feature set.

We state in the Discussion (lines 353-356)

“The observation of increased cross-validated test accuracy (as shown in Figure 3 – Figure Supplement 6) indicates that the spatially overlapping information in parcel- and voxel-space time-series in the hybrid decoder was complementary, rather than redundant [41].”

To gain insight into the complimentary information contributed by the two spatial scales to the hybrid-space decoder, we first independently computed the matrix rank for whole-brain parcel- and voxel-space input features for each participant (shown in Author response image 1). The results indicate that whole-brain parcel-space input features are full rank (rank = 148) for all participants (i.e. - MEG activity is orthogonal between all parcels). The matrix rank of voxelspace input features (rank = 267± 17 SD), exceeded the parcel-space rank for all participants and approached the number of useable MEG sensor channels (n = 272). Thus, voxel-space features provide both additional and complimentary information to representations at the parcel-space scale.

**Author response image 1. sa4fig1:** Matrix rank computed for whole-brain parcel- and voxel-space time-series in individual subjects across the training run. The results indicate that whole-brain parcel-space input features are full rank (rank = 148) for all participants (i.e. - MEG activity is orthogonal between all parcels). The matrix rank of voxel-space input features (rank = 267 ± 17 SD), on the other hand, approached the number of useable MEG sensor channels (n = 272). Although not full rank, the voxel-space rank exceeded the parcel-space rank for all participants. Thus, some voxel-space features provide additional orthogonal information to representations at the parcel-space scale. An expression of this is shown in the correlation distribution between parcel and constituent voxel time-series in Figure 2—figure Supplement 2.

Figure 2—figure Supplement 2 in the revised manuscript now shows that the degree of dependence between the two spatial scales varies over the regional voxel-space. That is, some voxels within a given parcel correlate strongly with the time-series of the parcel they belong to, while others do not. This finding is consistent with a documented increase in correlational structure of neural activity across spatial scales that does not reflect perfect dependency or orthogonality [9]. Notably, the regional voxel-spaces included in the hybridspace decoder are significantly less correlated with the averaged parcel-space time-series than excluded voxels. We now point readers to this new figure in the results.

Taken together, these results indicate that the multi-scale information in the hybrid feature set is complimentary rather than orthogonal. This is consistent with the idea that hybridspace features better represent multi-scale temporospatial dynamics reported to be a fundamental characteristic of how the brain stores and adapts memories, and generates behavior across species [9].

**Reviewer #2 (Public review):**
Summary:The current paper consists of two parts. The first part is the rigorous feature optimization of the MEG signal to decode individual finger identity performed in a sequence (4-1-3-2-4; 1~4 corresponds to little~index fingers of the left hand). By optimizing various parameters for the MEG signal, in terms of (i) reconstructed source activity in voxel- and parcel-level resolution and their combination, (ii) frequency bands, and (iii) time window relative to press onset for each finger movement, as well as the choice of decoders, the resultant "hybrid decoder" achieved extremely high decoding accuracy (~95%). This part seems driven almost by pure engineering interest in gaining as high decoding accuracy as possible.In the second part of the paper, armed with the successful 'hybrid decoder,' the authors asked more scientific questions about how neural representation of individual finger movement that is embedded in a sequence, changes during a very early period of skill learning and whether and how such representational change can predict skill learning. They assessed the difference in MEG feature patterns between the first and the last press 4 in sequence 41324 at each training trial and found that the pattern differentiation progressively increased over the course of early learning trials. Additionally, they found that this pattern differentiation specifically occurred during the rest period rather than during the practice trial. With a significant correlation between the trial-by-trial profile of this pattern differentiation and that for accumulation of offline learning, the authors argue that such "contextualization" of finger movement in a sequence (e.g., what-where association) underlies the early improvement of sequential skill. This is an important and timely topic for the field of motor learning and beyond.Strengths:Each part has its own strength. For the first part, the use of temporally rich neural information (MEG signal) has a significant advantage over previous studies testing sequential representations using fMRI. This allowed the authors to examine the earliest period (= the first few minutes of training) of skill learning with finer temporal resolution. Through the optimization of MEG feature extraction, the current study achieved extremely high decoding accuracy (approx. 94%) compared to previous works. For the second part, the finding of the early "contextualization" of the finger movement in a sequence and its correlation to early (offline) skill improvement is interesting and important. The comparison between "online" and "offline" pattern distance is a neat idea.Weaknesses:Despite the strengths raised, the specific goal for each part of the current paper, i.e., achieving high decoding accuracy and answering the scientific question of early skill learning, seems not to harmonize with each other very well. In short, the current approach, which is solely optimized for achieving high decoding accuracy, does not provide enough support and interpretability for the paper's interesting scientific claim. This reminds me of the accuracy-explainability tradeoff in machine learning studies (e.g., Linardatos et al., 2020). More details follow.There are a number of different neural processes occurring before and after a key press, such as planning of upcoming movement and ahead around premotor/parietal cortices, motor command generation in primary motor cortex, sensory feedback related processes in sensory cortices, and performance monitoring/evaluation around the prefrontal area. Some of these may show learning-dependent change and others may not.

In this paper, the focus as stated in the Introduction was to evaluate “the millisecond-level differentiation of discrete action representations during learning”, a proposal that first required the development of more accurate computational tools. Our first step, reported here, was to develop that tool. With that in hand, we then proceeded to test if neural representations differentiated during early skill learning. Our results showed they did. Addressing the question the Reviewer asks is part of exciting future work, now possible based on the results presented in this paper. We acknowledge this issue in the revised Discussion:

Discussion (Lines 428-434):

“In this study, classifiers were trained on MEG activity recorded during or immediately after each keypress, emphasizing neural representations related to action execution, memory consolidation and recall over those related to planning. An important direction for future research is determining whether separate decoders can be developed to distinguish the representations or networks separately supporting these processes. Ongoing work in our lab is addressing this question. The present accuracy results across varied decoding window durations and alignment with each keypress action support the feasibility of this approach (Figure 3—figure supplement 5).”

Given the use of whole-brain MEG features with a wide time window (up to ~200 ms after each key press) under the situation of 3~4 Hz (i.e., 250~330 ms press interval) typing speed, these different processes in different brain regions could have contributed to the expression of the "contextualization," making it difficult to interpret what really contributed to the "contextualization" and whether it is learning related. Critically, the majority of data used for decoder training has the chance of such potential overlap of signal, as the typing speed almost reached a plateau already at the end of the 11th trial and stayed until the 36th trial. Thus, the decoder could have relied on such overlapping features related to the future presses. If that is the case, a gradual increase in "contextualization" (pattern separation) during earlier trials makes sense, simply because the temporal overlap of the MEG feature was insufficient for the earlier trials due to slower typing speed. Several direct ways to address the above concern, at the cost of decoding accuracy to some degree, would be either using the shorter temporal window for the MEG feature or training the model with the early learning period data only (trials 1 through 11) to see if the main results are unaffected would be some example.

We now include additional analyses carried out with decoding time windows ranging from 50 to 250ms in duration, which have been added to the revised manuscript as follows:

Results (lines 258-261):

“The improved decoding accuracy is supported by greater differentiation in neural representations of the index finger keypresses performed at positions 1 and 5 of the sequence (Figure 4A), and by the trial-by-trial increase in 2-class decoding accuracy over early learning (Figure 4C) across different decoder window durations (Figure 4 – figure supplement 2).”

Results (lines 310-312):

“Offline contextualization strongly correlated with cumulative micro-offline gains (r = 0.903, R^2^ = 0.816, p < 0.001; Figure 5 – figure supplement 1A, inset) across decoder window durations ranging from 50 to 250ms (Figure 5 – figure supplement 1B, C).“

Discussion (lines 382-385):

“This was further supported by the progressive differentiation of neural representations of the index finger keypress (Figure 4A) and by the robust trial-by trial increase in 2-class decoding accuracy across time windows ranging between 50 and 250ms (Figure 4C; Figure 4 – figure supplement 2).”

Discussion (lines 408-9):

“Offline contextualization consistently correlated with early learning gains across a range of decoding windows (50–250ms; Figure 5 – figure supplement 1).”

Several new control analyses are also provided addressing the question of overlapping keypresses:

**Reviewer #3 (Public review):**
Summary:One goal of this paper is to introduce a new approach for highly accurate decoding of finger movements from human magnetoencephalography data via dimension reduction of a "multi-scale, hybrid" feature space. Following this decoding approach, the authors aim to show that early skill learning involves "contextualization" of the neural coding of individual movements, relative to their position in a sequence of consecutive movements.Furthermore, they aim to show that this "contextualization" develops primarily during short rest periods interspersed with skill training and correlates with a performance metric which the authors interpret as an indicator of offline learning.Strengths:A strength of the paper is the innovative decoding approach, which achieves impressive decoding accuracies via dimension reduction of a "multi-scale, hybrid space". This hybridspace approach follows the neurobiologically plausible idea of concurrent distribution of neural coding across local circuits as well as large-scale networks. A further strength of the study is the large number of tested dimension reduction techniques and classifiers.Weaknesses:A clear weakness of the paper lies in the authors' conclusions regarding "contextualization". Several potential confounds, which partly arise from the experimental design (mainly the use of a single sequence) and which are described below, question the neurobiological implications proposed by the authors and provide a simpler explanation of the results. Furthermore, the paper follows the assumption that short breaks result in offline skill learning, while recent evidence, described below, casts doubt on this assumption.

Please, see below for detailed response to each of these points.

Specifically: The authors interpret the ordinal position information captured by their decoding approach as a reflection of neural coding dedicated to the local context of a movement (Figure 4). One way to dissociate ordinal position information from information about the moving effectors is to train a classifier on one sequence and test the classifier on other sequences that require the same movements, but in different positions (Kornysheva et al., Neuron 2019). In the present study, however, participants trained to repeat a single sequence (4-1-3-2-4).

A crucial difference between our present study and the elegant study from Kornysheva et al. (2019) in Neuron highlighted by the Reviewer is that while ours is a learning study, the Kornysheva et al. study is not. Kornysheva et al. included an initial separate behavioral training session (i.e. – performed outside of the MEG) during which participants learned associations between fractal image patterns and different keypress sequences. Then in a separate, later MEG session—after the stimulus-response associations had been already learned in the first session—participants were tasked with recalling the learned sequences in response to a presented visual cue (i.e. – the paired fractal pattern).

Our rationale for not including multiple sequences in the same Day 1 training session of our study design was that it would lead to prominent interference effects, as widely reported in the literature [10-12]. Thus, while we had to take the issue of interference into consideration for our design, the Kornysheva et al. study did not. While Kornysheva et al. aimed to “dissociate ordinal position information from information about the moving effectors”, we tested various untrained sequences on Day 2 allowing us to determine that the contextualization result was specific to the trained sequence. By using this approach, we avoided interference effects on the learning of the primary skill caused by simultaneous acquisition of a second skill.

The revised manuscript states our findings related to the Day 2 Control data in the following locations:

Results (lines 117-122):

“On the following day, participants were retested on performance of the same sequence (4-1-3-2-4) over 9 trials (*Day 2 Retest*), as well as on the single-trial performance of 9 different untrained control sequences (*Day 2 Controls*: 2-1-3-4-2, 4-2-4-3-1, 3-4-2-3-1, 1-4-3-4-2, 3-2-4-3-1, 1-4-2-3-1, 3-2-4-2-1, 3-2-1-4-2, and 4-23-1-4). As expected, an upward shift in performance of the trained sequence (0.68 ± SD 0.56 keypresses/s; t = 7.21, *p* < 0.001) was observed during Day 2 Retest, indicative of an overnight skill consolidation effect (Figure 1 – figure supplement 1A).”

Results (lines 212-219):

“Utilizing the highest performing decoders that included LDA-based manifold extraction, we assessed the robustness of hybrid-space decoding over multiple sessions by applying it to data collected on the following day during the Day 2 Retest (9-trial retest of the trained sequence) and Day 2 Control (single-trial performance of 9 different untrained sequences) blocks. The decoding accuracy for Day 2 MEG data remained high (87.11% ± SD 8.54% for the trained sequence during Retest, and 79.44% ± SD 5.54% for the untrained Control sequences; Figure 3 – figure supplement 4). Thus, index finger classifiers constructed using the hybrid decoding approach robustly generalized from Day 1 to Day 2 across trained and untrained keypress sequences.”

Results (lines 269-273):

“On Day 2, incorporating contextual information into the hybrid-space decoder enhanced classification accuracy for the trained sequence only (improving from 87.11% for 4-class to 90.22% for 5-class), while performing at or below-chance levels for the Control sequences (≤ 30.22% ± SD 0.44%). Thus, the accuracy improvements resulting from inclusion of contextual information in the decoding framework was specific for the trained skill sequence.”

As a result, ordinal position information is potentially confounded by the fixed finger transitions around each of the two critical positions (first and fifth press). Across consecutive correct sequences, the first keypress in a given sequence was always preceded by a movement of the index finger (=last movement of the preceding sequence), and followed by a little finger movement. The last keypress, on the other hand, was always preceded by a ring finger movement, and followed by an index finger movement (=first movement of the next sequence). Figure 4 - supplement 2 shows that finger identity can be decoded with high accuracy (>70%) across a large time window around the time of the keypress, up to at least +/-100 ms (and likely beyond, given that decoding accuracy is still high at the boundaries of the window depicted in that figure). This time window approaches the keypress transition times in this study. Given that distinct finger transitions characterized the first and fifth keypress, the classifier could thus rely on persistent (or "lingering") information from the preceding finger movement, and/or "preparatory" information about the subsequent finger movement, in order to dissociate the first and fifth keypress.Currently, the manuscript provides little evidence that the context information captured by the decoding approach is more than a by-product of temporally extended, and therefore overlapping, but independent neural representations of consecutive keypresses that are executed in close temporal proximity - rather than a neural representation dedicated to context.During the review process, the authors pointed out that a "mixing" of temporally overlapping information from consecutive keypresses, as described above, should result in systematic misclassifications and therefore be detectable in the confusion matrices in Figures 3C and 4B, which indeed do not provide any evidence that consecutive keypresses are systematically confused. However, such absence of evidence (of systematic misclassification) should be interpreted with caution, and, of course, provides no evidence of absence. The authors also pointed out that such "mixing" would hamper the discriminability of the two ordinal positions of the index finger, given that "ordinal position 5" is systematically followed by "ordinal position 1". This is a valid point which, however, cannot rule out that "contextualization" nevertheless reflects the described "mixing".

The revised manuscript contains several control analyses which rule out this potential confound.

Results (lines 318-328):

“Within-subject correlations were consistent with these group-level findings. The average correlation between offline contextualization and micro-offline gains within individuals was significantly greater than zero (Figure 5 – figure supplement 4, left; t = 3.87, p = 0.00035, df = 25, Cohen's d = 0.76) and stronger than correlations between online contextualization and either micro-online (Figure 5 – figure supplement 4, middle; t = 3.28, p = 0.0015, df = 25, Cohen's d = 1.2) or micro-offline gains (Figure 5 – figure supplement 4, right; t = 3.7021, p = 5.3013e-04, df = 25, Cohen's d = 0.69). These findings were not explained by behavioral changes of typing rhythm (t = -0.03, *p* = 0.976; Figure 5 – figure supplement 5), adjacent keypress transition times (R^2^ = 0.00507, F[1,3202] = 16.3; Figure 5 – figure supplement 6), or overall typing speed (between-subject; R^2^ = 0.028, *p* = 0.41; Figure 5 – figure supplement 7).”

Results (lines 385-390):

“Further, the 5-class classifier—which directly incorporated information about the sequence location context of each keypress into the decoding pipeline—improved decoding accuracy relative to the 4-class classifier (Figure 4C). Importantly, testing on Day 2 revealed specificity of this representational differentiation for the trained skill but not for the same keypresses performed during various unpracticed control sequences (Figure 5C).”

Discussion (lines 408-423):

“Offline contextualization consistently correlated with early learning gains across a range of decoding windows (50–250ms; Figure 5 – figure supplement 1). This result remained unchanged when measuring offline contextualization between the last and second sequence of consecutive trials, inconsistent with a possible confounding effect of pre-planning [30] (Figure 5 – figure supplement 2A). On the other hand, online contextualization did not predict learning (Figure 5 – figure supplement 3). Consistent with these results the average within-subject correlation between offline contextualization and micro-offline gains was significantly stronger than within subject correlations between online contextualization and either micro-online or micro-offline gains (Figure 5 – figure supplement 4).

Offline contextualization was not driven by trial-by-trial behavioral differences, including typing rhythm (Figure 5 – figure supplement 5) and adjacent keypress transition times (Figure 5 – figure supplement 6) nor by between-subject differences in overall typing speed (Figure 5 – figure supplement 7)—ruling out a reliance on differences in the temporal overlap of keypresses. Importantly, offline contextualization documented on Day 1 stabilized once a performance plateau was reached (trials 11-36), and was retained on Day 2, documenting overnight consolidation of the differentiated neural representations.”

During the review process, the authors responded to my concern that training of a single sequence introduces the potential confound of "mixing" described above, which could have been avoided by training on several sequences, as in Kornysheva et al. (Neuron 2019), by arguing that Day 2 in their study did include control sequences. However, the authors' findings regarding these control sequences are fundamentally different from the findings in Kornysheva et al. (2019), and do not provide any indication of effector-independent ordinal information in the described contextualization - but, actually, the contrary. In Kornysheva et al. (Neuron 2019), ordinal, or positional, information refers purely to the rank of a movement in a sequence. In line with the idea of competitive queuing, Kornysheva et al. (2019) have shown that humans prepare for a motor sequence via a simultaneous representation of several of the upcoming movements, weighted by their rank in the sequence. Importantly, they could show that this gradient carries information that is largely devoid of information about the order of specific effectors involved in a sequence, or their timing, in line with competitive queuing. They showed this by training a classifier to discriminate between the five consecutive movements that constituted one specific sequence of finger movements (five classes: 1st, 2nd, 3rd, 4th, 5th movement in the sequence) and then testing whether that classifier could identify the rank (1st, 2nd, 3rd, etc) of movements in another sequence, in which the fingers moved in a different order, and with different timings. Importantly, this approach demonstrated that the graded representations observed during preparation were largely maintained after this cross decoding, indicating that the sequence was represented via ordinal position information that was largely devoid of information about the specific effectors or timings involved in sequence execution. This result differs completely from the findings in the current manuscript. Dash et al. report a drop in detected ordinal position information (degree of contextualization in figure 5C) when testing for contextualization in their novel, untrained sequences on Day 2, indicating that context and ordinal information as defined in Dash et al. is not at all devoid of information about the specific effectors involved in a sequence. In this regard, a main concern in my public review, as well as the second reviewer's public review, is that Dash et al. cannot tell apart, by design, whether there is truly contextualization in the neural representation of a sequence (which they claim), or whether their results regarding "contextualization" are explained by what they call "mixing" in their author response, i.e., an overlap of representations of consecutive movements, as suggested as an alternative explanation by Reviewer 2 and myself.

Again, as stated in response to a related comment by the Reviewer above, it is not surprising that our results differ from the study by Kornysheva et al. (2019) . A crucial difference between the studies that the Reviewer fails to recognize is that while ours is a learning study, the Kornysheva et al. study is not. Our rationale for not including multiple sequences in the same Day 1 training session of our study design was that it would lead to prominent interference effects, as widely reported in the literature [10-12]. Thus, while we had to take the issue of interference into consideration for our design, the Kornysheva et al. study did not, since it was not concerned with learning dynamics. The strengths of the elegant Kornysheva study highlighted by the Reviewer—that the pre-planned sequence queuing gradient of sequence actions was independent of the effectors or timings used—is precisely due to the fact that participants were selecting between sequence options that had been previously—and equivalently—learned. The decoders in the Kornynsheva study were trained to classify effector- and timing-independent sequence position information— by design—so it is not surprising that this is the information they reflect.

The questions asked in our study were different: (1) Do the neural representations of the same sequence action executed in different skill (ordinal sequence) locations differentiate (contextualize) during early learning? and (2) Is the observed contextualization specific to the learned sequence? Thus, while Kornysheva et al. aimed to “dissociate ordinal position information from information about the moving effectors”, we tested various untrained sequences on Day 2 allowing us to determine that the contextualization result was specific to the trained sequence. By using this approach, we avoided interference effects on the learning of the primary skill caused by simultaneous acquisition of a second skill.

Such temporal overlap of consecutive, independent finger representations may also account for the dynamics of "ordinal coding"/"contextualization", i.e., the increase in 2class decoding accuracy, across Day 1 (Figure 4C). As learning progresses, both tapping speed and the consistency of keypress transition times increase (Figure 1), i.e., consecutive keypresses are closer in time, and more consistently so. As a result, information related to a given keypress is increasingly overlapping in time with information related to the preceding and subsequent keypresses. The authors seem to argue that their regression analysis in Figure 5 - figure supplement 3 speaks against any influence of tapping speed on "ordinal coding" (even though that argument is not made explicitly in the manuscript). However, Figure 5 - figure supplement 3 shows inter-individual differences in a between-subject analysis (across trials, as in panel A, or separately for each trial, as in panel B), and, therefore, says little about the within-subject dynamics of "ordinal coding" across the experiment. A regression of trial-by-trial "ordinal coding" on trial-by-trial tapping speed (either within-subject, or at a group-level, after averaging across subjects) could address this issue. Given the highly similar dynamics of "ordinal coding" on the one hand (Figure 4C), and tapping speed on the other hand (Figure 1B), I would expect a strong relationship between the two in the suggested within-subject (or group-level) regression.

The aim of the between-subject regression analysis presented in the Results (see below) and in Figure 5—figure supplement 7 (previously Figure 5—figure supplement 3) of the revised manuscript, was to rule out a general effect of tapping speed on the magnitude of contextualization observed. If temporal overlap of neural representations was driving their differentiation, then participants typing at higher speeds should also show greater contextualization scores. We made the decision to use a between-subject analysis to address this issue since within-subject skill speed variance was rather small over most of the training session.

The Reviewer’s request that we additionally carry-out a “regression of trial-by-trial "ordinal coding" on trial-by-trial tapping speed (either within-subject, or at a group-level, after averaging across subjects)” is essentially the same request of Reviewer 2 above. That request was to perform a modified simple linear regression analysis where the predictor is the sum the 4-4 and 4-1 transition times, since these transitions are where any temporal overlaps of neural representations would occur. A new Figure 5 – figure supplement 6 in the revised manuscript includes a scatter plot showing the sum of adjacent index finger keypress transition times (i.e. – the 4-4 transition at the conclusion of one sequence iteration and the 4-1 transition at the beginning of the next sequence iteration) versus online contextualization distances measured during practice trials. Both the keypress transition times and online contextualization scores were z-score normalized within individual subjects, and then concatenated into a single data superset. As is clear in the figure data, results of the regression analysis showed a very weak linear relationship between the two (R^2^ = 0.00507, F[1,3202] = 16.3). Thus, contextualization score magnitudes do not reflect the amount of overlap between adjacent keypresses when assessed either within- or between-subject.

The revised manuscript now states:

Results (lines 318-328):

“Within-subject correlations were consistent with these group-level findings. The average correlation between offline contextualization and micro-offline gains within individuals was significantly greater than zero (Figure 5 – figure supplement 4, left; t = 3.87, p = 0.00035, df = 25, Cohen's d = 0.76) and stronger than correlations between online contextualization and either micro-online (Figure 5 – figure supplement 4, middle; t = 3.28, p = 0.0015, df = 25, Cohen's d = 1.2) or micro-offline gains (Figure 5 – figure supplement 4, right; t = 3.7021, p = 5.3013e-04, df = 25, Cohen's d = 0.69). These findings were not explained by behavioral changes of typing rhythm (t = -0.03, *p* = 0.976; Figure 5 – figure supplement 5), adjacent keypress transition times (R^2^ = 0.00507, F[1,3202] = 16.3; Figure 5 – figure supplement 6), or overall typing speed (between-subject; R^2^ = 0.028, *p* = 0.41; Figure 5 – figure supplement 7).”

Furthermore, learning should increase the number of (consecutively) correct sequences, and, thus, the consistency of finger transitions. Therefore, the increase in 2-class decoding accuracy may simply reflect an increasing overlap in time of increasingly consistent information from consecutive keypresses, which allows the classifier to dissociate the first and fifth keypress more reliably as learning progresses, simply based on the characteristic finger transitions associated with each. In other words, given that the physical context of a given keypress changes as learning progresses - keypresses move closer together in time and are more consistently correct - it seems problematic to conclude that the mental representation of that context changes. To draw that conclusion, the physical context should remain stable (or any changes to the physical context should be controlled for).

The revised manuscript now addresses specifically the question of mixing of temporally overlapping information:

Results (Lines 310-328)

“Offline contextualization strongly correlated with cumulative micro-offline gains (r = 0.903, R^2^ = 0.816, p < 0.001; Figure 5 – figure supplement 1A, inset) across decoder window durations ranging from 50 to 250ms (Figure 5 – figure supplement 1B, C). The offline contextualization between the final sequence of each trial and the second sequence of the subsequent trial (excluding the first sequence) yielded comparable results. This indicates that pre-planning at the start of each practice trial did not directly influence the offline contextualization measure [30] (Figure 5 – figure supplement 2A, *1st vs. 2nd Sequence approaches*). Conversely, online contextualization (using either measurement approach) did not explain early online learning gains (i.e. – Figure 5 – figure supplement 3). Within-subject correlations were consistent with these group-level findings. The average correlation between offline contextualization and micro-offline gains within individuals was significantly greater than zero (Figure 5 – figure supplement 4, left; t = 3.87, p = 0.00035, df = 25, Cohen's d = 0.76) and stronger than correlations between online contextualization and either micro-online (Figure 5 – figure supplement 4, middle; t = 3.28, p = 0.0015, df = 25, Cohen's d = 1.2) or micro-offline gains (Figure 5 – figure supplement 4, right; t = 3.7021, p = 5.3013e-04, df = 25, Cohen's d = 0.69). These findings were not explained by behavioral changes of typing rhythm (t = -0.03, *p* = 0.976; Figure 5 – figure supplement 5), adjacent keypress transition times (R^2^ = 0.00507, F[1,3202] = 16.3; Figure 5 – figure supplement 6), or overall typing speed (between-subject; R^2^ = 0.028, *p* = 0.41; Figure 5 – figure supplement 7). “

Discussion (Lines 417-423)

“Offline contextualization was not driven by trial-by-trial behavioral differences, including typing rhythm (Figure 5 – figure supplement 5) and adjacent keypress transition times (Figure 5 – figure supplement 6) nor by between-subject differences in overall typing speed (Figure 5 – figure supplement 7)—ruling out a reliance on differences in the temporal overlap of keypresses. Importantly, offline contextualization documented on Day 1 stabilized once a performance plateau was reached (trials 11-36), and was retained on Day 2, documenting overnight consolidation of the differentiated neural representations.”

A similar difference in physical context may explain why neural representation distances ("differentiation") differ between rest and practice (Figure 5). The authors define "offline differentiation" by comparing the hybrid space features of the last index finger movement of a trial (ordinal position 5) and the first index finger movement of the next trial (ordinal position 1). However, the latter is not only the first movement in the sequence but also the very first movement in that trial (at least in trials that started with a correct sequence), i.e., not preceded by any recent movement. In contrast, the last index finger of the last correct sequence in the preceding trial includes the characteristic finger transition from the fourth to the fifth movement. Thus, there is more overlapping information arising from the consistent, neighbouring keypresses for the last index finger movement, compared to the first index finger movement of the next trial. A strong difference (larger neural representation distance) between these two movements is, therefore, not surprising, given the task design, and this difference is also expected to increase with learning, given the increase in tapping speed, and the consequent stronger overlap in representations for consecutive keypresses. Furthermore, initiating a new sequence involves pre-planning, while ongoing practice relies on online planning (Ariani et al., eNeuro 2021), i.e., two mental operations that are dissociable at the level of neural representation (Ariani et al., bioRxiv 2023).

The revised manuscript now addresses specifically the question of pre-planning:

Results (lines 310-318):

“Offline contextualization strongly correlated with cumulative micro-offline gains (r = 0.903, R^2^ = 0.816, p < 0.001; Figure 5 – figure supplement 1A, inset) across decoder window durations ranging from 50 to 250ms (Figure 5 – figure supplement 1B, C). The offline contextualization between the final sequence of each trial and the second sequence of the subsequent trial (excluding the first sequence) yielded comparable results. This indicates that pre-planning at the start of each practice trial did not directly influence the offline contextualization measure [30] (Figure 5 – figure supplement 2A, *1st vs. 2nd Sequence approaches*). Conversely, online contextualization (using either measurement approach) did not explain early online learning gains (i.e. – Figure 5 – figure supplement 3).”

Discussion (lines 408-416):

“Offline contextualization consistently correlated with early learning gains across a range of decoding windows (50–250ms; Figure 5 – figure supplement 1). This result remained unchanged when measuring offline contextualization between the last and second sequence of consecutive trials, inconsistent with a possible confounding effect of pre-planning [30] (Figure 5 – figure supplement 2A). On the other hand, online contextualization did not predict learning (Figure 5 – figure supplement 3). Consistent with these results the average within-subject correlation between offline contextualization and micro-offline gains was significantly stronger than within-subject correlations between online contextualization and either micro-online or micro-offline gains (Figure 5 – figure supplement 4).”

A further complication in interpreting the results stems from the visual feedback that participants received during the task. Each keypress generated an asterisk shown above the string on the screen. It is not clear why the authors introduced this complicating visual feedback in their task, besides consistency with their previous studies. The resulting systematic link between the pattern of visual stimulation (the number of asterisks on the screen) and the ordinal position of a keypress makes the interpretation of "contextual information" that differentiates between ordinal positions difficult. During the review process, the authors reported a confusion matrix from a classification of asterisks position based on eye tracking data recorded during the task and concluded that the classifier performed at chance level and gaze was, thus, apparently not biased by the visual stimulation. However, the confusion matrix showed a huge bias that was difficult to interpret (a very strong tendency to predict one of the five asterisk positions, despite chance-level performance). Without including additional information for this analysis (or simply the gaze position as a function of the number of astersisk on the screen) in the manuscript, this important control analysis cannot be properly assessed, and is not available to the public.

We now include the gaze position data requested by the Reviewer alongside the confusion matrix results in Figure 4 – figure supplement 3.

Results (lines 207-211):

“An alternate decoder trained on ICA components labeled as movement or physiological artefacts (e.g. – head movement, ECG, eye movements and blinks; Figure 3 – figure supplement 3A, D) and removed from the original input feature set during the pre-processing stage approached chance-level performance (Figure 4 – figure supplement 3), indicating that the 4-class hybrid decoder results were not driven by task-related artefacts.” Results (lines 261-268):

“As expected, the 5-class hybrid-space decoder performance approached chance levels when tested with randomly shuffled keypress labels (18.41%± SD 7.4% for Day 1 data; Figure 4 – figure supplement 3C). Task-related eye movements did not explain these results since an alternate 5-class hybrid decoder constructed from three eye movement features (gaze position at the KeyDown event, gaze position 200ms later, and peak eye movement velocity within this window; Figure 4 – figure supplement 3A) performed at chance levels (cross-validated test accuracy = 0.2181; Figure 4 – figure supplement 3B, C). “

Discussion (Lines 362-368):

“Task-related movements—which also express in lower frequency ranges—did not explain these results given the near chance-level performance of alternative decoders trained on (a) artefact-related ICA components removed during MEG preprocessing (Figure 3 – figure supplement 3A-C) and on (b) task-related eye movement features (Figure 4 – figure supplement 3B, C). This explanation is also inconsistent with the minimal average head motion of 1.159 mm (± 1.077 SD) across the MEG recording (Figure 3 – figure supplement 3D).”

The rationale for the task design including the asterisks is presented below:

Methods (Lines 500-514)

“The five-item sequence was displayed on the computer screen for the duration of each practice round and participants were directed to fix their gaze on the sequence. Small asterisks were displayed above a sequence item after each successive keypress, signaling the participants' present position within the sequence. Inclusion of this feedback minimizes working memory loads during task performance [73]. Following the completion of a full sequence iteration, the asterisk returned to the first sequence item. The asterisk did not provide error feedback as it appeared for both correct and incorrect keypresses. At the end of each practice round, the displayed number sequence was replaced by a string of five "X" symbols displayed on the computer screen, which remained for the duration of the rest break. Participants were instructed to focus their gaze on the screen during this time. The behavior in this explicit, motor learning task consists of generative action sequences rather than sequences of stimulus-induced responses as in the serial reaction time task (SRTT). A similar real-world example would be manually inputting a long password into a secure online application in which one intrinsically generates the sequence from memory and receives similar feedback about the password sequence position (also provided as asterisks), which is typically ignored by the user.”

The authors report a significant correlation between "offline differentiation" and cumulative micro-offline gains. However, this does not address the question whether there is a trial-by-trial relation between the degree of "contextualization" and the amount of micro-offline gains - i.e., the question whether performance changes (micro-offline gains) are less pronounced across rest periods for which the change in "contextualization" is relatively low. The single-subject correlation between contextualization changes "during" rest and micro-offline gains (Figure 5 - figure supplement 4) addresses this question, however, the critical statistical test (are correlation coefficients significantly different from zero) is not included. Given the displayed distribution, it seems unlikely that correlation coefficients are significantly above zero.

As recommend by the Reviewer, we now include one-way right-tailed t-test results which provide further support to the previously reported finding. The mean of within-subject correlations between offline contextualization and cumulative micro-offline gains was significantly greater than zero (t = 3.87, p = 0.00035, df = 25, Cohen's d = 0.76; see Figure 5 – figure supplement 4, left), while correlations for online contextualization versus cumulative micro-online (t = -1.14, p = 0.8669, df = 25, Cohen's d = -0.22) or micro-offline gains t = -0.097, p = 0.5384, df = 25, Cohen's d = -0.019 were not. We have incorporated the significant one-way t-test for offline contextualization and cumulative micro-offline gains in the Results section of the revised manuscript (lines 313-318) and the Figure 5 – figure supplement 4 legend.

The authors follow the assumption that micro-offline gains reflect offline learning.However, there is no compelling evidence in the literature, and no evidence in the present manuscript, that micro-offline gains (during any training phase) reflect offline learning. Instead, emerging evidence in the literature indicates that they do not (Das et al., bioRxiv 2024), and instead reflect transient performance benefits when participants train with breaks, compared to participants who train without breaks, however, these benefits vanish within seconds after training if both groups of participants perform under comparable conditions (Das et al., bioRxiv 2024). During the review process, the authors argued that differences in the design between Das et al. (2024) on the one hand (Experiments 1 and 2), and the study by Bönstrup et al. (2019) on the other hand, may have prevented Das et al. (2024) from finding the assumed (lasting) learning benefit by micro-offline consolidation. However, the Supplementary Material of Das et al. (2024) includes an experiment (Experiment S1) whose design closely follows the early learning phase of Bönstrup et al. (2019), and which, nevertheless, demonstrates that there is no lasting benefit of taking breaks for the acquired skill level, despite the presence of micro-offline gains.

We thank the Reviewer for alerting us to this new data added to the revised supplementary materials of Das et al. (2024) posted to bioRxiv. However, despite the Reviewer’s claim to the contrary, a careful comparison between the Das et al and Bönstrup et al studies reveal more substantive differences than similarities and does not “closely follows a large proportion of the early learning phase of Bönstrup et al. (2019)” as stated.

In the Das et al. Experiment S1, sixty-two participants were randomly assigned to “with breaks” or “no breaks” skill training groups. The “with breaks” group alternated 10 seconds of skill sequence practice with 10 seconds of rest over seven trials (2 min and 2 sec total training duration). This amounts to 66.7% of the early learning period defined by Bönstrup et al. (2019) (i.e. - eleven 10-second-long practice periods interleaved with ten 10-second-long rest breaks; 3 min 30 sec total training duration).

Also, please note that while no performance feedback nor reward was given in the Bönstrup et al. (2019) study, participants in the Das et al. study received explicit performance-based monetary rewards, a potentially crucial driver of differentiated behavior between the two studies:

“Participants were incentivized with bonus money based on the total number of correct sequences completed throughout the experiment.”

The “no breaks” group in the Das et al. study practiced the skill sequence for 70 continuous seconds. Both groups (despite one being labeled “no breaks”) follow training with a long 3-minute break (also note that since the “with breaks” group ends with 10 seconds of rest their break is actually longer), before finishing with a skill “*test*” over a continuous 50-second-long block. During the 70 seconds of training, the “with breaks” group shows more learning than the “no breaks” group. Interestingly, following the long 3minute break the “with breaks” group display a performance drop (relative to their performance at the end of training) that is stable over the full 50-second test, while the “no breaks” group shows an immediate performance improvement following the long break that continues to increase over the 50-second test.

Separately, there are important issues regarding the Das et al. study that should be considered through the lens of recent findings not referred to in the preprint. A major element of their experimental design is that both groups—“with breaks” and “no breaks”— actually receive quite a long 3-minute break just before the skill test. This long break is more than 2.5x the cumulative interleaved rest experienced by the “with breaks” group. Thus, although the design is intended to contrast the presence or absence of rest “breaks”, that difference between groups is no longer maintained at the point of the skill test.

The Das et al. results are most consistent with an alternative interpretation of the data— that the “no breaks” group experiences offline learning during their long 3-minute break. This is supported by the recent work of Griffin et al. (2025) where micro-array recordings from primary and premotor cortex were obtained from macaque monkeys while they performed blocks of ten continuous reaching sequences up to 81.4 seconds in duration (see source data for Extended Data Figure 1h) with 90 seconds of interleaved rest. Griffin et al. observed offline improvement in skill immediately following the rest break that was causally related to neural reactivations (i.e. – neural replay) that occurred *during* the rest break. Importantly, the highest density of reactivations was present in the very first 90second break between Blocks 1 and 2 (see Fig. 2f in Griffin et al., 2025). This supports the interpretation that both the “with breaks” and “no breaks” group express offline learning gains, with these gains being delayed in the “no breaks” group due to the practice schedule.

On the other hand, if offline learning can occur during this longer break, then why would the “with breaks” group show no benefit? Again, it could be that most of the offline gains for this group were front-loaded during the seven shorter 10-second rest breaks. Another possible, though not mutually exclusive, explanation is that the observed drop in performance in the “with breaks” group is driven by contextual interference. Specifically, similar to Experiments 1 and 2 in Das et al. (2024), the skill test is conducted under very different conditions than those which the “with breaks” group practiced the skill under (short bursts of practiced alternating with equally short breaks). On the other hand, the “no breaks” group is tested (50 seconds of continuous practice) under quite similar conditions to their training schedule (70 seconds of continuous practice). Thus, it is possible that this dissimilarity between training and test could lead to reduced performance in the “with breaks” group.

We made the following manuscript revisions related to these important issues:

Introduction (Lines 26-56)

“Practicing a new motor skill elicits rapid performance improvements (early learning) [1] that precede skill performance plateaus [5]. Skill gains during early learning accumulate over rest periods (micro-offline) interspersed with practice [1, 6-10], and are up to four times larger than offline performance improvements reported following overnight sleep [1]. During this initial interval of prominent learning, retroactive interference immediately following each practice interval reduces learning rates relative to interference after passage of time, consistent with stabilization of the motor memory [11]. Micro-offline gains observed during early learning are reproducible [7, 10-13] and are similar in magnitude even when practice periods are reduced by half to 5 seconds in length, thereby confirming that they are not merely a result of recovery from performance fatigue [11]. Additionally, they are unaffected by the random termination of practice periods, which eliminates the possibility of predictive motor slowing as a contributing factor [11]. Collectively, these behavioral findings point towards the interpretation that micro offline gains during early learning represent a form of memory consolidation [1].

This interpretation has been further supported by brain imaging and electrophysiological studies linking known memory-related networks and consolidation mechanisms to rapid offline performance improvements. In humans, the rate of hippocampo-neocortical neural replay predicts micro-offline gains [6]. Consistent with these findings, Chen et al. [12] and Sjøgård et al. [13] furnished direct evidence from intracranial human EEG studies, demonstrating a connection between the density of hippocampal sharp-wave ripples (80-120 Hz)—recognized markers of neural replay—and micro-offline gains during early learning. Further, Griffin et al. reported that neural replay of task-related ensembles in the motor cortex of macaques during brief rest periods— akin to those observed in humans [1, 6-8, 14]—are not merely correlated with, but are causal drivers of micro-offline learning [15]. Specifically, the same reach directions that were replayed the most during rest breaks showed the greatest reduction in path length (i.e. – more efficient movement path between two locations in the reach sequence) during subsequent trials, while stimulation applied during rest intervals preceding performance plateau reduced reactivation rates and virtually abolished micro-offline gains [15]. Thus, converging evidence in humans and non-human primates across indirect non-invasive and direct invasive recording techniques link hippocampal activity, neural replay dynamics and offline skill gains in early motor learning that precede performance plateau.”

Next, in the Methods, we articulate important constrains formulated by Pan and Rickard and Bonstrup et al for meaningful measurements:

Methods (Lines 493-499)

“The study design followed specific recommendations by Pan and Rickard (2015): (1) utilizing 10-second practice trials and (2) constraining analysis of micro-offline gains to early learning trials (where performance monotonically increases and 95% of overall performance gains occur) that precede the emergence of “scalloped” performance dynamics strongly linked to reactive inhibition effects ([29, 72]). This is precisely the portion of the learning curve Pan and Rickard referred to when they stated “…rapid learning during that period masks any reactive inhibition effect” [29].”

We finally discuss the implications of neglecting some or all of these recommendations:

Discussion (Lines 444-452):

“Finally, caution should be exercised when extrapolating findings during early skill learning, a period of steep performance improvements, to findings reported after insufficient practice [67], post-plateau performance periods [68], or non-learning situations (e.g. performance of non-repeating keypress sequences in [67]) when reactive inhibition or contextual interference effects are prominent. Ultimately, it will be important to develop new paradigms allowing one to independently estimate the different coincident or antagonistic features (e.g. - memory consolidation, planning, working memory and reactive inhibition) contributing to micro-online and micro-offline gains during and after early skill learning within a unifying framework.”

Along these lines, the authors' claim, based on Bönstrup et al. 2020, that "retroactive interference immediately following practice periods reduces micro-offline learning", is not supported by that very reference. Citing Bönstrup et al. (2020), "Regarding early learning dynamics (trials 1-5), we found no differences in microscale learning parameters (micro online/offline) or total early learning between both interference groups." That is, contrary to Dash et al.'s current claim, Bönstrup et al. (2020) did not find any retroactive interference effect on the specific behavioral readout (micro-offline gains) that the authors assume to reflect consolidation.

Please, note that the Bönstrup et al. 2020 paper abstract states:

“Third, retroactive interference immediately after each practice period reduced the learning rate relative to interference after passage of time (N = 373), indicating stabilization of the motor memory at a microscale of several seconds.”

which is further supported by this statement in the Results:

“The model comprised three parameters representing the initial performance, maximum performance and learning rate (see Eq. 1, “Methods”, “Data Analysis” section). We then statistically compared the model parameters between the interference groups (Fig. 2d). The late interference group showed a higher learning rate compared with the early interference group (late: 0.26 ± 0.23, early: 2.15 ± 0.20, P=0.04). The effect size of the group difference was small to medium (Cohen’s d 0.15)[29]. Similar differences with a stronger rise in the learning curve of a late interference groups vs. an early interference group were found in a smaller sample collected in the lab environment (Supplementary Fig. 3).”

We have modified the statement in the revised manuscript to specify that the difference observed was between learning rates: Introduction (Lines 30-32)

“During this initial interval of prominent learning, retroactive interference immediately following each practice interval reduces learning rates relative to interference after passage of time, consistent with stabilization of the motor memory [11].”

The authors conclude that performance improves, and representation manifolds differentiate, "during" rest periods (see, e.g., abstract). However, micro-offline gains (as well as offline contextualization) are computed from data obtained during practice, not rest, and may, thus, just as well reflect a change that occurs "online", e.g., at the very onset of practice (like pre-planning) or throughout practice (like fatigue, or reactive inhibition).

The Reviewer raises again the issue of a potential confound of “pre-planning” on our contextualization measures as in the comment above:

“Furthermore, initiating a new sequence involves pre-planning, while ongoing practice relies on online planning (Ariani et al., eNeuro 2021), i.e., two mental operations that are dissociable at the level of neural representation (Ariani et al., bioRxiv 2023).”

The cited studies by Ariani et al. indicate that effects of pre-planning are likely to impact the first 3 keypresses of the initial sequence iteration in each trial. As stated in the response to this comment above, we conducted a control analysis of contextualization that ignores the first sequence iteration in each trial to partial out any potential preplanning effect. This control analyses yielded comparable results, indicating that preplanning is not a major driver of our reported contextualization effects. We now report this in the revised manuscript:

We also state in the Figure 1 legend (Lines 99-103) in the revised manuscript that preplanning has no effect on the behavioral measures of micro-offline and micro-online gains in our dataset:

The Reviewer also raises the issue of possible effects stemming from “fatigue” and “reactive inhibition” which inhibit performance and are indeed relevant to skill learning studies. We designed our task to specifically mitigate these effects. We now more clearly articulate this rationale in the description of the task design as well as the measurement constraints essential for minimizing their impact.

We also discuss the implications of fatigue and reactive inhibition effects in experimental designs that neglect to follow these recommendations formulated by Pan and Rickard in the Discussion section and propose how this issue can be better addressed in future investigations.

To summarize, the results of our study indicate that: (a) offline contextualization effects are not explained by pre-planning of the first action sequence iteration in each practice trial; and (b) the task design implemented in this study purposefully minimize any possible effects of reactive inhibition or fatigue. Circling back to the Reviewer’s proposal that “contextualization…may just as well reflect a change that occurs "online"”, we show in this paper direct empirical evidence that contextualization develops to a greater extent across rest periods rather than across practice trials, contrary to the Reviewer’s proposal.

That is, the definition of micro-offline gains (as well as offline contextualization) conflates online and "offline" processes. This becomes strikingly clear in the recent Nature paper by Griffin et al. (2025), who computed micro-offline gains as the difference in average performance across the first five sequences in a practice period (a block, in their terminology) and the last five sequences in the previous practice period. Averaging across sequences in this way minimises the chance to detect online performance changes and inflates changes in performance "offline". The problem that "online" gains (or contextualization) is actually computed from data entirely generated online, and therefore subject to processes that occur online, is inherent in the very definition of micro-online gains, whether, or not, they computed from averaged performance.

We would like to make it clear that the issue raised by the Reviewer with respect to averaging across sequences done in the Griffin et al. (2025) study does not impact our study in any way. The primary skill measure used in all analyses reported in our paper is not temporally averaged. We estimated instantaneous correct sequence speed over the entire trial. Once the first sequence iteration within a trial is completed, the speed estimate is then updated at the resolution of individual keypresses. All micro-online and -offline behavioral changes are measured as the difference in instantaneous speed at the beginning and end of individual practice trials.

Methods (lines 528-530):

“The instantaneous correct sequence speed was calculated as the inverse of the average KTT across a single correct sequence iteration and was updated for each correct keypress.”

The instantaneous speed measure used in our analyses, in fact, maximizes the likelihood of detecting changes in online performance, as the Reviewer indicates. Despite this optimally sensitive measurement of online changes, our findings remained robust, consistently converging on the same outcome across our original analyses and the multiple controls recommended by the reviewers. Notably, online contextualization changes are significantly weaker than offline contextualization in all comparisons with different measurement approaches.

Results (lines 302-309)

“The Euclidian distance between neural representations of Index_OP1_ (i.e. - index finger keypress at ordinal position 1 of the sequence) and Index_OP5_ (i.e. - index finger keypress at ordinal position 5 of the sequence) increased progressively during early learning (Figure 5A)—predominantly during rest intervals (offline contextualization) rather than during practice (online) (t = 4.84, p < 0.001, df = 25, Cohen's d = 1.2; Figure 5B; Figure 5 – figure supplement 1A). An alternative online contextualization determination equalling the time interval between online and offline comparisons (*Trial-based;* 10 seconds between Index_OP1_ and Index_OP5_ observations in both cases) rendered a similar result (Figure 5 – figure supplement 2B).

Results (lines 316-318)

“Conversely, online contextualization (using either measurement approach) did not explain early online learning gains (i.e. – Figure 5 – figure supplement 3).”

Results (lines 318-328)

“Within-subject correlations were consistent with these group-level findings. The average correlation between offline contextualization and micro-offline gains within individuals was significantly greater than zero (Figure 5 – figure supplement 4, left; t = 3.87, p = 0.00035, df = 25, Cohen's d = 0.76) and stronger than correlations between online contextualization and either micro-online (Figure 5 – figure supplement 4, middle; t = 3.28, p = 0.0015, df = 25, Cohen's d = 1.2) or microoffline gains (Figure 5 – figure supplement 4, right; t = 3.7021, p = 5.3013e-04, df = 25, Cohen's d = 0.69). These findings were not explained by behavioral changes of typing rhythm (t = -0.03, *p* = 0.976; Figure 5 – figure supplement 5), adjacent keypress transition times (R^2^ = 0.00507, F[1,3202] = 16.3; Figure 5 – figure supplement 6), or overall typing speed (between-subject; R^2^ = 0.028, *p* = 0.41; Figure 5 – figure supplement 7).”

We disagree with the Reviewer’s statement that “the definition of micro-offline gains (as well as offline contextualization) conflates online and "offline" processes”. From a strictly behavioral point of view, it is obviously true that one can only measure skill (rather than the absence of it during rest) to determine how it changes over time. While skill changes surrounding rest are used to infer offline learning processes, recovery of skill decay following intense practice is used to infer “unmeasurable” recovery from fatigue or reactive inhibition. In other words, the alternative processes proposed by the Reviewer also rely on the same inferential reasoning.

Importantly, inferences can be validated through the identification of mechanisms. Our experiment constrained the study to evaluation of changes in neural representations of the same action in different contexts, while minimized the impact of mechanisms related to fatigue/reactive inhibition [13, 14]. In this way, we observed that behavioral gains and neural contextualization occurs to a greater extent over rest breaks rather than during practice trials and that offline contextualization changes strongly correlate with the offline behavioral gains, while online contextualization does not. This result was supported by the results of all control analyses recommended by the Reviewers. Specifically:

Methods (Lines 493-499)

“The study design followed specific recommendations by Pan and Rickard (2015): (1) utilizing 10-second practice trials and (2) constraining analysis of micro-offline gains to early learning trials (where performance monotonically increases and 95% of overall performance gains occur) that precede the emergence of “scalloped” performance dynamics strongly linked to reactive inhibition effects ([29, 72]). This is precisely the portion of the learning curve Pan and Rickard referred to when they stated “…rapid learning during that period masks any reactive inhibition effect” [29]*.”*

And Discussion (Lines 444-448):

“Finally, caution should be exercised when extrapolating findings during early skill learning, a period of steep performance improvements, to findings reported after insufficient practice [67], post-plateau performance periods [68], or non-learning situations (e.g. performance of non-repeating keypress sequences in [67]) when reactive inhibition or contextual interference effects are prominent.”

Next, we show that offline contextualization is greater than online contextualization and predicts offline behavioral gains across all measurement approaches, including all controls suggested by the Reviewer’s comments and recommendations.

Results (lines 302-318):

“The Euclidian distance between neural representations of Index_OP1_ (i.e. - index finger keypress at ordinal position 1 of the sequence) and Index_OP5_ (i.e. - index finger keypress at ordinal position 5 of the sequence) increased progressively during early learning (Figure 5A)—predominantly during rest intervals (*offline contextualization*) rather than during practice (*online*) (t = 4.84, p < 0.001, df = 25, Cohen's d = 1.2; Figure 5B; Figure 5 – figure supplement 1A). An alternative online contextualization determination equalling the time interval between online and offline comparisons (*Trial-based;* 10 seconds between Index_OP1_ and Index_OP5_ observations in both cases) rendered a similar result (Figure 5 – figure supplement 2B).

Offline contextualization strongly correlated with cumulative micro-offline gains (r = 0.903, R^2^ = 0.816, p < 0.001; Figure 5 – figure supplement 1A, inset) across decoder window durations ranging from 50 to 250ms (Figure 5 – figure supplement 1B, C). The offline contextualization between the final sequence of each trial and the second sequence of the subsequent trial (excluding the first sequence) yielded comparable results. This indicates that pre-planning at the start of each practice trial did not directly influence the offline contextualization measure [30] (Figure 5 – figure supplement 2A, *1st vs. 2nd Sequence approaches*). Conversely, online contextualization (using either measurement approach) did not explain early online learning gains (i.e. – Figure 5 – figure supplement 3).”

Results (lines 318-324)

“Within-subject correlations were consistent with these group-level findings. The average correlation between offline contextualization and micro-offline gains within individuals was significantly greater than zero (Figure 5 – figure supplement 4, left; t = 3.87, p = 0.00035, df = 25, Cohen's d = 0.76) and stronger than correlations between online contextualization and either micro-online (Figure 5 – figure supplement 4, middle; t = 3.28, p = 0.0015, df = 25, Cohen's d = 1.2) or microoffline gains (Figure 5 – figure supplement 4, right; t = 3.7021, p = 5.3013e-04, df = 25, Cohen's d = 0.69).”

Discussion (lines 408-416):

“Offline contextualization consistently correlated with early learning gains across a range of decoding windows (50–250ms; Figure 5 – figure supplement 1). This result remained unchanged when measuring offline contextualization between the last and second sequence of consecutive trials, inconsistent with a possible confounding effect of pre-planning [30] (Figure 5 – figure supplement 2A). On the other hand, online contextualization did not predict learning (Figure 5 – figure supplement 3). Consistent with these results the average within-subject correlation between offline contextualization and micro-offline gains was significantly stronger than within subject correlations between online contextualization and either micro-online or micro-offline gains (Figure 5 – figure supplement 4).”

We then show that offline contextualization is not explained by pre-planning of the first action sequence:

Results (lines 310-316):

“Offline contextualization strongly correlated with cumulative micro-offline gains (r = 0.903, R^2^ = 0.816, p < 0.001; Figure 5 – figure supplement 1A, inset) across decoder window durations ranging from 50 to 250ms (Figure 5 – figure supplement 1B, C). The offline contextualization between the final sequence of each trial and the second sequence of the subsequent trial (excluding the first sequence) yielded comparable results. This indicates that pre-planning at the start of each practice trial did not directly influence the offline contextualization measure [30] (Figure 5 – figure supplement 2A, 1st vs. 2nd Sequence approaches).”

Discussion (lines 409-412):

“This result remained unchanged when measuring offline contextualization between the last and second sequence of consecutive trials, inconsistent with a possible confounding effect of pre-planning [30] (Figure 5 – figure supplement 2A).”

In summary, none of the presented evidence in this paper—including results of the multiple control analyses carried out in response to the Reviewers’ recommendations— supports the Reviewer’s position.

Please note that the micro-offline learning "inference" has extensive mechanistic support across species and neural recording techniques (see Introduction, lines 26-56). In contrast, the reactive inhibition "inference," which is the Reviewer's alternative interpretation, has no such support yet [15].

Introduction (Lines 26-56)

“Practicing a new motor skill elicits rapid performance improvements (early learning) [1] that precede skill performance plateaus [5]. Skill gains during early learning accumulate over rest periods (micro-offline) interspersed with practice [1, 6-10], and are up to four times larger than offline performance improvements reported following overnight sleep [1]. During this initial interval of prominent learning, retroactive interference immediately following each practice interval reduces learning rates relative to interference after passage of time, consistent with stabilization of the motor memory [11]. Micro-offline gains observed during early learning are reproducible [7, 10-13] and are similar in magnitude even when practice periods are reduced by half to 5 seconds in length, thereby confirming that they are not merely a result of recovery from performance fatigue [11]. Additionally, they are unaffected by the random termination of practice periods, which eliminates the possibility of predictive motor slowing as a contributing factor [11]. Collectively, these behavioral findings point towards the interpretation that microoffline gains during early learning represent a form of memory consolidation [1].

This interpretation has been further supported by brain imaging and electrophysiological studies linking known memory-related networks and consolidation mechanisms to rapid offline performance improvements. In humans, the rate of hippocampo-neocortical neural replay predicts micro-offline gains [6].

Consistent with these findings, Chen et al. [12] and Sjøgård et al. [13] furnished direct evidence from intracranial human EEG studies, demonstrating a connection between the density of hippocampal sharp-wave ripples (80-120 Hz)—recognized markers of neural replay—and micro-offline gains during early learning. Further, Griffin et al. reported that neural replay of task-related ensembles in the motor cortex of macaques during brief rest periods— akin to those observed in humans [1, 6-8, 14]—are not merely correlated with, but are causal drivers of micro-offline learning [15]. Specifically, the same reach directions that were replayed the most during rest breaks showed the greatest reduction in path length (i.e. – more efficient movement path between two locations in the reach sequence) during subsequent trials, while stimulation applied during rest intervals preceding performance plateau reduced reactivation rates and virtually abolished micro-offline gains [15]. Thus, converging evidence in humans and non-human primates across indirect non-invasive and direct invasive recording techniques link hippocampal activity, neural replay dynamics and offline skill gains in early motor learning that precede performance plateau.”

That said, absence of evidence, is not evidence of absence and for that reason we also state in the Discussion (lines 448-452):

A simple control analysis based on shuffled class labels could lend further support to the authors' complex decoding approach. As a control analysis that completely rules out any source of overfitting, the authors could test the decoder after shuffling class labels. Following such shuffling, decoding accuracies should drop to chance-level for all decoding approaches, including the optimized decoder. This would also provide an estimate of actual chance-level performance (which is informative over and beyond the theoretical chance level). During the review process, the authors reported this analysis to the reviewers. Given that readers may consider following the presented decoding approach in their own work, it would have been important to include that control analysis in the manuscript to convince readers of its validity.

As requested, the label-shuffling analysis was carried out for both 4- and 5-class decoders and is now reported in the revised manuscript.

Results (lines 204-207):

“Testing the keypress state (4-class) hybrid decoder performance on Day 1 after randomly shuffling keypress labels for held-out test data resulted in a performance drop approaching expected chance levels (22.12%± SD 9.1%; Figure 3 – figure supplement 3C).”

Results (lines 261-264):

“As expected, the 5-class hybrid-space decoder performance approached chance levels when tested with randomly shuffled keypress labels (18.41%± SD 7.4% for Day 1 data; Figure 4 – figure supplement 3C).”

Furthermore, the authors' approach to cortical parcellation raises questions regarding the information carried by varying dipole orientations within a parcel (which currently seems to be ignored?) and the implementation of the mean-flipping method (given that there are two dimensions - space and time - it is unclear what the authors refer to when they talk about the sign of the "average source", line 477).

The revised manuscript now provides a more detailed explanation of the parcellation, and sign-flipping procedures implemented:

Methods (lines 604-611):

“Source-space parcellation was carried out by averaging all voxel time-series located within distinct anatomical regions defined in the Desikan-Killiany Atlas [31]. Since source time-series estimated with beamforming approaches are inherently sign-ambiguous, a custom Matlab-based implementation of the *mne.extract_label_time_course* with “*mean_flip”* sign-flipping procedure in MNEPython [78] was applied prior to averaging to prevent within-parcel signal cancellation. All voxel time-series within each parcel were extracted and the timeseries sign was flipped at locations where the orientation difference was greater than 90° from the parcel mode. A mean time-series was then computed across all voxels within the parcel after sign-flipping.”

**Recommendations for the authors:**

**Reviewer #1 (Recommendations for the authors):**
Comments on the revision:The authors have made large efforts to address all concerns raised. A couple of suggestions remain:- formally show if and how movement artefacts may contribute to the signal and analysis; it seems that the authors have data to allow for such an analysis

We have implemented the requested control analyses addressing this issue. They are reported in: Results (lines 207-211 and 261-268), Discussion (Lines 362-368):

- formally show that the signals from the intra- and inter parcel spaces are orthogonal.

Please note that, despite the Reviewer’s statement above, we never claim in the manuscript that the parcel-space and regional voxel-space features show “complete independence”.

Furthermore, the machine learning-based decoding methods used in the present study do not require input feature orthogonality, but instead non-redundancy [7], which is a requirement satisfied by our data (see below and the new Figure 2 – figure supplement 2 in the revised manuscript). Finally, our results already show that the hybrid space decoder outperformed all other methods even after input features were fully orthogonalized with LDA or PCA dimensionality reduction procedures prior to the classification step (Figure 3 – figure supplement 2).

We also highlight several additional results that are informative regarding this issue. For example, if spatially overlapping parcel- and voxel-space time-series only provided redundant information, inclusion of both as input features should increase model overfitting to the training dataset and decrease overall cross-validated test accuracy [8]. In the present study however, we see the opposite effect on decoder performance. First, Figure 3 – figure supplements 1 & 2 clearly show that decoders constructed from hybrid-space features outperform the other input feature (sensor-, whole-brain parcel- and whole-brain voxel-) spaces in every case (e.g. – wideband, all narrowband frequency ranges, and even after the input space is fully orthogonalized through dimensionality reduction procedures prior to the decoding step). Furthermore, Figure 3 – figure supplement 6 shows that hybridspace decoder performance supers when parcel-time series that spatially overlap with the included regional voxel-spaces are removed from the input feature set. We state in the Discussion (lines 353-356)

“The observation of increased cross-validated test accuracy (as shown in Figure 3 – Figure Supplement 6) indicates that the spatially overlapping information in parcel- and voxel-space time-series in the hybrid decoder was complementary, rather than redundant [41].”

To gain insight into the complimentary information contributed by the two spatial scales to the hybrid-space decoder, we first independently computed the matrix rank for whole-brain parcel- and voxel-space input features for each participant (shown in Author response image 1). The results indicate that whole-brain parcel-space input features are full rank (rank = 148) for all participants (i.e. - MEG activity is orthogonal between all parcels). The matrix rank of voxelspace input features (rank = 267± 17 SD), exceeded the parcel-space rank for all participants and approached the number of useable MEG sensor channels (n = 272). Thus, voxel-space features provide both additional and complimentary information to representations at the parcel-space scale.

Figure 2—figure Supplement 2 in the revised manuscript now shows that the degree of dependence between the two spatial scales varies over the regional voxel-space. That is, some voxels within a given parcel correlate strongly with the time-series of the parcel they belong to, while others do not. This finding is consistent with a documented increase in correlational structure of neural activity across spatial scales that does not reflect perfect dependency or orthogonality [9]. Notably, the regional voxel-spaces included in the hybridspace decoder are significantly less correlated with the averaged parcel-space time-series than excluded voxels. We now point readers to this new figure in the results.

Taken together, these results indicate that the multi-scale information in the hybrid feature set is complimentary rather than orthogonal. This is consistent with the idea that hybridspace features better represent multi-scale temporospatial dynamics reported to be a fundamental characteristic of how the brain stores and adapts memories, and generates behavior across species [9].

**Reviewer #2 (Recommendations for the authors):**
I appreciate the authors' efforts in addressing the concerns I raised. The responses generally made sense to me. However, I had some trouble finding several corrections/additions that the authors claim they made in the revised manuscript:"We addressed this question by conducting a new multivariate regression analysis to directly assess whether the neural representation distance score could be predicted by the 4-1, 2-4, and 4-4 keypress transition times observed for each complete correct sequence (both predictor and response variables were z-score normalized within-subject). The results of this analysis also affirmed that the possible alternative explanation that contextualization effects are simple reflections of increased mixing is not supported by the data (Adjusted R^2^ = 0.00431; F = 5.62). We now include this new negative control analysis in the revised manuscript."

This approach is now reported in the manuscript in the Results (Lines 324-328 and Figure 5-Figure Supplement 6 legend).

"We strongly agree with the Reviewer that the issue of generalizability is extremely important and have added a new paragraph to the Discussion in the revised manuscript highlighting the strengths and weaknesses of our study with respect to this issue."

Discussion (Lines 436-441)

“One limitation of this study is that contextualization was investigated for only one finger movement (index finger or digit 4) embedded within a relatively short 5-item skill sequence. Determining if representational contextualization is exhibited across multiple finger movements embedded within for example longer sequences (e.g. – two index finger and two little finger keypresses performed within a short piece of piano music) will be an important extension to the present results.”

"We strongly agree with the Reviewer that any intended clinical application must carefully consider the specific input feature constraints dictated by the clinical cohort, and in turn impose appropriate and complimentary constraints on classifier parameters that may differ from the ones used in the present study. We now highlight this issue in the Discussion of the revised manuscript and relate our present findings to published clinical BCI work within this context."

Discussion (Lines 441-444)

“While a supervised manifold learning approach (LDA) was used here because it optimized hybrid-space decoder performance, unsupervised strategies (e.g. - PCA and MDS, which also substantially improved decoding accuracy in the present study; Figure 3 – figure supplement 2) are likely more suitable for real-time BCI applications.”

and"The Reviewer makes a good point. We have now implemented the suggested normalization procedure in the analysis provided in the revised manuscript."

Results (lines 275-282)

“We used a Euclidian distance measure to evaluate the differentiation of the neural representation manifold of the same action (i.e. - an index-finger keypress) executed within different local sequence contexts (i.e. - ordinal position 1 vs. ordinal position 5; Figure 5). To make these distance measures comparable across participants, a new set of classifiers was then trained with group-optimal parameters (i.e. – broadband hybrid-space MEG data with subsequent manifold extraction Figure 3 – figure supplements 2) and LDA classifiers (Figure 3 – figure supplements 7) trained on 200ms duration windows aligned to the KeyDown event (see Methods, Figure 3 – figure supplements 5). “

Where are they in the manuscript? Did I read the wrong version? It would be more helpful to specify with page/line numbers. Please also add the detailed procedure of the control/additional analyses in the Method.

As requested, we now refer to all manuscript revisions with specific line numbers. We have also included all detailed procedures related to any additional analyses requested by reviewers.

I also have a few other comments back to the authors' following responses:"Thus, increased overlap between the "4" and "1" keypresses (at the start of the sequence) and "2" and "4" keypresses (at the end of the sequence) could artefactually increase contextualization distances even if the underlying neural representations for the individual keypresses remain unchanged. One must also keep in mind that since participants repeat the sequence multiple times within the same trial, a majority of the index finger keypresses are performed adjacent to one another (i.e. - the "4-4" transition marking the end of one sequence and the beginning of the next). Thus, increased overlap between consecutive index finger keypresses as typing speed increased should increase their similarity and mask contextualization- related changes to the underlying neural representations." "We also re-examined our previously reported classification results with respect to this issue.We reasoned that if mixing effects reflecting the ordinal sequence structure is an important driver of the contextualization finding, these effects should be observable in the distribution of decoder misclassifications. For example, "4" keypresses would be more likely to be misclassified as "1" or "2" keypresses (or vice versa) than as "3" keypresses. The confusion matrices presented in Figures 3C and 4B and Figure 3-figure supplement 3A display a distribution of misclassifications that is inconsistent with an alternative mixing effect explanation of contextualization.""Based upon the increased overlap between adjacent index finger keypresses (i.e. - "4-4" transition), we also reasoned that the decoder tasked with separating individual index finger keypresses into two distinct classes based upon sequence position, should show decreased performance as typing speed increases. However, Figure 4C in our manuscript shows that this is not the case. The 2-class hybrid classifier actually displays improved classification performance over early practice trials despite greater temporal overlap. Again, this is inconsistent with the idea that the contextualization effect simply reflects increased mixing of individual keypress features."As the time window for MEG feature is defined after the onset of each press, it is more likely that the feature overlap is the current and the future presses, rather than the current and the past presses (of course the three will overlap at very fast typing speed). Therefore, for sequence 41324, if we note the planning-related processes by a Roman numeral, the overlapping features would be '4i', '1iii', '3ii', '2iv', and '4iv'. Assuming execution-related process (e.g., 1) and planning-related process (e.g., i) are not necessarily similar, especially in finer temporal resolution, the patterns for '4i' and '4iv' are well separated in terms of process 'i' and 'iv,' and this advantage will be larger in faster typing speed. This also applies to the other presses. Thus, the author's arguments about the masking of contextualization and misclassification due to pattern overlap seem odd. The most direct and probably easiest way to resolve this would be to use a shorter time window for the MEG feature. Some decrease in decoding accuracy in this case is totally acceptable for the science purpose.

The revised manuscript now includes analyses carried out with decoding time windows ranging from 50 to 250ms in duration. These additional results are now reported in:

Results (lines 258-268):

“The improved decoding accuracy is supported by greater differentiation in neural representations of the index finger keypresses performed at positions 1 and 5 of the sequence (Figure 4A), and by the trial-by-trial increase in 2-class decoding accuracy over early learning (Figure 4C) across different decoder window durations (Figure 4 – figure supplement 2). As expected, the 5-class hybrid-space decoder performance approached chance levels when tested with randomly shuffled keypress labels (18.41%± SD 7.4% for Day 1 data; Figure 4 – figure supplement 3C). Task-related eye movements did not explain these results since an alternate 5-class hybrid decoder constructed from three eye movement features (gaze position at the KeyDown event, gaze position 200ms later, and peak eye movement velocity within this window; Figure 4 – figure supplement 3A) performed at chance levels (crossvalidated test accuracy = 0.2181; Figure 4 – figure supplement 3B, C).”

Results (lines 310-316):

“Offline contextualization strongly correlated with cumulative micro-offline gains (r = 0.903, R² = 0.816, p < 0.001; Figure 5 – figure supplement 1A, inset) across decoder window durations ranging from 50 to 250ms (Figure 5 – figure supplement 1B, C). The offline contextualization between the final sequence of each trial and the second sequence of the subsequent trial (excluding the first sequence) yielded comparable results. This indicates that pre-planning at the start of each practice trial did not directly influence the offline contextualization measure [30] (Figure 5 – figure supplement 2A, 1st vs. 2nd Sequence approaches). “

Discussion (lines 380-385):

“The first hint of representational differentiation was the highest false-negative and lowest false-positive misclassification rates for index finger keypresses performed at different locations in the sequence compared with all other digits (Figure 3C). This was further supported by the progressive differentiation of neural representations of the index finger keypress (Figure 4A) and by the robust trial-by-trial increase in 2class decoding accuracy across time windows ranging between 50 and 250ms (Figure 4C; Figure 4 – figure supplement 2).”

Discussion (lines 408-9):

“Offline contextualization consistently correlated with early learning gains across a range of decoding windows (50–250ms; Figure 5 – figure supplement 1).”

"We addressed this question by conducting a new multivariate regression analysis to directly assess whether the neural representation distance score could be predicted by the 4-1, 2-4 and 4-4 keypress transition times observed for each complete correct sequence"For regression analysis, I recommend to use total keypress time per a sequence (or sum of 4-1 and 4-4) instead of specific transition intervals, because there likely exist specific correlational structure across the transition intervals. Using correlated regressors may distort the result.

This approach is now reported in the manuscript:

Results (Lines 324-328) and Figure 5-Figure Supplement 6 legend.

"We do agree with the Reviewer that the naturalistic, generative, self-paced task employed in the present study results in overlapping brain processes related to planning, execution, evaluation and memory of the action sequence. We also agree that there are several tradeoffs to consider in the construction of the classifiers depending on the study aim. Given our aim of optimizing keypress decoder accuracy in the present study, the set of tradeoffs resulted in representations reflecting more the latter three processes, and less so the planning component. Whether separate decoders can be constructed to tease apart the representations or networks supporting these overlapping processes is an important future direction of research in this area. For example, work presently underway in our lab constrains the selection of windowing parameters in a manner that allows individual classifiers to be temporally linked to specific planning, execution, evaluation or memoryrelated processes to discern which brain networks are involved and how they adaptively reorganize with learning. Results from the present study (Figure 4-figure supplement 2) showing hybrid-space decoder prediction accuracies exceeding 74% for temporal windows spanning as little as 25ms and located up to 100ms prior to the KeyDown event strongly support the feasibility of such an approach."I recommend that the authors add this paragraph or a paragraph like this to the Discussion. This perspective is very important and still missing in the revised manuscript.

We now included in the manuscript the following sections addressing this point:

Discussion (lines 334-338)

“The main findings of this study during which subjects engaged in a naturalistic, self-paced task were that individual sequence action representations differentiate during early skill learning in a manner reflecting the local sequence context in which they were performed, and that the degree of representational differentiation— particularly prominent over rest intervals—correlated with skill gains. “

Discussion (lines 428-434)

“In this study, classifiers were trained on MEG activity recorded during or immediately after each keypress, emphasizing neural representations related to action execution, memory consolidation and recall over those related to planning. An important direction for future research is determining whether separate decoders can be developed to distinguish the representations or networks separately supporting these processes. Ongoing work in our lab is addressing this question. The present accuracy results across varied decoding window durations and alignment with each keypress action support the feasibility of this approach (Figure 3—figure supplement 5).”

"The rapid initial skill gains that characterize early learning are followed by micro-scale fluctuations around skill plateau levels (i.e. following trial 11 in Figure 1B)" Is this a mention of Figure 1 Supplement 1 A?

The sentence was replaced with the following: Results (lines 108-110)

“Participants reached 95% of maximal skill (i.e. - Early Learning) within the initial 11 practice trials (Figure 1B), with improvements developing over inter-practice rest periods (micro-offline gains) accounting for almost all total learning across participants (Figure 1B, inset) [1].”

The citation below seems to have been selected by mistake;"9. Chen, S. & Epps, J. Using task-induced pupil diameter and blink rate to infer cognitive load. Hum Comput Interact 29, 390-413 (2014)."

We thank the Reviewer for bringing this mistake to our attention. This citation has now been corrected.

**Reviewer #3 (Recommendations for the authors):**
The authors write in their response that "We now provide additional details in the Methods of the revised manuscript pertaining to the parcellation procedure and how the sign ambiguity problem was addressed in our analysis." I could not find anything along these lines in the (redlined) version of the manuscript and therefore did not change the corresponding comment in the public review.

The revised manuscript now provides a more detailed explanation of the parcellation, and sign-flipping procedure implemented:

Methods (lines 604-611):

“Source-space parcellation was carried out by averaging all voxel time-series located within distinct anatomical regions defined in the Desikan-Killiany Atlas [31]. Since source time-series estimated with beamforming approaches are inherently sign-ambiguous, a custom Matlab-based implementation of the *mne.extract_label_time_course* with “*mean_flip”* sign-flipping procedure in MNEPython [78] was applied prior to averaging to prevent within-parcel signal cancellation. All voxel time-series within each parcel were extracted and the timeseries sign was flipped at locations where the orientation difference was greater than 90° from the parcel mode. A mean time-series was then computed across all voxels within the parcel after sign-flipping.”

The control analysis based on a multivariate regression that assessed whether the neural representation distance score could be predicted by the 4-1, 2-4 and 4-4 keypress transition times, as briefly mentioned in the authors' responses to Reviewer 2 and myself, was not included in the manuscript and could not be sufficiently evaluated.

This approach is now reported in the manuscript: Results (Lines 324-328) and Figure 5-Figure Supplement 6 legend.

The authors argue that differences in the design between Das et al. (2024) on the one hand (Experiments 1 and 2), and the study by Bönstrup et al. (2019) on the other hand, may have prevented Das et al. (2024) from finding the assumed learning benefit by micro-offline consolidation. However, the Supplementary Material of Das et al. (2024) includes an experiment (Experiment S1) whose design closely follows a large proportion of the early learning phase of Bönstrup et al. (2019), and which, nevertheless, demonstrates that there is no lasting benefit of taking breaks with respect to the acquired skill level, despite the presence of micro-offline gains.

We thank the Reviewer for alerting us to this new data added to the revised supplementary materials of Das et al. (2024) posted to bioRxiv. However, despite the Reviewer’s claim to the contrary, a careful comparison between the Das et al and Bönstrup et al studies reveal more substantive differences than similarities and does not “closely follows a large proportion of the early learning phase of Bönstrup et al. (2019)” as stated.

In the Das et al. Experiment S1, sixty-two participants were randomly assigned to “with breaks” or “no breaks” skill training groups. The “with breaks” group alternated 10 seconds of skill sequence practice with 10 seconds of rest over seven trials (2 min and 2 sec total training duration). This amounts to 66.7% of the early learning period defined by Bönstrup et al. (2019) (i.e. - eleven 10-second long practice periods interleaved with ten 10-second long rest breaks; 3 min 30 sec total training duration). Also, please note that while no performance feedback nor reward was given in the Bönstrup et al. (2019) study, participants in the Das et al. study received explicit performance-based monetary rewards, a potentially crucial driver of differentiated behavior between the two studies:

“Participants were incentivized with bonus money based on the total number of correct sequences completed throughout the experiment.”

The “no breaks” group in the Das et al. study practiced the skill sequence for 70 continuous seconds. Both groups (despite one being labeled “no breaks”) follow training with a long 3-minute break (also note that since the “with breaks” group ends with 10 seconds of rest their break is actually longer), before finishing with a skill “*test*” over a continuous 50-second-long block. During the 70 seconds of training, the “with breaks” group shows more learning than the “no breaks” group. Interestingly, following the long 3minute break the “with breaks” group display a performance drop (relative to their performance at the end of training) that is stable over the full 50-second test, while the “no breaks” group shows an immediate performance improvement following the long break that continues to increase over the 50-second test.

Separately, there are important issues regarding the Das et al study that should be considered through the lens of recent findings not referred to in the preprint. A major element of their experimental design is that both groups—“with breaks” and “no breaks”— actually receive quite a long 3-minute break just before the skill test. This long break is more than 2.5x the cumulative interleaved rest experienced by the “with breaks” group. Thus, although the design is intended to contrast the presence or absence of rest “breaks”, that difference between groups is no longer maintained at the point of the skill test.

The Das et al results are most consistent with an alternative interpretation of the data— that the “no breaks” group experiences offline learning during their long 3-minute break. This is supported by the recent work of Griffin et al. (2025) where micro-array recordings from primary and premotor cortex were obtained from macaque monkeys while they performed blocks of ten continuous reaching sequences up to 81.4 seconds in duration (see source data for Extended Data Figure 1h) with 90 seconds of interleaved rest. Griffin et al. observed offline improvement in skill immediately following the rest break that was causally related to neural reactivations (i.e. – neural replay) that occurred *during* the rest break. Importantly, the highest density of reactivations was present in the very first 90second break between Blocks 1 and 2 (see Fig. 2f in Griffin et al., 2025). This supports the interpretation that both the “with breaks” and “no breaks” group express offline learning gains, with these gains being delayed in the “no breaks” group due to the practice schedule.

On the other hand, if offline learning can occur during this longer break, then why would the “with breaks” group show no benefit? Again, it could be that most of the offline gains for this group were front-loaded during the seven shorter 10-second rest breaks. Another possible, though not mutually exclusive, explanation is that the observed drop in performance in the “with breaks” group is driven by contextual interference. Specifically, similar to Experiments 1 and 2 in Das et al. (2024), the skill test is conducted under very different conditions than those which the “with breaks” group practiced the skill under (short bursts of practiced alternating with equally short breaks). On the other hand, the “no breaks” group is tested (50 seconds of continuous practice) under quite similar conditions to their training schedule (70 seconds of continuous practice). Thus, it is possible that this dissimilarity between training and test could lead to reduced performance in the “with breaks” group.

We made the following manuscript revisions related to these important issues:

Introduction (Lines 26-56)

“Practicing a new motor skill elicits rapid performance improvements (early learning) [1] that precede skill performance plateaus [5]. Skill gains during early learning accumulate over rest periods (micro-offline) interspersed with practice [1, 6-10], and are up to four times larger than offline performance improvements reported following overnight sleep [1]. During this initial interval of prominent learning, retroactive interference immediately following each practice interval reduces learning rates relative to interference after passage of time, consistent with stabilization of the motor memory [11]. Micro-offline gains observed during early learning are reproducible [7, 10-13] and are similar in magnitude even when practice periods are reduced by half to 5 seconds in length, thereby confirming that they are not merely a result of recovery from performance fatigue [11]. Additionally, they are unaffected by the random termination of practice periods, which eliminates the possibility of predictive motor slowing as a contributing factor [11]. Collectively, these behavioral findings point towards the interpretation that microoffline gains during early learning represent a form of memory consolidation [1].

This interpretation has been further supported by brain imaging and electrophysiological studies linking known memory-related networks and consolidation mechanisms to rapid offline performance improvements. In humans, the rate of hippocampo-neocortical neural replay predicts micro-offline gains [6]. Consistent with these findings, Chen et al. [12] and Sjøgård et al. [13] furnished direct evidence from intracranial human EEG studies, demonstrating a connection between the density of hippocampal sharp-wave ripples (80-120 Hz)—recognized markers of neural replay—and micro-offline gains during early learning. Further, Griffin et al. reported that neural replay of task-related ensembles in the motor cortex of macaques during brief rest periods— akin to those observed in humans [1, 6-8, 14]—are not merely correlated with, but are causal drivers of micro-offline learning [15]. Specifically, the same reach directions that were replayed the most during rest breaks showed the greatest reduction in path length (i.e. – more efficient movement path between two locations in the reach sequence) during subsequent trials, while stimulation applied during rest intervals preceding performance plateau reduced reactivation rates and virtually abolished micro-offline gains [15]. Thus, converging evidence in humans and non-human primates across indirect non-invasive and direct invasive recording techniques link hippocampal activity, neural replay dynamics and offline skill gains in early motor learning that precede performance plateau.”

Next, in the Methods, we articulate important constraints formulated by Pan and Rickard (2015) and Bönstrup et al. (2019) for meaningful measurements:

Methods (Lines 493-499)

“The study design followed specific recommendations by Pan and Rickard (2015): (1) utilizing 10-second practice trials and (2) constraining analysis of micro-offline gains to early learning trials (where performance monotonically increases and 95% of overall performance gains occur) that precede the emergence of “scalloped” performance dynamics strongly linked to reactive inhibition effects ([29, 72]). This is precisely the portion of the learning curve Pan and Rickard referred to when they stated “…rapid learning during that period masks any reactive inhibition effect” [29].”

We finally discuss the implications of neglecting some or all of these recommendations:

Discussion (Lines 444-452):

“Finally, caution should be exercised when extrapolating findings during early skill learning, a period of steep performance improvements, to findings reported after insufficient practice [67], post-plateau performance periods [68], or non-learning situations (e.g. performance of non-repeating keypress sequences in [67]) when reactive inhibition or contextual interference effects are prominent. Ultimately, it will be important to develop new paradigms allowing one to independently estimate the different coincident or antagonistic features (e.g. - memory consolidation, planning, working memory and reactive inhibition) contributing to micro-online and micro-offline gains during and after early skill learning within a unifying framework.”

Personally, given that the idea of (micro-offline) consolidation seems to attract a lot of interest (and therefore cause a lot of future effort/cost public money) in the scientific community, I would find it extremely important to be cautious in interpreting results in this field. For me, this would include abstaining from the claim that processes occur "during" a rest period (see abstract, for example), given that micro-offline gains (as well as offline contextualization) are computed from data obtained during practice, not rest, and may, thus, just as well reflect a change that occurs "online", e.g., at the very onset of practice (like pre-planning) or throughout practice (like fatigue, or reactive inhibition). In addition, I would suggest to discuss in more depth the actual evidence not only in favour, but also against, the assumption of micro-offline gains as a phenomenon of learning.

We agree with the reviewer that caution is warranted. Based upon these suggestions, we have now expanded the manuscript to very clearly define the experimental constraints under which different groups have successfully studied micro-offline learning and its mechanisms, the impact of fatigue/reactive inhibition on micro-offline performance changes unrelated to learning, as well as the interpretation problems that emerge when those recommendations are not followed.

We clearly articulate the crucial constrains recommended by Pan and Rickard (2015) and Bönstrup et al. (2019) for meaningful measurements and interpretation of offline gains in the revised manuscript.

Methods (Lines 493-499)

“The study design followed specific recommendations by Pan and Rickard (2015): (1) utilizing 10-second practice trials and (2) constraining analysis of micro-offline gains to early learning trials (where performance monotonically increases and 95% of overall performance gains occur) that precede the emergence of “scalloped” performance dynamics strongly linked to reactive inhibition effects ([29, 72]). This is precisely the portion of the learning curve Pan and Rickard referred to when they stated “…rapid learning during that period masks any reactive inhibition effect” [29].”

In the Introduction, we review the extensive evidence emerging from LFP and microelectrode recordings in humans and monkeys (including causality of neural replay with respect to micro-offline gains and early learning in the Griffin et al. Nature 2025 publication):

Introduction (Lines 26-56)

“Practicing a new motor skill elicits rapid performance improvements (early learning) [1] that precede skill performance plateaus [5]. Skill gains during early learning accumulate over rest periods (micro-offline) interspersed with practice [1, 6-10], and are up to four times larger than offline performance improvements reported following overnight sleep [1]. During this initial interval of prominent learning, retroactive interference immediately following each practice interval reduces learning rates relative to interference after passage of time, consistent with stabilization of the motor memory [11]. Micro-offline gains observed during early learning are reproducible [7, 10-13] and are similar in magnitude even when practice periods are reduced by half to 5 seconds in length, thereby confirming that they are not merely a result of recovery from performance fatigue [11]. Additionally, they are unaffected by the random termination of practice periods, which eliminates the possibility of predictive motor slowing as a contributing factor [11]. Collectively, these behavioral findings point towards the interpretation that microoffline gains during early learning represent a form of memory consolidation [1].

This interpretation has been further supported by brain imaging and electrophysiological studies linking known memory-related networks and consolidation mechanisms to rapid offline performance improvements. In humans, the rate of hippocampo-neocortical neural replay predicts micro-offline gains [6]. Consistent with these findings, Chen et al. [12] and Sjøgård et al. [13] furnished direct evidence from intracranial human EEG studies, demonstrating a connection between the density of hippocampal sharp-wave ripples (80-120 Hz)—recognized markers of neural replay—and micro-offline gains during early learning. Further, Griffin et al. reported that neural replay of task-related ensembles in the motor cortex of macaques during brief rest periods— akin to those observed in humans [1, 6-8, 14]—are not merely correlated with, but are causal drivers of micro-offline learning [15]. Specifically, the same reach directions that were replayed the most during rest breaks showed the greatest reduction in path length (i.e. – more efficient movement path between two locations in the reach sequence) during subsequent trials, while stimulation applied during rest intervals preceding performance plateau reduced reactivation rates and virtually abolished micro-offline gains [15]. Thus, converging evidence in humans and non-human primates across indirect non-invasive and direct invasive recording techniques link hippocampal activity, neural replay dynamics and offline skill gains in early motor learning that precede performance plateau.”

Following the reviewer’s advice, we have expanded our discussion in the revised manuscript of alternative hypotheses put forward in the literature and call for caution when extrapolating results across studies with fundamental differences in design (e.g. – different practice and rest durations, or presence/absence of extrinsic reward, etc).

Discussion (Lines 444-452):

“Finally, caution should be exercised when extrapolating findings during early skill learning, a period of steep performance improvements, to findings reported after insufficient practice [67], post-plateau performance periods [68], or non-learning situations (e.g. performance of non-repeating keypress sequences in [67]) when reactive inhibition or contextual interference effects are prominent. Ultimately, it will be important to develop new paradigms allowing one to independently estimate the different coincident or antagonistic features (e.g. - memory consolidation, planning, working memory and reactive inhibition) contributing to micro-online and micro-offline gains during and after early skill learning within a unifying framework.”

References

(1) Zimerman, M., et al., Disrupting the Ipsilateral Motor Cortex Interferes with Training of a Complex Motor Task in Older Adults. Cereb Cortex, 2012.

(2) Waters, S., T. Wiestler, and J. Diedrichsen, Cooperation Not Competition: Bihemispheric tDCS and fMRI Show Role for Ipsilateral Hemisphere in Motor Learning. J Neurosci, 2017. 37(31): p. 7500-7512.

(3) Sawamura, D., et al., Acquisition of chopstick-operation skills with the nondominant hand and concomitant changes in brain activity. Sci Rep, 2019. 9(1): p. 20397.

(4) Lee, S.H., S.H. Jin, and J. An, The dieerence in cortical activation pattern for complex motor skills: A functional near- infrared spectroscopy study. Sci Rep, 2019. 9(1): p. 14066.

(5) Grafton, S.T., E. Hazeltine, and R.B. Ivry, Motor sequence learning with the nondominant left hand. A PET functional imaging study. Exp Brain Res, 2002. 146(3): p. 369-78.

(6) Buch, E.R., et al., Consolidation of human skill linked to waking hippocamponeocortical replay. Cell Rep, 2021. 35(10): p. 109193.

(7) Wang, L. and S. Jiang, A feature selection method via analysis of relevance, redundancy, and interaction, in Expert Systems with Applications, Elsevier, Editor. 2021.

(8) Yu, L. and H. Liu, Eeicient feature selection via analysis of relevance and redundancy. Journal of Machine Learning Research, 2004. 5: p. 1205-1224.

(9) Munn, B.R., et al., Multiscale organization of neuronal activity unifies scaledependent theories of brain function. Cell, 2024.

(10) Borragan, G., et al., Sleep and memory consolidation: motor performance and proactive interference eeects in sequence learning. Brain Cogn, 2015. 95: p. 54-61.

(11) Landry, S., C. Anderson, and R. Conduit, The eeects of sleep, wake activity and timeon-task on oeline motor sequence learning. Neurobiol Learn Mem, 2016. 127: p. 5663.

(12) Gabitov, E., et al., Susceptibility of consolidated procedural memory to interference is independent of its active task-based retrieval. PLoS One, 2019. 14(1): p. e0210876.

(13) Pan, S.C. and T.C. Rickard, Sleep and motor learning: Is there room for consolidation? Psychol Bull, 2015. 141(4): p. 812-34.

(14) , M., et al., A Rapid Form of Oeline Consolidation in Skill Learning. Curr Biol, 2019. 29(8): p. 1346-1351 e4.

(15) Gupta, M.W. and T.C. Rickard, Comparison of online, oeline, and hybrid hypotheses of motor sequence learning using a quantitative model that incorporate reactive inhibition. Sci Rep, 2024. 14(1): p. 4661.